# BIAS–VARIANCE TRADEOFF IN TENSOR ESTIMATION

## ABSTRACT

We study denoising of a third-order tensor when the ground-truth tensor is **not** necessarily Tucker low-rank. Specifically, we observe

$$Y = X^* + Z \in \mathbb{R}^{p_1 \times p_2 \times p_3},$$

where $X^*$ is the ground-truth tensor, and $Z$ is the noise tensor. We propose a simple variant of the higher-order SVD (HOSVD) estimator, denoted $\widetilde{X}$. We show that, uniformly over any user-specified Tucker ranks $(r_1, r_2, r_3)$ for $\widetilde{X}$, with high probability,

$$\|\widetilde{X} - X^*\|_{\mathrm{F}}^2 = O\Big(\kappa^2\Big\{r_1 r_2 r_3 + \sum_{k=1}^{3} p_k r_k\Big\} + \xi_{(r_1, r_2, r_3)}^2\Big).$$

Here, the bias term $\xi_{(r_1, r_2, r_3)}$ corresponds to the best achievable approximation error of $X^*$ over the class of tensors with Tucker ranks $(r_1, r_2, r_3)$; $\kappa^2$ quantifies the noise level; and the variance term $\kappa^2\{r_1 r_2 r_3 + \sum_{k=1}^{3} p_k r_k\}$ scales with the effective number of free parameters in the estimator $\widetilde{X}$. Our analysis achieves a clean rank-adaptive bias–variance tradeoff: as we increase the ranks of estimator $\widetilde{X}$, the bias $\xi(r_1, r_2, r_3)$ decreases and the variance increases. As a byproduct we also obtain a convenient bias-variance decomposition for the vanilla low-rank SVD matrix estimators.

## 1 INTRODUCTION

Low-rank tensor models are a cornerstone of modern machine learning. They capture essential structure in high-dimensional signals while offering substantial gains in computation and storage. Applications include recommender systems (Koren et al., 2009), topic modeling (Blei et al., 2003), community detection (Abbe, 2018; Diakonikolas & Kane, 2024), and, more recently, latent variable learning (Anandkumar et al., 2014; Sherman & Kolda, 2020; Zhang & Kileel, 2023; Diakonikolas & Kane, 2024) and generative modeling (Hur et al., 2023; Peng et al., 2023). The effectiveness of these methods demonstrate that many real-world signals embedded in high-dimensional ambient spaces can be well-approximated by low-rank structures.

The study of exactly low-rank estimation is well-developed, particularly in the matrix setting. A large body of work has introduced efficient algorithms for tasks such as matrix completion and denoising, including nuclear norm relaxation (Koltchinskii et al., 2011), convex optimization (Candes & Recht, 2012), nonconvex methods (Chi et al., 2019), and SVD guarantees with unbalanced matrices (Cai & Zhang, 2018).

Parallel efforts in low-rank tensor estimation are well-developed, including the study of higher-order tensor singular value decompositions in the non-random setting (De Lathauwer et al., 2000b;a), symmetric tensor decomposition in non-random setting (Jin et al., 2024), and Riemannian optimization methods for low-rank Tucker tensor completion (Kressner et al., 2014). Nevertheless, the tensor setting poses additional challenges: many fundamental problems are computationally intractable, multiple inequivalent notions of rank (e.g. CP and Tucker) exist, and best low-rank approximations need not be unique (e.g. Hillar & Lim, 2013). Most theoretical guarantees in tensor estimation (e.g. Anandkumar et al., 2017; Zhang & Xia, 2018; Han et al., 2022), like their matrix counterparts, rely on the assumption that the underlying signal is exactly low-rank.

In practice, however, signals are not exactly low-rank: their singular spectra typically decay gradually rather than vanishing at a finite rank. This creates an interesting gap between theory and practice, since existing analyses provide guarantees under idealized exact low-rankness and leave open the question of how to analyze estimation error in more realistic settings.

We address this gap by designing and analyzing low-rank tensor estimation when the signal is not low-rank. In such settings, estimation error is governed by a *bias–variance tradeoff*: truncating to a smaller rank incurs approximation bias, while larger ranks inflate variance by amplifying noise. In particular, we propose a simple variant of the higher-order singular value decomposition (HOSVD) algorithm and provide the analysis of tensor estimation without assuming low-rankness. Our main result (Theorem 1) establishes an explicit *bias–variance decomposition* of the estimation error:

- The *bias term* quantifies the approximation error incurred by truncating a spectrum.
- The *variance term* matches the optimal rate for exactly low-rank models.

The success of SVD and HOSVD has been well established by their decades of widespread use. The objective of our manuscript is to provide mathematical justification for the information-efficient bias-variance tradeoff achieved by these two classical algorithms, and to establish uniform performance guarantees that hold for any user-specified ranks.

To achieve this clean bias-variance tradeoff, we utilize classical linear algebra results such as Mirsky's and Ky Fan's theorems to develop non-trivial extensions of classical perturbation tools to general tensors. As a convenient side result, we also recover a clean bias–variance characterization for matrices, thereby unifying the matrix and tensor perspectives. Finally, we validate our theory with simulations that confirm the existence of tradeoffs across different regimes of spectral decay and noise.

## 1.1 NOTATION

**Matrices**: For positive integers $p, r$, let $\mathbb{O}^{p \times r} = \{V \in \mathbb{R}^{p \times r} : V^\top V = I_r\}$, the set of column-orthonormal matrices. Let $M \in \mathbb{R}^{p \times q}$ and suppose the full singular value decomposition (SVD) satisfies $M = U\Sigma V^\top$. Here $s = \operatorname{rank}(M)$, $U \in \mathbb{O}^{p \times s}$, $V \in \mathbb{O}^{q \times s}$, and $\Sigma \in \mathbb{R}^{s \times s}$ is diagonal with singular values $\sigma_1(M) \geq \sigma_2(M) \geq \cdots \geq \sigma_s(M) \geq 0$. We write the smallest singular value of $M$ as $\sigma_{\min}(M)$. The operator norm and Frobenius norm of $M$ are defined as

$$\|M\| = \sigma_1(M), \qquad \|M\|_{\mathrm{F}} = \Big( \sum_{i=1}^p \sum_{j=1}^q M_{i,j}^2 \Big)^{1/2}.$$

For $r \leq \operatorname{rank}(M)$, the rank-$r$ truncated SVD is $M_{(r)} = U_{(r)}\Sigma_{(r)}V_{(r)}^\top$, where $U_{(r)} \in \mathbb{O}^{p \times r}$ and $V_{(r)} \in \mathbb{O}^{q \times r}$ corresponds to the leading $r$ left and right singular vectors respectively, and $\Sigma_{(r)} = \operatorname{diag}\{\sigma_1(M), \ldots, \sigma_r(M)\}$. For brevity, we use $\mathrm{SVD}_r(M)$ to denote the leading $r$ left singular vectors, i.e.

$$\mathrm{SVD}_r(M) = U_{(r)}.$$

For two matrices $M_1 \in \mathbb{R}^{p_1 \times q_1}$ and $M_2 \in \mathbb{R}^{p_2 \times q_2}$, their Kronecker product is $M_1 \otimes M_2 \in \mathbb{R}^{(p_1 p_2) \times (q_1 q_2)}$.

Let $U, \widehat{U} \in \mathbb{O}^{p \times r}$ be two singular subspaces. We write the principal angles between $U$ and $\widehat{U}$ as

$$\Theta(U, \widehat{U}) = \operatorname{diag}\{\arccos(\sigma_1(U^\top \widehat{U})), \ldots, \arccos(\sigma_r(U^\top \widehat{U}))\}.$$

We use $\|\sin\Theta(U, \widehat{U})\|$ and $\|\sin\Theta(U, \widehat{U})\|_{\mathrm{F}}$ to measure the distance between the two singular subspaces.

**Tensors**: For any tensor $B \in \mathbb{R}^{p_1 \times p_2 \times p_3}$, the Frobenius norm is

$$\|B\|_{\mathrm{F}} = \left( \sum_{i_1=1}^{p_1} \sum_{i_2=1}^{p_2} \sum_{i_3=1}^{p_3} B_{i_1,i_2,i_3}^2 \right)^{1/2}.$$

The mode-1 matricization $\mathcal{M}_1(B)$ is the unfolding of $B$ into a $p_1 \times p_2 p_3$ matrix. The mode-2 and mode-3 matricization of $B$ are defined similarly. The mode-1 product of $B$ with a matrix

$M \in \mathbb{R}^{q \times p_1}$ is defined as

$$(B \times_1 M)(j, i_2, i_3) = \sum_{i_1=1}^{p_1} M(i_1, j) B(i_1, i_2, i_3).$$

We write the Tucker rank of $B$ as $(r_1, r_2, r_3)$ if

$$\operatorname{rank}(\mathcal{M}_j(B)) = r_j, \text{ for } j = 1, 2, 3.$$

**Random variables**: For a random variable $X \in \mathbb{R}$, we denote the sub-Gaussian and sub-Exponential norms as

$$\|X\|_{\psi_2} = \inf\{K > 0 : \mathbb{E}\exp(X^2/K^2) \leq 2\} \quad \text{and} \quad \|X\|_{\psi_1} = \inf\{K > 0 : \mathbb{E}\exp(|X|/K) \leq 2\}.$$

We write $X \sim \mathrm{subGaussian}(0, \kappa^2)$ if $\mathbb{E}(X) = 0$ and $\|X\|_{\psi_2} \leq \kappa$.

**Universal constants**: We use $C_1, C_2, \ldots$ and $c_1, c_2, \ldots$ to denote positive constants whose values may differ from place to place.

## 2 BIAS–VARIANCE TRADEOFF IN TENSOR ESTIMATION

We consider the noisy tensor model

$$Y = X^* + Z \in \mathbb{R}^{p_1 \times p_2 \times p_3}, \tag{1}$$

where $X^*$ is an unknown signal and $Z$ is a random perturbation. Even when $X^*$ has a full Tucker rank $(p_1, p_2, p_3)$, it is common to approximate $X^*$ by a Tucker rank-$(r_1, r_2, r_3)$ estimator. This is because (i) low-rank structure yields dramatic gains in computation and memory, see Remark 2; and (ii) the approximation bias can decay quickly when approximating $X^*$ by a Tucker low-rank tensor, so balancing bias and variance may lead to a smaller estimation error. Given a target Tucker rank $(r_1, r_2, r_3)$, we consider the following simple variant of the HOSVD algorithm (De Lathauwer et al., 2000a).

---

**Algorithm 1** One-step HOSVD

**INPUT:** Tensor $Y$; target Tucker rank $(r_1, r_2, r_3)$.
    **for** $k = 1, 2, 3$ **do**
        $U_k^{(0)} \leftarrow \mathrm{SVD}_{r_k}(\mathcal{M}_k(Y)) \in \mathbb{O}^{p_k \times r_k}$.
    **end for**
$U_1^{(1)} \leftarrow \mathrm{SVD}_{r_1}\Big(\mathcal{M}_1(Y) \cdot \{U_2^{(0)} \otimes U_3^{(0)}\}\Big) \in \mathbb{O}^{p_1 \times r_1}$,
$U_2^{(1)} \leftarrow \mathrm{SVD}_{r_2}\Big(\mathcal{M}_2(Y) \cdot \{U_1^{(0)} \otimes U_3^{(0)}\}\Big) \in \mathbb{O}^{p_2 \times r_2}$,
$U_3^{(1)} \leftarrow \mathrm{SVD}_{r_3}\Big(\mathcal{M}_3(Y) \cdot \{U_1^{(0)} \otimes U_2^{(0)}\}\Big) \in \mathbb{O}^{p_3 \times r_3}$.
**OUTPUT:** $\widetilde{X} \leftarrow Y \times_1 U_1^{(1)} U_1^{(1)\top} \times_2 U_2^{(1)} U_2^{(1)\top} \times_3 U_3^{(1)} U_3^{(1)\top}$.

---

Let $\mathcal{T}_{(r_1, r_2, r_3)}$ denote the class of tensors in $\mathbb{R}^{p_1 \times p_2 \times p_3}$ with Tucker rank at most $(r_1, r_2, r_3)$, i.e.

$$\mathcal{T}_{(r_1, r_2, r_3)} = \big\{ A \in \mathbb{R}^{p_1 \times p_2 \times p_3} : \operatorname{rank}(\mathcal{M}_k(A)) \leq r_k, \ k = 1, 2, 3 \big\}. \tag{2}$$

For any tensor $X^* \in \mathbb{R}^{p_1 \times p_2 \times p_3}$, let $\xi_{(r_1, r_2, r_3)}$ denote the best Tucker rank-$(r_1, r_2, r_3)$ approximation error:

$$\xi_{(r_1, r_2, r_3)} = \inf_{A \in \mathcal{T}_{(r_1, r_2, r_3)}} \|A - X^*\|_{\mathrm{F}}.$$

**Theorem 1.** *Suppose $Y$ follows* (1) *with $Z_{\mu_1, \mu_2, \mu_3} \overset{\text{i.i.d.}}{\sim} \mathrm{subGaussian}(0, \kappa^2)$. Let $\widetilde{X}$ be the output of Algorithm 1 with target Tucker rank $(r_1, r_2, r_3)$. Define $r_{\max} = \max_k r_k$ and $p_{\min} = \min_k p_k$. Assume the singular gaps satisfy*

$$\big\{ \sigma_{r_k}(\mathcal{M}_k(X^*)) - \sigma_{r_k+1}(\mathcal{M}_k(X^*)) \big\}^2 \geq C_{\mathrm{gap}} \kappa^2 \Big( \sqrt{p_1 p_2 p_3 r_{\max}} + \sum_{k=1}^{3} p_k r_{\max} \Big), \tag{3}$$

*for all $k = 1, 2, 3$ and for a sufficiently large constant $C_{\mathrm{gap}} > 0$. Then, with probability at least $1 - C_1 p_1 p_2 p_3 \exp(-C_2 p_{\min})$,*

$$\|\widetilde{X} - X^*\|_{\mathrm{F}} \leq C_3 \left\{ \sqrt{\kappa^2 \left( \textstyle\sum_{k=1}^3 p_k r_k + r_1 r_2 r_3 \right)} + \xi_{(r_1, r_2, r_3)} \right\}. \tag{4}$$

**Remark 1** (Assumptions in Theorem 1). *In Theorem 1, we assume that the entries of the noise tensor $Z$ is i.i.d. sub-Gaussian. This is a commonly seen condition in the tensor literature (e.g. Zhang & Xia, 2018; Han et al., 2022).*

*The signal-to-noise ratio (SNR) condition (3) in Theorem 1 is also commonly seen in literature. In fact, if the ground truth $X^*$ is exactly Tucker low-rank with Tucker rank $(r_1, r_2, r_3)$, then $\sigma_{r_k+1}(\mathcal{M}_k(X^*)) = 0$ and (3) reduces to the SNR condition used in Zhang & Xia (2018), when $r_{\max}$ is a bounded constant.*

**Remark 2** (Computational efficiency). *Suppose $Y \in \mathbb{R}^{p_1 \times p_2 \times p_3}$ and let*

$$Y_{(r_1, r_2, r_3)} = S \times_1 U_1 \times_2 U_2 \times_3 U_3,$$

*be a Tucker rank-$(r_1, r_2, r_3)$ approximation of $Y$, where $U_k \in \mathbb{R}^{p_k \times r_k}$ satisfying $U_k^\top U_k = I_{r_k}$ for $k = 1, 2, 3$, and $S \in \mathbb{R}^{r_1 \times r_2 \times r_3}$ is the core tensor.*

*For dense multiplications with $v_k \in \mathbb{R}^{p_k}$ along each mode $k$, computing $Y \times_1 v_1 \times_2 v_2 \times_3 v_3$ costs $O(p_1 p_2 p_3)$ operations. In contrast, using the Tucker decomposition, we compute $Y_{(r_1, r_2, r_3)} \times_1 v_1 \times_2 v_2 \times_3 v_3$ successively as*

$$t_k = U_k^\top v_k \in \mathbb{R}^{r_k} \ (cost \approx 2 \sum_{k=1}^3 p_k r_k), \quad S' = S \times_1 t_1 \in \mathbb{R}^{r_2 \times r_3} \ (cost \approx 2 r_1 r_2 r_3),$$

$$s = t_2^\top S' \in \mathbb{R}^{r_3} \quad (cost \approx 2 r_2 r_3), \quad w = s^\top t_3 \in \mathbb{R} \quad (cost \approx r_3).$$

*Thus, vector multiplication with Tucker factors reduce the computational cost from $O(p_1 p_2 p_3)$ to $O(\sum_{k=1}^3 p_k r_k + r_1 r_2 r_3)$. The storage requirements of Tucker decomposition also drop significantly. Storing the full tensor $Y$ requires $O(p_1 p_2 p_3)$ memory, while the Tucker representation requires only $\sum_{k=1}^3 p_k r_k + r_1 r_2 r_3$ scalar storage.*

*Therefore, both computation and storage of Tucker decomposition scale with $O(\sum_{k=1}^3 p_k r_k + r_1 r_2 r_3)$, which is substantially smaller than the dense case when $r_k \ll p_k$.*

**Adaptivity in Theorem 1.** For any user specified target Tucker rank $(r_1, r_2, r_3)$, the estimator $\widetilde{X}$ satisfies (4). In other words, the result holds uniformly over all target Tucker ranks.

**Bias–variance tradeoff.** The error bound (4) exhibits a clear bias–variance decomposition for Tucker rank-$(r_1, r_2, r_3)$ estimators. The bias term $\xi_{(r_1, r_2, r_3)}$ is the best achievable approximation error of $X^*$ at the chosen ranks and decreases as the $r_k$ increase. Therefore, as the ranks grow, variance increases while bias shrinks. Balancing these two leads to improved accuracy.

**Proof techniques.** For the bias, we leverage classical linear algebra tools, such as Mirsky's and Ky Fan's theorems, to characterize the optimal approximation error appearing in (4). To handle the variance of the estimator $\widetilde{X}$, we adopt existing techniques from Zhang & Xia (2018).

**Theorem 2.** *Let $r_1, r_2, r_3, p_1, p_2, p_3$ be arbitrary integers such that $r_i \leq \min\{p_i, \prod_{j \neq i} p_j\}$ for all $i \in \{1, 2, 3\}$. Let $\mathcal{T}_{(r_1, r_2, r_3)}$ be the class of Tucker low-rank tensor as in (2) and for $\xi > 0$, let*

$$\mathcal{T}_{(r_1, r_2, r_3)}^\xi = \left\{ B \in \mathbb{R}^{p_1 \times p_2 \times p_3} : \inf_{A \in \mathcal{T}_{(r_1, r_2, r_3)}} \|B - A\|_{\mathrm{F}} \leq \xi \right\}.$$

*Consider the model $Y = X^* + Z \in \mathbb{R}^{p_1 \times p_2 \times p_3}$ where the entries of $Z$ are i.i.d. $\mathrm{subGaussian}(0, \kappa^2)$. For any ranks $(r_1, r_2, r_3)$ and any $0 \leq \xi \geq \kappa^2 p_1 p_2 p_3$, there exist a universal constant $c > 0$ such that*

$$\inf_{\widehat{X}} \sup_{X^* \in \mathcal{T}_{(r_1, r_2, r_3)}^\xi} \mathbb{E} \big\| \widehat{X} - X^* \big\|_{\mathrm{F}}^2 \geq c \left\{ \kappa^2 \left( \textstyle\sum_{k=1}^3 p_k r_k + r_1 r_2 r_3 \right) + \xi^2 \right\}. \tag{5}$$

**Tightness.** Our upper bound in (4) attains the minimax-optimal rate (up to constants) over the class of tensors with Tucker rank at most $(r_1, r_2, r_3)$. In particular, the lower bound in Theorem 2 matches this rate over the same model class, showing that no estimator can uniformly achieve a strictly smaller error. Thus, our bound is sharp and the proposed procedure is minimax optimal.

## 3 BIAS–VARIANCE TRADEOFF IN MATRIX ESTIMATION

We consider the model

$$Y = X^* + Z \in \mathbb{R}^{m \times n}, \tag{6}$$

where $X^*$ is an unknown matrix with arbitrary rank, and $Z$ is a perturbation (noise) matrix. Even when $\operatorname{rank}(X^*)$ is not small, it is common to approximate $X^*$ by a rank-$r$ estimator, because (i) low-rank decomposition can dramatically increase the computational and memory efficiency, see Remark 3 for more details; and (ii) the approximation bias can be small, so balancing the bias and variance may lead to smaller estimation error. Let

$$X^* = \sum_{i=1}^{\min\{m,n\}} \sigma_i(X^*)\, u_i^* v_i^{*\top}, \qquad X_{(r)}^* = \sum_{i=1}^{r} \sigma_i(X^*)\, u_i^* v_i^{*\top}.$$

By the Eckart–Young–Mirsky theorem (Eckart & Young, 1936), $X_{(r)}^*$ is the best rank-$r$ approximation to $X^*$ in Frobenius norm in the sense that

$$\|X^* - X_{(r)}^*\|_{\mathrm{F}} \le \|X^* - W\|_{\mathrm{F}} \text{ for any } W \in \mathbb{R}^{m \times n} \text{ with } \operatorname{rank}(W) \le r.$$

Suppose we observe $Y$ instead of $X^*$, and write the SVD of $Y$ as

$$Y = \sum_{i=1}^{\min\{m,n\}} \sigma_i(Y)\, u_i v_i^{\top}, \qquad Y_{(r)} = \sum_{i=1}^{r} \sigma_i(Y)\, u_i v_i^{\top}. \tag{7}$$

That is, $Y_{(r)}$ is the rank-$r$ truncation of the SVD of $Y$. We now provide an upper bound between $Y_{(r)}$ and the unknown matrix $X^*$.

**Theorem 3.** *Under* (6), *let $Y_{(r)}$ be defined as in* (7). *Then*

$$\|Y_{(r)} - X^*\|_{\mathrm{F}} \ \le \ C(\sqrt{r}\|Z\| + \xi_{(r)}), \tag{8}$$

*where $\xi_{(r)} = \|X_{(r)}^* - X^*\|_{\mathrm{F}} = \sqrt{\sum_{i=r+1}^{\min\{m,n\}} \sigma_i^2(X^*)}$.*

**Adaptivity in Theorem 3.** For any prescribed rank $r$, the rank-$r$ truncated SVD estimator $Y_{(r)}$ satisfies the guarantee in Theorem 3. In other words, the theorem holds uniformly over all $r \ge 1$, so it applies to the vanilla SVD at any user-specified target rank.

**Constant in** (8). Tracing the proof of Theorem 3 shows that one may take the explicit constant $C = 2 + \sqrt{2} \approx 3.414$ in (8). We did not attempt to optimize this further.

**Bias–variance tradeoff.** Theorem 3 exhibits a clear bias–variance tradeoff for rank-$r$ estimation. The bias term $\xi_{(r)} = \|X^* - X_{(r)}^*\|_{\mathrm{F}}$ is the best possible rank-$r$ approximation error of $X^*$ and decreases as $r$ increases. The standard deviation term, being order $(\sqrt{r}\|Z\|)$, captures the cost of estimating additional singular components under noise and increases with $r$.

**Remark 3.** *Suppose $Y \in \mathbb{R}^{m \times n}$ and let $Y_{(r)} = U_{(r)} \Sigma_{(r)} V_{(r)}^{\top}$ be a rank-$r$ factorization (e.g. the truncated SVD), where $U_{(r)} \in \mathbb{R}^{m \times r}$, $V_{(r)} \in \mathbb{R}^{n \times r}$ satisfy $U_{(r)}^{\top} U_{(r)} = I_r$, $V_{(r)}^{\top} V_{(r)} = I_r$, and $\Sigma_{(r)} = \operatorname{diag}(\sigma_1, \ldots, \sigma_r) \in \mathbb{R}^{r \times r}$. For a dense multiplication with $v \in \mathbb{R}^n$, computing $Yv$ costs $O(mn)$ operations. In contrast, using the factorization we compute successively*

$$t = V_{(r)}^{\top} v \in \mathbb{R}^r \ (cost \ \approx 2nr), \quad s = \Sigma_{(r)} t \in \mathbb{R}^r \ (cost \ \approx r), \quad w = U_{(r)} s \in \mathbb{R}^m \quad (cost \ \approx 2mr).$$

*Hence $Y_{(r)} v = U_{(r)} \Sigma_{(r)} V_{(r)}^{\top} v$ can be computed in $O(mr + nr)$ operations.*

*Beyond computational savings, the SVD factorization also reduces storage costs. Storing the full matrix $Y$ requires $O(mn)$ memory, whereas storing the factors $U_{(r)} \in \mathbb{R}^{m \times r}$, $\Sigma_{(r)} \in \mathbb{R}^{r \times r}$, and $V_{(r)} \in \mathbb{R}^{n \times r}$ requires only $O(mr + nr)$ scalars.*

*Therefore, both computation and storage using the truncated SVD are significantly smaller than in the full matrix case, especially when $r \ll \min\{m, n\}$.*

In the following, we illustrate the application of Theorem 3 in three matrix estimation settings.

**Corollary 4.** *Suppose* (6) *holds and that the entries of $Z$ are independent sub-Gaussian random variables with*

$$\|Z_{ij}\|_{\psi_2} \leq \kappa.$$

*Then with probability at least $1 - \exp(-(m+n))$, there exists a universal constant $C > 0$ such that*

$$\|Y_{(r)} - X^*\|_{\mathrm{F}} \leq C\big(\|X^*_{(r)} - X^*\|_{\mathrm{F}} + \kappa\sqrt{r(m+n)}\big).$$

*Proof of Corollary 4.* From Vershynin (2018) section 4.4.2, it follows that $\|Z\| \leq C_1\kappa\sqrt{m+n}$ with probability at least $1 - \exp(-(m+n))$ . The desired result follows immediately from Theorem 3. □

**Corollary 5.** *Suppose $Z \in \mathbb{R}^{m \times n}$ is a sub-Gaussian random matrix in the sense that for any $v \in \mathbb{R}^m$ with $\|v\|_2 = 1$, and $w \in \mathbb{R}^n$ with $\|w\|_2 = 1$, it holds that*

$$\|v^\top Z w\|_{\psi_2} \leq \kappa.$$

*Then with probability at least $1 - \exp(-(m+n))$, there exists a universal constant $C > 0$ such that*

$$\|Y_{(r)} - X^*\|_{\mathrm{F}} \leq C\big(\|X^*_{(r)} - X^*\|_{\mathrm{F}} + \kappa\sqrt{r(m+n)}\big).$$

*Proof of Corollary 5.* By Lemma 15, $\|Z\| \leq C_1\sqrt{m+n}$ with probability at least $1 - \exp(-(m+n))$. The desired result follows immediately from Theorem 3. □

**Corollary 6.** *Let $Z_1, \ldots, Z_N \in \mathbb{R}^n$ be i.i.d. mean-zero sub-Gaussian random vectors with covariance $\Sigma$. Define the sample covariance matrix*

$$Y = \frac{1}{N}\sum_{k=1}^{N} Z_k Z_k^\top,$$

*and let $X^* = \mathbb{E}[Y]$ be the population covariance matrix. Suppose $Z_k$ is sub-Gaussian in the sense that*

$$\|u^\top Z_k\|_{\psi_2} \leq \kappa \quad \text{for all } u \in \mathbb{R}^n, \ \|u\|_2 = 1.$$

*Then with probability at least $1 - 2\exp(-n)$, there exists a universal constant $C > 0$ such that*

$$\|Y_{(r)} - X^*\|_{\mathrm{F}} \leq C\Big(\|X^*_{(r)} - X^*\|_{\mathrm{F}} + \kappa^2\sqrt{r}\Big[\sqrt{\frac{n}{N}} + \frac{n}{N}\Big]\Big).$$

*Proof of Corollary 6.* Let $\widetilde{Z} := Y - X^*$. By Remark 4.7.3 of Vershynin (2018), with probability at least $1 - \exp(-cn)$ one has $\|\widetilde{Z}\| \leq C_1\kappa^2\Big[\sqrt{\frac{n}{N}} + \frac{n}{N}\Big]$. Applying Theorem 3 with $Z$ replaced by $\widetilde{Z}$ yields the desired bound. □

**Theorem 7.** *Let $r, m, n$ be arbitrary integers such that $r \leq \min\{m, n\}$. For $\xi > 0$, let $\mathcal{M}_r^\xi$ be such that*

$$\mathcal{M}_r^\xi = \big\{B \in \mathbb{R}^{m \times n} : \text{ there exits } A \text{ such that } \mathrm{rank}(A) \leq r \text{ and } \|B - A\|_{\mathrm{F}} \leq \xi\big\}.$$

*Consider the model $Y = X^* + Z \in \mathbb{R}^{m \times n}$ where the entries of $Z$ are i.i.d. $\mathrm{subGaussian}(0, \kappa^2)$. Suppose in addition that $\xi^2 \leq (m-r)n\kappa^2$. Then there exist a universal constant $c > 0$ such that*

$$\inf_{\widehat{X}} \sup_{X^* \in \mathcal{M}_r^\xi} \mathbb{E}\big\|\widehat{X} - X^*\big\|_{\mathrm{F}}^2 \geq c\Big\{\kappa^2(m+n)r + \xi^2\Big\}.$$

The upper bound in (8) attains the minimax-optimal rate (up to constants) over the class of matrices with rank at most $r$. The lower bound in Theorem 7 matches this rate on the same model class.

# 4 NUMERICAL EXPERIMENTS

In many applied settings, selecting the single *best* rank $r$ is inherently difficult in finite samples with unknown latent structure. Our analysis instead provides guarantees for singular-value decompositions in both the matrix (SVD) and tensor (HOSVD) regimes across a broad range of $r$. Consequently, rather than hinging on a brittle rank-selection heuristic, we empirically demonstrate that both truncated SVD and one-step HOSVD (Algorithm 1) are robust whenever ranks are chosen neither too small (underfitting/bias) nor too large (overfitting/variance). This is consistent with the bias–variance tradeoff formalized in Theorems 1 and 3. Moreover, in practice the choice of $r$, is frequently driven by diverse aims other than merely minimizing mean-squared error, including visualization, interpretability, or downstream decision tasks. This is reflected in widespread, long-standing use across biology/genomics (Price et al., 2006), economics (Filmer & Pritchett, 2001), and computer vision (Turk & Pentland, 1991).

We evaluate these predictions on (i) a real 3D brain-MRI dataset with controlled additive noise (IXI; T1-weighted volumes (IXI, 2002)), and (ii) simulated data for both matrices and third–order tensors.

## 4.1 REAL DATA: 3D BRAIN MRI WITH CONTROLLED NOISE

We evaluate on five T1-weighted brain MRI volume drawn from the IXI dataset (IXI, 2002). The selected volumes are third-order tensors of shape $256 \times 256 \times 150$. To probe low-rank behavior in a realistic preprocessing pipeline, all volume are prepared in two wavelet-smoothed variants using standard orthonormal families—(a) Daubechies-6 (`db6`) and (b) Symlet-8 (`sym8`) (Daubechies, 1988; 1992).

We implement separable 3D multilevel decompositions (`wavedecn`) with periodic boundary handling, and apply subband-wise soft-thresholding via BayesShrink (Chang et al., 2000; Donoho & Johnstone, 1994). The literature supports wavelet shrinkage as an effective denoising/smoothing step for MR magnitude images (Nowak, 1999; Wood & Johnson, 1999).

**Noise model and ranks.** For our prepared volumes $X^*$ and each noise level $\lambda \in \{0.2, 0.3, 0.4, 0.5, 0.6, 0.7\}$ we generate

$$Y = X^* + Z, \qquad Z_{ijk} \overset{\text{i.i.d.}}{\sim} \mathcal{N}(0, \kappa^2), \quad \kappa = \lambda \frac{\|X^*\|_{\text{F}}}{\sqrt{p_1 p_2 p_3}},$$

which keeps the per-entry SNR comparable across images and variants. We compute one-step HOSVD reconstructions (Algorithm 1) $\widetilde{X}$ at Tucker rank $(r, r, r)$ for $r \in \{35, 50, 60, 65, 70, 80, 90, 95\}$.

**Findings.** Figure 1 shows representative mid-sagittal slices from the reconstructions of the wavelet-smoothed variants of a selected volume. As intermediate ranks, the images sharpen and cortical detail appears while background fluctuations remain suppressed. We provide selected quantitative results in Section 4.1, which show that all variants exhibit this expected progression.

## 4.2 SYNTHETIC DATA: TENSORS

We generate third-order tensors $X^* \in \mathbb{R}^{p \times p \times p}$ in the Tucker form

$$X^* = \mathcal{G} \times_1 U_1 \times_2 U_2 \times_3 U_3, \qquad U_1, U_2, U_3 \in \mathbb{R}^{p \times p}, \ U_1^\top U_1 = U_2^\top U_2 = U_3^\top U_3 = I_p,$$

where $U_1, U_2, U_3$ are drawn independently with orthonormal columns. The core tensor $\mathcal{G} \in \mathbb{R}^{p \times p \times p}$ is defined entrywise by

$$\mathcal{G}_{ijk} = \gamma_{ijk} \beta^{i+j+k}, \qquad \beta = 0.8,$$

where the signs $\gamma_{ijk}$ are independent with $\mathbb{P}(\gamma_{ijk} = -1) = 1/3, \mathbb{P}(\gamma_{ijk} = 1) = 2/3$. We observe $Y = X^* + Z$ with i.i.d. Gaussian noise $Z_{ijk} \sim \mathcal{N}(0, 1)$ entry-wise. The SNR is controlled by scaling the signal such that

$$\|X^*\|_{\text{F}} = \lambda \sqrt{p^3}, \qquad \lambda \in \{10, 50\}.$$

The estimator $\widetilde{X}$ is obtained via the one-step HOSVD (Algorithm 1) at the Tucker rank $(r, r, r)$.

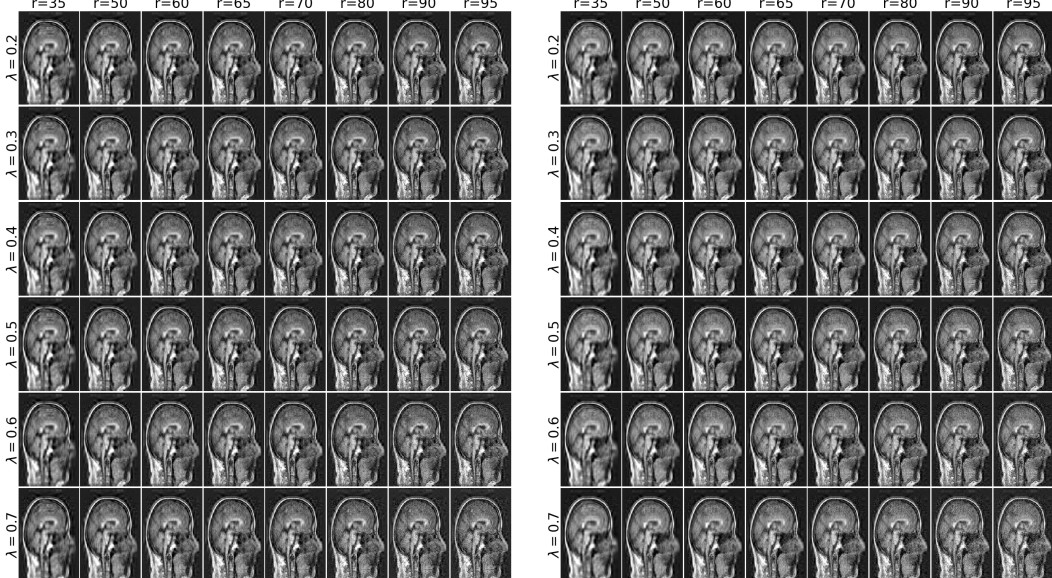

Figure 1: Mid-sagittal MRI slice of the reconstructed volume at selected ranks and noise levels $\lambda$ for the `db6` variant (left), and for the `sym8` variant (right).

| $\lambda$ | $r$ | original | db6 | sym8 |
|---|---|---|---|---|
| 0.4 | 50 | 0.1753 (0.0330) | 0.1747 (0.0537) | 0.1681 (0.0466) |
| | 65 | 0.1553 (0.0268) | 0.1561 (0.0435) | 0.1501 (0.0373) |
| | 80 | 0.1490 (0.0183) | 0.1499 (0.0291) | 0.1451 (0.0245) |
| | 95 | 0.1550 (0.0102) | 0.1558 (0.0159) | 0.1525 (0.0128) |
| 0.5 | 50 | 0.1797 (0.0322) | 0.1793 (0.0521) | 0.1728 (0.0451) |
| | 65 | 0.1655 (0.0248) | 0.1668 (0.0400) | 0.1610 (0.0341) |
| | 80 | 0.1677 (0.0154) | 0.1691 (0.0244) | 0.1647 (0.0201) |
| | 95 | 0.1840 (0.0073) | 0.1847 (0.0116) | 0.1819 (0.0089) |
| 0.6 | 50 | 0.1851 (0.0311) | 0.1850 (0.0503) | 0.1787 (0.0434) |
| | 65 | 0.1776 (0.0224) | 0.1793 (0.0362) | 0.1738 (0.0307) |
| | 80 | 0.1890 (0.0124) | 0.1904 (0.0200) | 0.1866 (0.0162) |
| | 95 | 0.2150 (0.0049) | 0.2154 (0.0085) | 0.2132 (0.0062) |

Table 1: Mean relative reconstruction errors (entries are Mean (SE) across volumes).

**Dimensions and evaluated ranks.** We vary $p \in \{20, 50, 75, 100\}$ and for each $p$, we evaluate two ranks $r$ as outlined in Table 2.

Each configuration $(p, \lambda, r)$ is repeated $R = 50$ times with freshly drawn $(U_1, U_2, U_3)$ and noise. Implementations use exact batched SVD on GPU via TensorLy (PyTorch backend) (Kossaifi et al., 2019). We report the sample mean and standard error of the relative Frobenius error

$$\mathrm{RelErr}(\widetilde{X}; X^*) = \frac{\|\widetilde{X} - X^*\|_{\mathrm{F}}}{\|X^*\|_{\mathrm{F}}},$$

summarized on the right panel of Table 2. We observe that the error consistently decreases as the SNR parameter $\lambda$ increases. Overall, one-step HOSVD is robust across the tested sizes and ranks, yielding accurate estimates on the synthetic tensors.

### 4.3 SYNTHETIC DATA: MATRICES

We examine the matrix analogue under a controlled SNR design that mirrors the tensor experiments. For each configuration, we draw an $m \times n$ latent signal

$$X^* = U \Sigma V^\top, \qquad U \in \mathbb{R}^{m \times n}, \ V \in \mathbb{R}^{n \times n}, \ U^\top U = V^\top V = I_n,$$

where $\Sigma = \text{diag}(\sigma_1, \ldots, \sigma_n)$ has exponentially decaying singular values, i.e. $\sigma_i = \beta^i$ with $\beta = 0.8$. We choose $m$ and $n$ as outlined in Table 2. We observe $Y = X^* + Z$, where $Z_{ij} \sim \mathcal{N}(0,1)$ is i.i.d. Gaussian noise entry-wise. The SNR is controlled by scaling the signal such that

$$\|X^*\|_{\mathrm{F}} = \lambda \sqrt{mn}, \qquad \lambda \in \{10, 50\}.$$

This design varies SNR via $\lambda$ while keeping the per–entry noise scale identical across sizes. Each configuration $(m, n, \lambda, r)$ is repeated $R = 50$ times with independent draws of $(U, V)$ and noise, and we report the mean and standard error of RelErr. We use the truncated SVD estimator $Y_{(r)}$ as in Theorem 3. Results are shown in the left panel of Table 2. The truncated SVD is robust over a wide range of selected ranks.

Table 2: Results for matrix (left) and tensor (right).

| Matrix | | | | | Tensor | | | |
|---|---|---|---|---|---|---|---|---|
| $\lambda$ | $m$ | $n$ | $r$ | Mean (SE) | $\lambda$ | $p$ | $r$ | Mean (SE) |
| 10 | 100 | 15 | 10 | 0.1178 (0.00125) | 10 | 20 | 10 | 0.1420 (0.00350) |
| | | | 12 | 0.0906 (0.00164) | | | 12 | 0.0977 (0.00208) |
| | 250 | 25 | 10 | 0.1254 (0.00044) | | 50 | 10 | 0.1370 (0.00347) |
| | | | 15 | 0.0868 (0.00077) | | | 15 | 0.0459 (0.00102) |
| | 375 | 20 | 10 | 0.1279 (0.00045) | | 75 | 10 | 0.1367 (0.00348) |
| | | | 15 | 0.0927 (0.00076) | | | 15 | 0.0427 (0.00113) |
| | 500 | 80 | 30 | 0.0698 (0.00032) | | 100 | 30 | 0.0203 (0.00008) |
| | | | 40 | 0.0795 (0.00035) | | | 40 | 0.0294 (0.00009) |
| 50 | 100 | 15 | 10 | 0.0844 (0.00007) | 50 | 20 | 10 | 0.1361 (0.00381) |
| | | | 12 | 0.0181 (0.00037) | | | 12 | 0.0840 (0.00207) |
| | 250 | 25 | 10 | 0.1079 (0.00002) | | 50 | 10 | 0.1362 (0.00359) |
| | | | 15 | 0.0378 (0.00009) | | | 15 | 0.0410 (0.00109) |
| | 375 | 20 | 10 | 0.1068 (0.00002) | | 75 | 10 | 0.1368 (0.00374) |
| | | | 15 | 0.0349 (0.00008) | | | 15 | 0.0412 (0.00098) |
| | 500 | 80 | 30 | 0.0134 (0.00008) | | 100 | 30 | 0.0039 (0.00002) |
| | | | 40 | 0.0156 (0.00006) | | | 40 | 0.0058 (0.00002) |

## 5 DISCUSSION

This work provides a rank–adaptive analysis of HOSVD-based tensor denoising without assuming exact low-rank. Our main theorem (Theorem 1) yields an explicit bias–variance decomposition

$$\|\widetilde{X} - X^*\|_{\mathrm{F}} \lesssim \kappa \sqrt{\textstyle\sum_{k=1}^3 p_k r_k + r_1 r_2 r_3} + \xi_{(r_1, r_2, r_3)},$$

uniformly over all user–specified target Tucker ranks. The variance term scales with the effective degrees of freedom of the Tucker model, while the bias term is the best achievable approximation error at those ranks. Together with the matrix counterpart (Theorem 3), the results unify classical SVD intuition with the multilinear (tensor) setting and rigorously justify a practice common in applications: choose ranks large enough to suppress bias but not so large as to amplify noise. Our experiments on IXI brain MRI and controlled synthetic data corroborate this picture.

**Future work.** *(i) Beyond i.i.d. noise and full observations.* Extending the theory to heteroskedastic or correlated perturbations (e.g. spatially correlated fields, Rician-like MRI noise) and to incomplete observations (tensor completion, masked entries) is natural. We expect the variance term to inherit problem-dependent effective dimensions (e.g. leverage scores or sampling densities), while the bias term remains $\xi_{(r_1, r_2, r_3)}$. A key technical step is replacing isotropic concentration with mode-wise covariance-aware bounds for $\mathcal{M}_k(Z)$ and their projections. *(ii) Higher orders and higher ranks.* The proof strategy appears to scale to $d$th-order tensors with the clean generalization

$$\|\widetilde{X} - X^*\|_{\mathrm{F}} \lesssim \kappa \sqrt{\textstyle\sum_{k=1}^d p_k r_k + \prod_{k=1}^d r_k} + \xi_{(r_1, \ldots, r_d)},$$

under appropriately formulated spectral gaps for each unfolding. This would offer a unified bias–variance law across orders and ranks, and would help explain the strong empirical performance of HOSVD-style estimators in truly high-order settings.

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

# Supplementary Materials for "Bias–variance Tradeoff in Tensor Estimation"

Without loss of generality, if $W \sim \mathrm{subGaussian}(0, \kappa^2)$, we assume throughout, unless otherwise specified in the appendix, that $\mathbb{E}[W^2] = \kappa^2$.

## A    ADDITIONAL NUMERICAL RESULTS

**Spectral gap assumption.**    Theorem 1 requires a mode-wise spectral gap condition on the singular values of $\mathcal{M}_k(X^*)$ as in (3). We empirically examine this assumption on the same 3D brain MRI dataset as in Section 4.1. Specifically, we randomly draw 50 T1-weighted brain volumes from the IXI dataset (IXI, 2002). For each volume and each mode $k \in \{1, 2, 3\}$, we compute the singular values of $\mathcal{M}_k(X^*)$ and record the gaps

$$\sigma_{r_k}\left(\mathcal{M}_k(X^*)\right) - \sigma_{r_k+1}(\mathcal{M}_k(X^*)), \qquad r_k = 1, \ldots, 145.$$

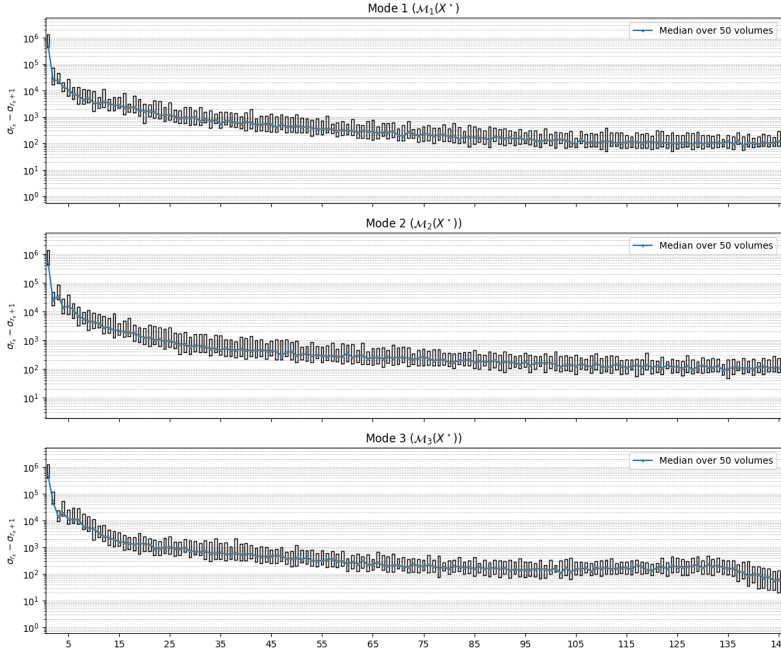

Figure 2:  Empirical median and interquartile range (Q1–Q3) of the mode-wise singular-value gaps (log scale) over 50 T1-weighted IXI brain MRI volumes.

Figure 2 summarizes the distribution of these gaps across subjects. The empirical medians remain strictly positive over all ranks and modes, and the inter-quartile bands indicate that this behavior is stable across the 50 volumes. This provides concrete evidence that the spectral gap condition in (3) is not merely a idealized technical assumption of the analysis, but is indeed observed in practice.

**Bias–variance curve.**    To empirically quantify the contributions of bias and variance to the error $\|Y - X^*\|_{\mathrm{F}}$, we consider the synthetic tensor model of Section 4.2 with $p = 100$. For each candidate Tucker rank $r \in \{12, 15, 18, \ldots, 84, 87\}$, we run 50 independent repetitions. The bias component is obtained as the error $\|\widetilde{X} - X^*\|_{\mathrm{F}}$, where $\widetilde{X}$ denotes the output of Algorithm 1 when applied directly to the noiseless tensor $X^*$. The variance component is obtained as

$$\max\left\{\sqrt{r}\left\|\mathcal{M}_1(X^*) \cdot \left\{U_2^{(1)} \otimes U_3^{(1)}\right\}\right\|, \sqrt{r}\left\|\mathcal{M}_2(X^*) \cdot \left\{U_1^{(1)} \otimes U_3^{(1)}\right\}\right\|, \sqrt{r}\left\|\mathcal{M}_3(X^*) \cdot \left\{U_1^{(1)} \otimes U_2^{(1)}\right\}\right\|, \left\|X^* \times_1 U_1^{(1)} \times_2 U_2^{(1)} \times_3 U_3^{(1)}\right\|_{\mathrm{F}}\right\}.$$

The Figure 3 reports the resulting mean bias and variance curves as functions of the target rank.

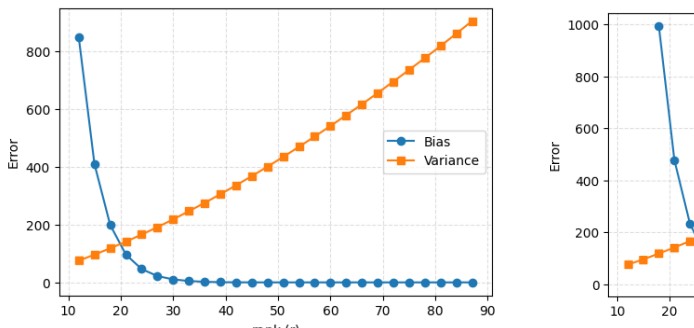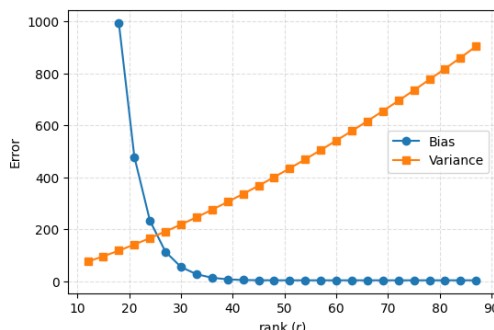

Figure 3: Mean bias and variance components of the error as functions of the target rank in the synthetic experiment with $p = 100$. The left panel reporesent the curve with scaling paramter $\lambda = 10$ and the right panel $\lambda = 50$.

Figure 3 exhibits the expected bias–variance trade-off: as the target rank increases, the bias term decreases while the variance term grows, leading to an intermediate rank that minimizes the overall error.

**Practical choice of target ranks.** There are several well-known ways to select the target Tucker ranks. To select the rank parameters in a data-driven way, we follow the discussion in (Ozdemir et al., 2017) and adopt an adaptive thresholding scheme. More precisely, we choose $r_k$ so that $\sigma_{r_k}^2(\mathcal{M}_k(Y)) \geq \tau$, where $\tau$ is a threshold. A simple choice is $\tau = \alpha\|Y\|_F^2$ with $\alpha = 0.01$; we retain all singular values of $\mathcal{M}_k(Y)$ up to the largest index $j$ satisfying $\sigma_j^2(\mathcal{M}_k(Y)) \geq \tau$ and discard the rest.

# B  LOWER BOUNDS

In this section we prove the complementary minimax lower bounds Theorem 7 and Theorem 2 that match the estimation rates in Theorem 3 and Theorem 1. These results show that our upper bounds are sharp (up to universal constants), and therefore establish the minimax optimality of our estimation procedure.

## B.1  TENSOR LOWER BOUND

*Proof of Theorem 2.*
**Step 1.** We follow a similar strategy as in the proof of Theorem 7 to show

$$\inf_{\widehat{X}} \sup_{X^* \in \mathcal{T}_{(r_1,r_2,r_3)}^{\xi}} \mathbb{E}\big\|\widehat{X} - X^*\big\|_F^2 \geq c(\kappa^2 r_1 r_2 r_3 + \xi^2). \tag{9}$$

Let
$$r_1' = \lfloor \xi^2/(r_2 r_3 \kappa^2) \rfloor.$$
Consider the set of tensors
$$\mathcal{B} = \{B \in \mathbb{R}^{p_1 \times p_2 \times p_3} : \|B\|_\infty \leq \kappa^2, B_{i,j,k} = 0 \text{ if } i > r_1 + r_1' \text{ or } j > r_2 \text{ or } k > r_3\}.$$

In this step, we show that

$$\mathcal{B} \subset \mathcal{T}_{(r_1,r_2,r_3)}^{\xi}. \tag{10}$$

To see this, let $B \in \mathcal{B}$, and denote $B'$ to be the matrix such that

$$B'_{i,j,k} = \begin{cases} B_{i,j,k} & \text{if } i \leq r_1 \text{ and } j \leq r_2 \text{ and } k \leq r_3; \\ 0 & \text{otherwise.} \end{cases}$$

Then $B' \in \mathcal{T}_{(r_1,r_2,r_3)}$ and

$$\|B - B'\|_F^2 \leq r_1' r_2 r_3 \|B\|_\infty = r_1' r_2 r_3 \kappa^2 \leq \xi^2.$$

Therefore $\mathcal{B} \subset \mathcal{T}^{\xi}_{(r_1,r_2,r_3)}$. So

$$\inf_{\widehat{X}} \sup_{X^* \in \mathcal{T}^{\xi}_{(r_1,r_2,r_3)}} \mathbb{E}\big\|\widehat{X} - X^*\big\|^2_{\mathrm{F}} \geq \inf_{\widehat{X}} \sup_{X^* \in \mathcal{B}} \mathbb{E}\big\|\widehat{X} - X^*\big\|^2_{\mathrm{F}}. \quad (11)$$

**Step 1.1.** Note that the tensors in $\mathcal{B}$ are having at most $(r_1 + r'_1)r_2 r_3$ nonzero entries, the set of matrices $\mathcal{B}$ can be viewed as vectors in $\mathbb{R}^d$, with $d = (r_1 + r'_1)r_2 r_3$. Denote

$$\mathcal{V} = \{V \in \mathbb{R}^d, \|V\|_\infty \leq \kappa\}.$$

Then

$$\inf_{\widehat{X}} \sup_{X^* \in \mathcal{B}} \mathbb{E}\big\|\widehat{X} - X^*\big\|^2_{\mathrm{F}} = \inf_{\widehat{V}} \sup_{V \in \mathcal{V}} \mathbb{E}\|\widehat{V} - V\|^2.$$

Since by Lemma 8, $\inf_{\widehat{V}} \sup_{V \in \mathcal{V}} \mathbb{E}\|\widehat{V} - V\|^2 \geq c_2 \kappa^2 d$, it follows that

$$\inf_{\widehat{X}} \sup_{X^* \in \mathcal{B}} \mathbb{E}\big\|\widehat{X} - X^*\big\|^2_{\mathrm{F}} = \inf_{\widehat{V}} \sup_{V \in \mathcal{V}} \mathbb{E}\|\widehat{V} - V\|^2 \geq c_2 \kappa^2 d = c_2 \kappa^2 (r_1 + r'_1)r_2 r_3.$$

**Step 1.2.** To show (9), there are two cases. **(a)** Suppose $\xi^2 \geq r_2 r_3 \kappa^2$, then $r'_1 = \lfloor \xi^2/(r_2 r_3 \kappa^2)\rfloor \geq 1$ and therefore

$$\inf_{\widehat{X}} \sup_{X^* \in \mathcal{T}^{\xi}_{(r_1,r_2,r_3)}} \mathbb{E}\big\|\widehat{X} - X^*\big\|^2_{\mathrm{F}} \geq c_2 \kappa^2 (r_1 + r'_1)r_2 r_3 \geq c_2 r_1 r_2 r_3 + c_3 \xi^2.$$

**(b)** Suppose $\xi^2 < r_2 r_3 \kappa^2$. Then $r'_1 = 0$

$$\inf_{\widehat{X}} \sup_{X^* \in \mathcal{T}^{\xi}_{(r_1,r_2,r_3)}} \mathbb{E}\big\|\widehat{X} - X^*\big\|^2_{\mathrm{F}} \geq c_2 \kappa^2 r_1 r_2 r_3 \geq \frac{c_2}{2}(\kappa^2 r_1 r_2 r_3 + \xi^2),$$

where the first inequality follows from **Step 1.1**, the second inequality follows from $\xi^2 < r_2 r_3 \kappa^2$.
**Step 2.** Note that

$$\inf_{\widehat{X}} \sup_{X^* \in \mathcal{T}^{\xi}_{(r_1,r_2,r_3)}} \mathbb{E}\big\|\widehat{X} - X^*\big\|^2_{\mathrm{F}} \geq \inf_{\widehat{X}} \sup_{X^* \in \mathcal{T}_{(r_1,r_2,r_3)}} \mathbb{E}\big\|\widehat{X} - X^*\big\|^2_{\mathrm{F}} \geq c_3 \kappa^2 \sum_{k=1}^{3} p_k r_k, \quad (12)$$

where the second inequality follows from Zhang & Xia (2018). The desired result directly follows from (9) and (12). $\qquad\square$

## B.2 MATRIX LOWER BOUND

*Proof.* Without loss of generality, suppose that $m \leq n$ and $r \leq m/2$. Therefore it suffices to show that

$$\inf_{\widehat{X}} \sup_{X^* \in \mathcal{M}^{\xi}_r} \mathbb{E}\big\|\widehat{X} - X^*\big\|^2_{\mathrm{F}} \geq c_1 \Big\{\kappa^2 n r + \xi^2\Big\}. \quad (13)$$

Let $r' = \lfloor \xi^2/(n\kappa^2)\rfloor$. Note that since $\xi^2 \leq (m - r)n\kappa^2$, it follows that

$$r' + r \leq m.$$

**Step 1.** Consider the set of matrices

$$\mathcal{B} = \{B \in \mathbb{R}^{m \times n} : \|B\|_\infty \leq \kappa^2, B_{i,j} = 0 \text{ if } i > r + r'\}.$$

In this step, we show that

$$\mathcal{B} \subset \mathcal{M}^{\xi}_r. \quad (14)$$

To see this, let $B \in \mathcal{B}$, and denote $B'$ to be the matrix such that the first $r$ rows of $B'$ equals the first $r$ rows of $B$, and the rest of $(m - r)$ rows of $B'$ are all $0$. Then since $B'$ has at most $r$ rows that are nonzero, $\mathrm{rank}(B') \leq r$ and

$$\|B - B'\|^2_{\mathrm{F}} \leq r'n\|B\|_\infty = r'n\kappa^2 \leq \xi^2.$$

Therefore $\mathcal{B} \subset \mathcal{M}_r^\xi$. So

$$\inf_{\widehat{X}} \sup_{X^* \in \mathcal{M}_r^\xi} \mathbb{E}\big\|\widehat{X} - X^*\big\|_{\mathrm{F}}^2 \geq \inf_{\widehat{X}} \sup_{X^* \in \mathcal{B}} \mathbb{E}\big\|\widehat{X} - X^*\big\|_{\mathrm{F}}^2. \tag{15}$$

**Step 2.** Note that matrices in $\mathcal{B}$ are having at most $(r + r')n$ nonzero entries, the set of matrices $\mathcal{B}$ can be viewed as vectors in $\mathbb{R}^d$, with $d = (r + r')n$. Denote

$$\mathcal{V} = \{V \in \mathbb{R}^d, \|V\|_\infty \leq \kappa\}.$$

Then

$$\inf_{\widehat{X}} \sup_{X^* \in \mathcal{B}} \mathbb{E}\big\|\widehat{X} - X^*\big\|_{\mathrm{F}}^2 = \inf_{\widehat{V}} \sup_{V \in \mathcal{V}} \mathbb{E}\|\widehat{V} - V\|^2.$$

Since by Lemma 8, $\inf_{\widehat{V}} \sup_{V \in \mathcal{V}} \mathbb{E}\|\widehat{V} - V\|^2 \geq c_2 \kappa^2 d$, it follows that

$$\inf_{\widehat{X}} \sup_{X^* \in \mathcal{M}_r^\xi} \mathbb{E}\big\|\widehat{X} - X^*\big\|_{\mathrm{F}}^2 \geq \inf_{\widehat{X}} \sup_{X^* \in \mathcal{B}} \mathbb{E}\big\|\widehat{X} - X^*\big\|_{\mathrm{F}}^2 = \inf_{\widehat{V}} \sup_{V \in \mathcal{V}} \mathbb{E}\|\widehat{V} - V\|^2 \geq c_2 \kappa^2 d = c_2 \kappa^2 (r + r')n.$$

**Step 3.** There are two cases. **(a)** Suppose $\xi^2 \geq n\kappa^2$, then $r' = \lfloor \xi^2/(n\kappa^2) \rfloor \geq 1$ and therefore

$$\inf_{\widehat{X}} \sup_{X^* \in \mathcal{M}_r^\xi} \mathbb{E}\big\|\widehat{X} - X^*\big\|_{\mathrm{F}}^2 \geq c_2 \kappa^2 (r + r')n \geq c_2 rn + c_3 \xi^2.$$

**(b)** Suppose $\xi^2 < n\kappa^2$. Then $r' = 0$

$$\inf_{\widehat{X}} \sup_{X^* \in \mathcal{M}_r^\xi} \mathbb{E}\big\|\widehat{X} - X^*\big\|_{\mathrm{F}}^2 \geq c_2 \kappa^2 rn \geq \frac{c_2}{2}(\kappa^2 rn + \xi^2), \tag{16}$$

where the first inequality follows from **Step 1** and **Step 2**, the second inequality follows from $\xi^2 < n\kappa^2$. $\qquad\square$

### B.3 Technical results

**Lemma 8.** *Let $\kappa > 0$ and $d \in \mathbb{Z}^+$, and denote*

$$\mathcal{V} = \{V \in \mathbb{R}^d, \|V\|_\infty \leq \kappa\}.$$

*Consider the model $Y \sim \mathcal{N}(V, \kappa^2 I_d)$ where $V \in \mathcal{V}$. Then there exists a universal constant $c$ such that*

$$\inf_{\widehat{V}} \sup_{V \in \mathcal{V}} \mathbb{E}\|\widehat{V} - V\|^2 \geq c\kappa^2 d,$$

*where infimum is taken over all the estimators $\widehat{V}$ based on $Y$.*

*Proof.* This is a standard minimax lower bound for bounded normal means; see for example Berry (1990) and Donoho et al. (1990). $\qquad\square$

## C   Proof of Theorem 1

Many of the proof techniques employed in this work build on ideas developed in the existing tensor literature, including Vannieuwenhoven et al. (2012); Zhang & Xia (2018); Han et al. (2022).

*Proof of Theorem 1.* For any orthogonal matrix $U \in \mathbb{O}^{p \times r}$, we denote $U_\perp \in \mathbb{O}^{p \times (p-r)}$ to be the orthogonal complement of $U$, and

$$\mathcal{P}_U = UU^\top \quad \text{and} \quad \mathcal{P}_{U_\perp} = U_\perp U_\perp^\top = I_p - \mathcal{P}_U.$$

For $k \in \{1, 2, 3\}$, let $U_k^* \in \mathbb{O}^{p_k \times r_k}$ be the matrix whose columns corresponds to the top $r_k$ singular vectors of $\mathcal{M}_k(X^*)$.

Throughout this proof, we assume the following good events hold:

$$\sup_{\substack{A \in \mathbb{R}^{p_1 \times p_2 \times p_3}, \\ \|A\|_{\mathrm{F}} \leq 1, A \in \mathcal{T}_{(r_1, r_2, r_3)}}} \langle Z, A \rangle \leq C\kappa \sqrt{r_1 r_2 r_3 + p_1 r_1 + p_2 r_2 + p_3 r_3}; \tag{17}$$

$$\|\mathcal{M}_1(Z) \cdot W_2 \otimes W_3\| \leq C\kappa \left(\sqrt{p_1 + s_2 s_3}\right) \text{ for non-random } W_2 \in \mathbb{O}^{p_2 \times s_2}, W_3 \in \mathbb{O}^{p_3 \times s_3}; \quad (18)$$

$$\left\|\sin\Theta(U_k^{(0)}, U_k^*)\right\| = \|U_k^{*\top} U_k^{(0)}\| \leq \frac{1}{2\sqrt{r_{\max}}} \text{ for } k \in \{1, 2, 3\}. \quad (19)$$

Indeed in Lemma 10, Lemma 11 and Corollary 17, we show that (17), (18), and (19) hold with probability at least $1 - C\exp(-cp_{\min})$.

Note that

$$\left\|\widetilde{X} - X^*\right\|_{\mathrm{F}} = \left\|Y \times_1 \mathcal{P}_{U_1^{(1)}} \times_2 \mathcal{P}_{U_2^{(1)}} \times_3 \mathcal{P}_{U_3^{(1)}} - X^*\right\|_{\mathrm{F}}$$

$$\leq \left\|X^* \times_1 \mathcal{P}_{U_1^{(1)}} \times_2 \mathcal{P}_{U_2^{(1)}} \times_3 \mathcal{P}_{U_3^{(1)}} - X^*\right\|_{\mathrm{F}} + \left\|Z \times_1 \mathcal{P}_{U_1^{(1)}} \times_2 \mathcal{P}_{U_2^{(1)}} \times_3 \mathcal{P}_{U_3^{(1)}}\right\|_{\mathrm{F}}$$

$$= \mathrm{I}_1 + \mathrm{I}_2.$$

**Step 1**. For the term $\mathrm{I}_2$, observe that

$$\mathrm{I}_2 = \left\|Z \times_1 \mathcal{P}_{U_1^{(1)}} \times_2 \mathcal{P}_{U_2^{(1)}} \times_3 \mathcal{P}_{U_3^{(1)}}\right\|_{\mathrm{F}}$$

$$= \sup_{W \in \mathbb{R}^{p_1 \times p_2 \times p_3}, \|W\|_{\mathrm{F}} \leq 1} \left\langle Z \times_1 \mathcal{P}_{U_1^{(1)}} \times_2 \mathcal{P}_{U_2^{(1)}} \times_3 \mathcal{P}_{U_3^{(1)}}, W \right\rangle$$

$$= \sup_{W \in \mathbb{R}^{p_1 \times p_2 \times p_3}, \|W\|_{\mathrm{F}} \leq 1} \left\langle Z, W \times_1 \mathcal{P}_{U_1^{(1)}} \times_2 \mathcal{P}_{U_2^{(1)}} \times_3 \mathcal{P}_{U_3^{(1)}} \right\rangle$$

$$\leq \sup_{\substack{A \in \mathbb{R}^{p_1 \times p_2 \times p_3}, \\ \|A\|_{\mathrm{F}} \leq 1, A \in \mathcal{T}_{(r_1, r_2, r_3)}}} \langle Z, A \rangle \leq C\kappa \left(r_1 r_2 r_3 + \sum_{k=1}^{3} p_k r_k\right)^{1/2},$$

where the second equality follows from the duality of the Frobenius norm, and the last inequality follows from (17).

**Step 2**. For the term $\mathrm{I}_1$, we have that

$$\mathrm{I}_1 = \left\|X^* \times_1 \mathcal{P}_{U_1^{(1)}} \times_2 \mathcal{P}_{U_2^{(1)}} \times_3 \mathcal{P}_{U_3^{(1)}} - X^*\right\|_{\mathrm{F}}$$

$$\leq \left\|X^* \times_1 (I_{p_1} - \mathcal{P}_{U_1^{(1)}})\right\|_{\mathrm{F}} + \left\|X^* \times_1 \mathcal{P}_{U_1^{(1)}} \times_2 (I_{p_2} - \mathcal{P}_{U_2^{(1)}})\right\|_{\mathrm{F}}$$

$$+ \left\|X^* \times_1 \mathcal{P}_{U_1^{(1)}} \times_2 \mathcal{P}_{U_2^{(1)}} \times_3 (I_{p_3} - \mathcal{P}_{U_3^{(1)}})\right\|_{\mathrm{F}}$$

$$\leq \sum_{k=1}^{3} \left\|X^* \times_k (I_{p_k} - \mathcal{P}_{U_k^{(1)}})\right\|_{\mathrm{F}},$$

where the last inequality follows from the observation that

$$\left\|X^* \times_1 \mathcal{P}_{U_1^{(1)}} \times_2 (I_{p_2} - \mathcal{P}_{U_2^{(1)}})\right\|_{\mathrm{F}} \leq \left\|X^* \times_2 (I_{p_2} - \mathcal{P}_{U_2^{(1)}})\right\|_{\mathrm{F}} \|\mathcal{P}_{U_1^{(1)}}\| \leq \left\|X^* \times_2 (I_{p_2} - \mathcal{P}_{U_2^{(1)}})\right\|_{\mathrm{F}}.$$

We only consider the case when $k = 1$, since the same arguments apply for $k = 2, 3$. We have that

$$\left\|X^* \times_1 (I_{p_1} - \mathcal{P}_{U_1^{(1)}})\right\|_{\mathrm{F}}$$

$$\leq \left\|X^* \times_1 (I_{p_1} - \mathcal{P}_{U_1^{(1)}}) \times_2 \mathcal{P}_{U_2^*}\right\|_{\mathrm{F}} + \left\|X^* \times_1 (I_{p_1} - \mathcal{P}_{U_1^{(1)}}) \times_2 (I_{p_2} - \mathcal{P}_{U_2^*})\right\|_{\mathrm{F}}$$

$$\leq \left\|X^* \times_1 (I_{p_1} - \mathcal{P}_{U_1^{(1)}}) \times_2 \mathcal{P}_{U_2^*}\right\|_{\mathrm{F}} + \left\|X^* \times_2 (I_{p_2} - \mathcal{P}_{U_2^*})\right\|_{\mathrm{F}}$$

$$\leq \left\|X^* \times_1 (I_{p_1} - \mathcal{P}_{U_1^{(1)}}) \times_2 \mathcal{P}_{U_2^*} \times_3 \mathcal{P}_{U_3^*}\right\|_{\mathrm{F}} + \left\|X^* \times_1 (I_{p_1} - \mathcal{P}_{U_1^{(1)}}) \times_2 \mathcal{P}_{U_2^*} \times_3 (I_{p_3} - \mathcal{P}_{U_3^*})\right\|_{\mathrm{F}}$$

$$+ \left\|X^* \times_2 (I_{p_2} - \mathcal{P}_{U_2^*})\right\|_{\mathrm{F}}$$

$$\leq \left\|X^* \times_1 (I_{p_1} - \mathcal{P}_{U_1^{(1)}}) \times_2 \mathcal{P}_{U_2^*} \times_3 \mathcal{P}_{U_3^*}\right\|_{\mathrm{F}} + \left\|X^* \times_3 (I_{p_3} - \mathcal{P}_{U_3^*})\right\|_{\mathrm{F}} + \left\|X^* \times_2 (I_{p_2} - \mathcal{P}_{U_2^*})\right\|_{\mathrm{F}}$$

$$= \left\| X^* \times_1 (I_{p_1} - \mathcal{P}_{U_1^{(1)}}) \times_2 \mathcal{P}_{U_2^*} \times_3 \mathcal{P}_{U_3^*} \right\|_{\mathrm{F}} + 2\xi_{(r_1, r_2, r_3)},$$

where the second follows from that $\|I_{p_1} - \mathcal{P}_{U_1^{(1)}}\| \le 1$, and the equality follows from Lemma 29.

**Step 3.** We bound $\|X^* \times_1 (I_{p_1} - \mathcal{P}_{U_1^{(1)}}) \times_2 \mathcal{P}_{U_2^*} \times_3 \mathcal{P}_{U_3^*}\|_{\mathrm{F}}$ in this step. Consider the different but relevant quantity

$$\left\| X^* \times_1 (I_{p_1} - \mathcal{P}_{U_1^{(1)}}) \times_2 \mathcal{P}_{U_2^{(0)}} \times_3 \mathcal{P}_{U_3^{(0)}} \right\|_{\mathrm{F}}$$

$$= \left\| (I_{p_1} - \mathcal{P}_{U_1^{(1)}}) \cdot \mathcal{M}_1(X^*) \cdot (\mathcal{P}_{U_2^{(0)}} \otimes \mathcal{P}_{U_3^{(0)}}) \right\|_{\mathrm{F}}$$

$$= \left\| (I_{p_1} - \mathcal{P}_{U_1^{(1)}}) \cdot \mathcal{M}_1(X^*) \cdot (U_2^{(0)} \otimes U_3^{(0)})(U_2^{(0)} \otimes U_3^{(0)})^\top \right\|_{\mathrm{F}}$$

$$= \left\| (I_{p_1} - \mathcal{P}_{U_1^{(1)}}) \cdot \mathcal{M}_1(X^*) \cdot (U_2^{(0)} \otimes U_3^{(0)}) \right\|_{\mathrm{F}}$$

$$\ge \left\| (I_{p_1} - \mathcal{P}_{U_1^{(1)}}) \cdot \mathcal{M}_1(X^*) \cdot \mathcal{P}_{U_2^* \otimes U_3^*} \cdot (U_2^{(0)} \otimes U_3^{(0)}) \right\|_{\mathrm{F}}$$

$$\quad - \left\| (I_{p_1} - \mathcal{P}_{U_1^{(1)}}) \cdot \mathcal{M}_1(X^*) \cdot (I_{p_2 p_3} - \mathcal{P}_{U_2^* \otimes U_3^*}) \cdot (U_2^{(0)} \otimes U_3^{(0)}) \right\|_{\mathrm{F}}$$

$$= \mathrm{II}_1 - \mathrm{II}_2,$$

where the first inequality follows from that $(U_2^{(0)} \otimes U_3^{(0)}) \in \mathbb{O}^{(p_2 p_3) \times (r_2 r_3)}$, and Lemma 22. Before analyzing the terms $\mathrm{II}_1$ and $\mathrm{II}_2$, we firstly note that

$$(U_2^* \otimes U_3^*)^\top \cdot (U_2^{(0)} \otimes U_3^{(0)}) = (U_2^{*\top} U_2^{(0)}) \otimes (U_3^{*\top} U_3^{(0)}),$$

$$\sigma_{\min}\left( (U_2^{*\top} U_2^{(0)}) \otimes (U_3^{*\top} U_3^{(0)}) \right) = \sigma_{\min}\left( U_2^{*\top} U_2^{(0)} \right) \sigma_{\min}\left( U_3^{*\top} U_3^{(0)} \right), \qquad (20)$$

$$\sigma_{\min}^2\left( U_k^{*\top} U_k^{(0)} \right) = 1 - \left\| U_{k\perp}^{*\top} U_k^{(0)} \right\|^2 = 1 - \left\| \sin\Theta(U_k^*, U_k^{(0)}) \right\|^2,$$

which hold due to the properties of the Kronecker product and Lemma 25. For the term $\mathrm{II}_1$, we have that

$$\mathrm{II}_1 = \left\| (I_{p_1} - \mathcal{P}_{U_1^{(1)}}) \cdot \mathcal{M}_1(X^*) \cdot (U_2^* \otimes U_3^*) \cdot (U_2^* \otimes U_3^*)^\top \cdot (U_2^{(0)} \otimes U_3^{(0)}) \right\|_{\mathrm{F}}$$

$$\ge \left\| (I_{p_1} - \mathcal{P}_{U_1^{(1)}}) \cdot \mathcal{M}_1(X^*) \cdot (U_2^* \otimes U_3^*) \right\|_{\mathrm{F}} \sigma_{\min}(U_2^{*\top} U_2^{(0)}) \sigma_{\min}(U_3^{*\top} U_3^{(0)})$$

$$= \left\| X^* \times_1 (I_{p_1} - \mathcal{P}_{U_1^{(1)}}) \times_2 \mathcal{P}_{U_2^*} \times_3 \mathcal{P}_{U_3^*} \right\|_{\mathrm{F}} \sqrt{\left( 1 - \left\| \sin\Theta(U_2^*, U_2^{(0)}) \right\|^2 \right) \left( 1 - \left\| \sin\Theta(U_3^*, U_3^{(0)}) \right\|^2 \right)}$$

$$\ge \frac{3}{4} \left\| X^* \times_1 (I_{p_1} - \mathcal{P}_{U_1^{(1)}}) \times_2 \mathcal{P}_{U_2^*} \times_3 \mathcal{P}_{U_3^*} \right\|_{\mathrm{F}}$$

where the first inequality follows from Lemma 21, and the second equality follows from (20), and the last inequality follows from (19).

For the term $\mathrm{II}_2$, we have that

$$\mathrm{II}_2 = \left\| (I_{p_1} - \mathcal{P}_{U_1^{(1)}}) \cdot \mathcal{M}_1(X^*) \cdot (I_{p_2 p_3} - \mathcal{P}_{U_2^* \otimes U_3^*}) \cdot (U_2^{(0)} \otimes U_3^{(0)}) \right\|_{\mathrm{F}}$$

$$= \left\| (I_{p_1} - \mathcal{P}_{U_1^{(1)}}) \cdot \mathcal{M}_1(X^*) \cdot (U_2^* \otimes U_3^*)_\perp \cdot (U_2^* \otimes U_3^*)_\perp^\top \cdot (U_2^{(0)} \otimes U_3^{(0)}) \right\|_{\mathrm{F}}$$

$$\le \left\| I_{p_1} - \mathcal{P}_{U_1^{(1)}} \right\| \left\| \mathcal{M}_1(X^*) \cdot (U_2^* \otimes U_3^*)_\perp \right\|_{\mathrm{F}} \left\| (U_2^* \otimes U_3^*)_\perp^\top \cdot (U_2^{(0)} \otimes U_3^{(0)}) \right\|$$

$$= \left\| \mathcal{M}_1(X^*) \cdot (U_2^* \otimes U_3^*)_\perp \right\|_{\mathrm{F}}.$$

Note that

$$\left\| \mathcal{M}_1(X^*) \cdot (U_2^* \otimes U_3^*)_\perp \right\|_{\mathrm{F}}$$

$$= \left\| \mathcal{M}_1(X^*) \cdot [U_{2\perp}^* \otimes U_3^* \quad U_2^* \otimes U_{3\perp}^* \quad U_{2\perp}^* \otimes U_{3\perp}^*] \right\|_{\mathrm{F}}$$

$$= \sqrt{\left\| \mathcal{M}_1(X^*) \cdot (U_{2\perp}^* \otimes U_3^*) \right\|_{\mathrm{F}}^2 + \left\| \mathcal{M}_1(X^*) \cdot (U_2^* \otimes U_{3\perp}^*) \right\|_{\mathrm{F}}^2 + \left\| \mathcal{M}_1(X^*) \cdot (U_{2\perp}^* \otimes U_{3\perp}^*) \right\|_{\mathrm{F}}^2}$$

$$\leq \|\mathcal{M}_1(X^*) \cdot (U_{2\perp}^* \otimes U_3^*)\|_{\mathrm{F}} + \|\mathcal{M}_1(X^*) \cdot (U_2^* \otimes U_{3\perp}^*)\|_{\mathrm{F}} + \|\mathcal{M}_1(X^*) \cdot (U_{2\perp}^* \otimes U_{3\perp}^*)\|_{\mathrm{F}}$$

$$= \|X^* \times_2 U_{2\perp}^* \times_3 U_3^*\|_{\mathrm{F}} + \|X^* \times_2 U_2^* \times_3 U_{3\perp}^*\|_{\mathrm{F}} + \|X^* \times_2 U_{2\perp}^* \times_3 U_{3\perp}^*\|_{\mathrm{F}}$$

$$\leq \|X^* \times_2 U_{2\perp}^*\|_{\mathrm{F}} + \|X^* \times_3 U_{3\perp}^*\|_{\mathrm{F}} + \|X^* \times_2 U_{2\perp}^*\|_{\mathrm{F}}$$

$$\leq 3\xi_{(r_1, r_2, r_3)}$$

where the last inequality follows from Lemma 29. Therefore,

$$\mathrm{II}_2 \leq 3\xi_{(r_1, r_2, r_3)}.$$

Combining $\mathrm{II}_1$ and $\mathrm{II}_2$, we have

$$\left\|X^* \times_1 (I_{p_1} - \mathcal{P}_{U_1^{(1)}}) \times_2 \mathcal{P}_{U_2^*} \times_3 \mathcal{P}_{U_3^*}\right\|_{\mathrm{F}} \leq \frac{4}{3} \left\|X^* \times_1 (I_{p_1} - \mathcal{P}_{U_1^{(1)}}) \times_2 \mathcal{P}_{U_2^{(0)}} \times_3 \mathcal{P}_{U_3^{(0)}}\right\|_{\mathrm{F}} + 4\xi_{(r_1, r_2, r_3)}.$$

**Step 4.** We bound $\left\|X^* \times_1 (I_{p_1} - \mathcal{P}_{U_1^{(1)}}) \times_2 \mathcal{P}_{U_2^{(0)}} \times_3 \mathcal{P}_{U_3^{(0)}}\right\|_{\mathrm{F}}$ in this step. Note that

$$\|X^* \times_1 (I_{p_1} - \mathcal{P}_{U_1^{(1)}}) \times_2 \mathcal{P}_{U_2^{(0)}} \times_3 \mathcal{P}_{U_3^{(0)}}\|_{\mathrm{F}}$$

$$= \|\mathcal{P}_{U_{1\perp}^{(1)}} \cdot \mathcal{M}_1(X^*) \cdot (\mathcal{P}_{U_2^{(0)}} \otimes \mathcal{P}_{U_3^{(0)}})\|_{\mathrm{F}} = \|\mathcal{P}_{U_{1\perp}^{(1)}} \cdot \mathcal{M}_1(X^*) \cdot (U_2^{(0)} \otimes U_3^{(0)})\|_{\mathrm{F}}$$

It suffices to apply Lemma 23 to bound

$$\|\mathcal{P}_{U_{1\perp}^{(1)}} \cdot \mathcal{M}_1(X^*) \cdot (U_2^{(0)} \otimes U_3^{(0)})\|_{\mathrm{F}}.$$

Since $U_{1\perp}^{(1)}$ corresponds to the SVD of $\mathcal{M}_1(Y) \cdot (U_2^{(0)} \otimes U_3^{(0)})$, let

$$A = \mathcal{M}_1(Y) \cdot (U_2^{(0)} \otimes U_3^{(0)}) \quad \text{and} \quad B = \mathcal{M}_1(X^*) \cdot (U_2^{(0)} \otimes U_3^{(0)}).$$

It follows from Lemma 23 that

$$\left\|X^* \times_1 (I_{p_1} - \mathcal{P}_{U_1^{(1)}}) \times_2 \mathcal{P}_{U_2^{(0)}} \times_3 \mathcal{P}_{U_3^{(0)}}\right\|_{\mathrm{F}} = \|\mathcal{P}_{U_{1\perp}^{(1)}} \cdot \mathcal{M}_1(X^*) \cdot (U_2^{(0)} \otimes U_3^{(0)})\|_{\mathrm{F}}$$

$$\leq C_1 \|B - B_{r_1}\|_F + C_2 \sqrt{r_1} \|A - B\|. \qquad (21)$$

Here

$$\|A - B\| = \left\|\mathcal{M}_1(Z) \cdot U_2^{(0)} \otimes U_3^{(0)}\right\|$$

$$= \left\|\mathcal{M}_1(Z) \cdot (U_2^* U_2^{*\top} + U_{2\perp}^* U_{2\perp}^{\top *}) U_2^{(0)} \otimes U_3^{(0)}\right\|$$

$$\leq \left\|\mathcal{M}_1(Z) \cdot (U_2^* U_2^{*\top} U_2^{(0)}) \otimes U_3^{(0)}\right\| + \left\|\mathcal{M}_1(Z) \cdot (U_{2\perp}^* U_{2\perp}^{\top *} U_2^{(0)}) \otimes U_3^{(0)}\right\|$$

$$\leq \left\|\mathcal{M}_1(Z) \cdot U_2^* \otimes U_3^{(0)}\right\| \|U_2^{*\top} U_2^{(0)}\| + \left\|\mathcal{M}_1(Z) \cdot U_{2\perp}^* \otimes U_3^{(0)}\right\| \|U_{2\perp}^{\top *} U_2^{(0)}\|. \qquad (22)$$

Note that

$$\left\|\mathcal{M}_1(Z) \cdot U_2^* \otimes U_3^{(0)}\right\| \|U_2^{*\top} U_2^{(0)}\| = \left\|\mathcal{M}_1(Z) \cdot U_2^* \otimes (U_3^* U_3^{*\top} + U_{3\perp}^* U_{3\perp}^{\top *}) U_3^{(0)}\right\|$$

$$\leq \left\|\mathcal{M}_1(Z) \cdot U_2^* \otimes (U_3^* U_3^{*\top} U_3^{(0)})\right\| + \left\|\mathcal{M}_1(Z) \cdot U_2^* \otimes (U_{3\perp}^* U_{3\perp}^{\top *}) U_3^{(0)}\right\|$$

$$\leq \|\mathcal{M}_1(Z) \cdot U_2^* \otimes U_3^*\| \|U_3^{*\top} U_3^{(0)}\| + \|\mathcal{M}_1(Z) \cdot U_2^* \otimes U_{3\perp}^*\| \|U_{3\perp}^{\top *} U_3^{(0)}\|$$

$$\leq C_4 \kappa (\sqrt{p_1 + r_2 r_3}) + C_5 \kappa (\sqrt{p_1 + p_3 r_2}) r_{\max}^{-1/2}, \qquad (23)$$

where the last inequality follows from (18) and the fact that $\|U_2^{*\top} U_2^{(0)}\| \leq 1$. In addition

$$\left\|\mathcal{M}_1(Z) \cdot U_{2\perp}^* \otimes U_3^{(0)}\right\| \|U_{2\perp}^{\top *} U_2^{(0)}\| \leq \left\|\mathcal{M}_1(Z) \cdot U_{2\perp}^* \otimes (U_3^* U_3^{\top *} + U_{3\perp}^* U_{3\perp}^{\top *}) U_3^{(0)}\right\| \|U_{2\perp}^{\top *} U_2^{(0)}\|$$

$$\leq \left\|\mathcal{M}_1(Z) \cdot U_{2\perp}^* \otimes (U_3^* U_3^{*\top} U_3^{(0)})\right\| \|U_{2\perp}^{\top *} U_2^{(0)}\| + \left\|\mathcal{M}_1(Z) \cdot U_2^* \otimes (U_{3\perp}^* U_{3\perp}^{\top *} U_3^{(0)})\right\| \|U_{2\perp}^{\top *} U_2^{(0)}\|$$

$$\leq \|\mathcal{M}_1(Z) \cdot U_{2\perp}^* \otimes U_3^*\| \|U_3^{*\top} U_3^{(0)}\| \|U_{2\perp}^{\top *} U_2^{(0)}\| + \|\mathcal{M}_1(Z) \cdot U_2^* \otimes U_{3\perp}^*\| \|U_{3\perp}^{\top *} U_3^{(0)}\| \|U_{2\perp}^{\top *} U_2^{(0)}\|$$

$$\leq C \kappa (\sqrt{p_1 + p_2 r_3}) r_{\max}^{-1/2} + C \kappa (\sqrt{p_1 + p_3 r_2}) r_{\max}^{-1}. \qquad (24)$$

Therefore (22), (23) and (24) leads to

$$\|A - B\| \leq C_4 \kappa(\sqrt{p_1} + \sqrt{r_2 r_3} + \sqrt{p_2 r_3} r_{\max}^{-1/2} + \sqrt{p_3 r_2} r_{\max}^{-1/2}). \tag{25}$$

In addition,

$$
\begin{aligned}
\|B - B_{r_1}\|_F &= \|\mathcal{M}_1(X^*) \cdot U_2^{(0)} \otimes U_3^{(0)} - \{\mathcal{M}_1(X^*) \cdot U_2^{(0)} \otimes U_3^{(0)}\}_{r_1}\|_F \\
&\leq \|\mathcal{M}_1(X^*) \cdot U_2^{(0)} \otimes U_3^{(0)} - \{\mathcal{M}_1(X^*)\}_{r_1} \cdot U_2^{(0)} \otimes U_3^{(0)}\|_F \\
&\leq \|(\mathcal{M}_1(X^*) - \{\mathcal{M}_1(X^*)\}_{r_1}) U_2^{(0)} \otimes U_3^{(0)}\|_F \\
&\leq \|\mathcal{M}_1(X^*) - \{\mathcal{M}_1(X^*)\}_{r_1}\|_F = \sqrt{\sum_{j=r_1+1}^{\mathrm{rank}(\mathcal{M}_1(X^*))} \sigma_j^2(\mathcal{M}_1(X^*))} \leq \xi_{(r_1, r_2, r_3)}.
\end{aligned}
\tag{26}
$$

Here $\{\mathcal{M}_1(X^*) \cdot U_2^{(0)} \otimes U_3^{(0)}\}_{r_1}$ indicate the best rank $r_1$ estimate of the matrix $\mathcal{M}_1(X^*) \cdot U_2^{(0)} \otimes U_3^{(0)}$ in the first equality, and so for any rank $r_1$ matrix $\Phi$,

$$\|\mathcal{M}_1(X^*) \cdot U_2^{(0)} \otimes U_3^{(0)} - \{\mathcal{M}_1(X^*) \cdot U_2^{(0)} \otimes U_3^{(0)}\}_{r_1}\|_F \leq \|\mathcal{M}_1(X^*) \cdot U_2^{(0)} \otimes U_3^{(0)} - \Phi\|_F;$$

the second inequality holds because $\{\mathcal{M}_1(X^*)\}_{r_1} \cdot U_2^{(0)} \otimes U_3^{(0)}$ is at most rank $r_1$, and the last inequality follows from Lemma 29.

It follows from (21), (25) and (26) that

$$\left\| X^* \times_1 (I_{p_1} - \mathcal{P}_{U_1^{(1)}}) \times_2 \mathcal{P}_{U_2^{(0)}} \times_3 \mathcal{P}_{U_3^{(0)}} \right\|_F \leq C_5 \kappa(\sqrt{p_1 r_1} + \sqrt{r_1 r_2 r_3} + \sqrt{p_2 r_2} + \sqrt{p_3 r_3}) + C_5 \xi_{(r_1, r_2, r_3)}.$$

**Step 5**. The conclusions of **Step 3** and **Step 4** lead to

$$\left\| X^* \times_1 (I_{p_1} - \mathcal{P}_{U_1^{(1)}}) \times_2 \mathcal{P}_{U_2^*} \times_3 \mathcal{P}_{U_3^*} \right\|_F \leq C_6 \kappa(\sqrt{p_1 r_1} + \sqrt{r_1 r_2 r_3} + \sqrt{p_2 r_2} + \sqrt{p_3 r_3}) + C_6 \xi_{(r_1, r_2, r_3)}.$$

This bound together with **Step 1** and **Step 2** leads to

$$\left\| \widetilde{X} - X^* \right\|_F \leq C_7 \kappa(\sqrt{p_1 r_1} + \sqrt{p_2 r_2} + \sqrt{p_3 r_3} + \sqrt{r_1 r_2 r_3}) + C_7 \xi_{(r_1, r_2, r_3)}.$$

$\square$

# D  PROOF OF THEOREM 3

*Proof of Theorem 3.* By the triangle inequality,

$$\|Y_{(r)} - X^*\|_F \leq \|X_{(r)}^* - X^*\|_F + \|Y_{(r)} - X_{(r)}^*\|_F = \xi_{(r)} + \|Y_{(r)} - X_{(r)}^*\|_F. \tag{27}$$

Since $\mathrm{rank}(Y_{(r)} - X_{(r)}^*) \leq 2r$, we have

$$\|Y_{(r)} - X_{(r)}^*\|_F^2 = \sum_{k=1}^{2r} \sigma_k^2(Y_{(r)} - X_{(r)}^*). \tag{28}$$

Write

$$Y_{(r)} - X_{(r)}^* = (X^* - X_{(r)}^*) + Z + (Y_{(r)} - Y).$$

It follows from Ky Fan's Theorem (Lemma 28) that,

$$\sqrt{\sum_{i=1}^{2r} \sigma_i^2(Y_{(r)} - X_{(r)}^*)} \leq \sqrt{\sum_{i=1}^{2r} \sigma_i^2(X^* - X_{(r)}^*)} + \sqrt{\sum_{i=1}^{2r} \sigma_i^2(Z)} + \sqrt{\sum_{i=1}^{2r} \sigma_i^2(Y_{(r)} - Y)}.$$

Using the identity $\sigma_i(X^* - X_{(r)}^*) = \sigma_{r+i}(X^*)$ and likewise for $Y$, we obtain

$$\|Y_{(r)} - X_{(r)}^*\|_F \leq \sqrt{\sum_{i=1}^{2r} \sigma_{r+i}^2(X^*)} + \sqrt{\sum_{i=1}^{2r} \sigma_i^2(Z)} + \sqrt{\sum_{i=1}^{2r} \sigma_{r+i}^2(Y)}. \tag{29}$$

To further simplify the third term, we apply Weyl's inequality (Theorem 27) to obtain

$$\sigma_{r+i}(Y) \leq \sigma_{r+i}(X^*) + \sigma_1(Z),$$

which gives

$$\sigma_{r+i}^2(Y) \leq 2 \left\{ \sigma_{r+i}^2(X^*) + \sigma_1^2(Z) \right\}.$$

Substituting this into (29), we get

$$\|Y_{(r)} - X_{(r)}^*\|_{\mathrm{F}} \leq \sqrt{\sum_{i=1}^{2r} \sigma_{r+i}^2(X^*)} + \sqrt{\sum_{i=1}^{2r} \sigma_i^2(Z)} + \sqrt{2\sum_{i=1}^{2r} \sigma_{r+i}^2(X^*) + 2\sqrt{r}\sigma_1(Z)}$$

$$\leq (1+\sqrt{2})\sqrt{\sum_{i=1}^{2r} \sigma_{r+i}^2(X^*)} + (2+\sqrt{2})\sqrt{r}\|Z\|$$

$$= (1+\sqrt{2})\sqrt{\sum_{i=r+1}^{3r} \sigma_i^2(X^*)} + (2+\sqrt{2})\sqrt{r}\|Z\|,$$

where the second inequality follows from the fact that for any $k \geq 1$, we have $\sigma_k(Z) \leq \sigma_1(Z) = \|Z\|$. Substituting the bound of $\|Y_{(r)} - X_{(r)}^*\|_{\mathrm{F}}$ into (27), we obtain

$$\|Y_{(r)} - X^*\|_{\mathrm{F}} \leq (2+\sqrt{2})\sqrt{\sum_{i=r+1}^{\min\{m,n\}} \sigma_i^2(X^*)} + (2+\sqrt{2})\sqrt{r}\|Z\|.$$

□

# E   DEVIATION BOUNDS

Throughout this appendix we work with centered sub-Gaussian random variables with parameter $\kappa^2$, and we assume (without loss of generality) that each such variable $X$ satisfies $\mathbb{E}X^2 = \kappa^2$. Any other case can be handled with additional constants in the bounds.

**Lemma 9** (Theorem 4.4.3 in Vershynin (2018)). *Assume all the entries of $Z \in \mathbb{R}^{m \times n}$ are independent mean-zero sub-Gaussian random variables, i.e.*

$$\|Z_{ij}\|_{\psi_2} = \sup_{q \geq 1} \mathbb{E}(|Z_{ij}|^q)^{1/q}/q^{1/2} \leq \kappa.$$

*Then there exist some universal constant $C > 0$, such that*

$$\|Z\| \leq C\kappa \left( \sqrt{m} + \sqrt{n} \right)$$

*with probability at least $1 - \exp(-(m+n)))$.*

**Lemma 10.** *Suppose all the entries of $Z \in \mathbb{R}^{p_1 \times p_2 \times p_3}$ are independent mean-zero sub-Gaussian random variables, i.e.*

$$\|Z_{ijk}\|_{\psi_2} = \sup_{q \geq 1} \mathbb{E}(|Z_{ijk}|^q)^{1/q}/q^{1/2} \leq \kappa.$$

*Then there exist some universal constants $C, c > 0$, such that*

$$\sup_{\substack{A \in \mathbb{R}^{p_1 \times p_2 \times p_3}, \\ \|A\|_{\mathrm{F}} \leq 1, A \in \mathcal{T}_{(r_1, r_2, r_3)}}} \langle Z, A \rangle \leq C\kappa \left( r_1 r_2 r_3 + \sum_{k=1}^{3} p_k r_k \right)^{1/2}$$

*with probability at least $1 - \exp(-c\sum_{k=1}^{3} p_k r_k)$.*

*Proof.* This directly follows from Lemma E.5 in Han et al. (2022). □

**Lemma 11.** *Suppose all the entries of $Z \in \mathbb{R}^{p_1 \times p_2 \times p_3}$ are independent mean-zero sub-Gaussian random variables, i.e.*

$$\|Z_{ijk}\|_{\psi_2} = \sup_{q \geq 1} \mathbb{E}(|Z_{ijk}|^q)^{1/q}/q^{1/2} \leq \kappa.$$

*Let $W_2 \in \mathbb{O}^{p_2 \times s_2}$ and $W_3 \in \mathbb{O}^{p_3 \times s_3}$ be non-random. Then there exists absolute positive constants $C_1, C_2$ and $c$ such that*

$$\mathbb{P}\bigg( \big\| \mathcal{M}_1(Z)\,(W_2 \otimes W_3) \big\| \geq C_1 \kappa(\sqrt{p_1 + s_2 s_3}) \bigg) \leq C_2 \exp\left(-cp_1\right),$$

*Proof.* It suffices to observe that $W_2 \otimes W_3 \in \mathbb{O}^{p_2 p_3 \times r_2 r_3}$. The desired result is a direct consequence of Lemma 13. $\qquad\square$

**Lemma 12.** *Suppose all the entries of $Z \in \mathbb{R}^{p_1 \times p_2 \times p_3}$ are independent mean-zero sub-Gaussian random variables, i.e.*

$$\|Z_{ijk}\|_{\psi_2} = \sup_{q \geq 1} \mathbb{E}(|Z_{ijk}|^q)^{1/q}/q^{1/2} \leq \kappa.$$

*Then there exists absolute positive constants $C_1, C_2$ and $c$ such that*

$$\mathbb{P}\bigg( \sup_{\substack{V_2 \in \mathbb{R}^{p_2 \times r_2},\, \|V_2\| \leq 1 \\ V_3 \in \mathbb{R}^{p_3 \times r_3},\, \|V_3\| \leq 1}} \big\| \mathcal{M}_1(Z)\,(V_2 \otimes V_3) \big\| \geq C_1 \kappa\Big( \sqrt{p_1 + r_2 r_3 + p_2 r_2 + p_3 r_3} \Big) \bigg)$$

$$\leq C_2 \exp\left(-c(p_1 + p_2 + p_3)\right),$$

*Proof.* It follows from the assumption that $(\mathcal{M}_1(Z))_{i,j} \overset{i.i.d}{\sim} \mathrm{subGaussian}(0, \kappa^2)$.

**Step 1.** For fixed $V_2 \in \mathbb{R}^{p_2 \times r_2}$ with $\|V_2\| \leq 1$ and $V_3 \in \mathbb{R}^{p_3 \times r_3}$ with $\|V_3\| \leq 1$, it follows that

$$\|V_2 \otimes V_3\| = \|V_2\| \|V_3\| \leq 1,$$

We upper bound $\|\mathcal{M}_1(Z) \cdot (V_2 \otimes V_3)\|$ for any fixed $V_2$ and $V_3$. Since,

$$\mathcal{M}(Z) \in \mathbb{R}^{p_1 \times p_2 p_3}, (\mathcal{M}(Z))_{i,j} \overset{i.i.d}{\sim} \mathrm{subGaussian}(0, \kappa^2), \quad \text{and} \quad \|V_2 \otimes V_3\| = \|V_2\| \|V_3\| = 1.$$

It follows from Lemma 13 that

$$\mathbb{P}\Big( \|\mathcal{M}_1(Z) \cdot (V_2 \otimes V_3)\| > x \Big) \leq 2 \exp\left( C(p_1 + r_2 r_3) - cx^2 \kappa^{-2} \right).$$

**Step 2.** Let $\mathcal{N}_{p_2, r_2}(\epsilon)$ denote an $\epsilon$ net of the set

$$\{A \in \mathbb{R}^{p_2 \times r_2} : \|A\| \leq 1\}$$

with respect to the operator norm $\| \cdot \|$. Similarly, let Let $\mathcal{N}_{p_3, r_3}(\epsilon)$ denote an $\epsilon$ net of the set

$$\{B \in \mathbb{R}^{p_3 \times r_3} : \|B\| \leq 1\}$$

with respect to the operator norm $\| \cdot \|$. Denote the random quantity

$$\psi = \sup_{\substack{V_2 \in \mathbb{R}^{p_2 \times r_2},\, \|V_2\| \leq 1 \\ V_3 \in \mathbb{R}^{p_3 \times r_3},\, \|V_3\| \leq 1}} \big\| \mathcal{M}_1(Z)\,(V_2 \otimes V_3) \big\|.$$

For any given $V_2 \in \mathbb{R}^{p_2 \times r_2}$ and $V_3 \in \mathbb{R}^q$ with $\|V_2\| \leq 1$ and $\|V_3\| \leq 1$, let $\widetilde{V}_2 \in \mathcal{N}_{p_2, r_2}(1/4)$ and $\widetilde{V}_3 \in \mathcal{N}_{p_3, r_3}(1/4)$ be such that

$$\|V_2 - \widetilde{V}_2\| \leq 1/4 \quad \text{and} \quad \|V_3 - \widetilde{V}_3\| \leq 1/4.$$

Then

$$\|\mathcal{M}_1(Z)V_2 \otimes V_3\| \leq \|\mathcal{M}_1(Z)(V_2 - \widetilde{V}_2) \otimes V_3\| + \|\mathcal{M}_1(Z)\widetilde{V}_2 \otimes (V_3 - \widetilde{V}_3)\| + \|\mathcal{M}_1(Z)\widetilde{V}_2 \otimes \widetilde{V}_3\|.$$

Note that $\|V_2 - \widetilde{V}_2\| \leq 1/4$. So

$$\|\mathcal{M}_1(Z)(V_2 - \widetilde{V}_2) \otimes V_3\| = \frac{1}{4}\|\mathcal{M}_1(Z)\{4(V_2 - \widetilde{V}_2)\} \otimes V_3\| \leq \frac{\psi}{4}.$$

Similarly

$$\|\mathcal{M}_1(Z)V_2 \otimes (V_3 - \widetilde{V}_3)\| \leq \frac{\psi}{4}.$$

In addition,

$$\|\mathcal{M}_1(Z)\widetilde{V}_2 \otimes \widetilde{V}_3\| \leq \sup_{V_2 \in \mathcal{N}_{p_2,r_2}(1/4), b \in \mathcal{N}_{p_3,r_3}(1/4)} \|\mathcal{M}_1(Z)V_2 \otimes V_3\|.$$

So for any $V_2$ and $V_3$,

$$\|\mathcal{M}_1(Z)V_2 \otimes V_3\| \leq \frac{1}{2}\psi + \sup_{V_2 \in \mathcal{N}_{p_2,r_2}(1/4), b \in \mathcal{N}_{p_3,r_3}(1/4)} \|\mathcal{M}_1(Z)V_2 \otimes V_3\|.$$

Taking sup over all $V_2 \in \{A \in \mathbb{R}^{p_2 \times r_2} : \|A\| \leq 1\}$ and $V_3 \in \{B \in \mathbb{R}^{p_3 \times r_3} : \|B\| \leq 1\}$, it follows that

$$\psi \leq \frac{1}{2}\psi + \sup_{V_2 \in \mathcal{N}_{p_2,r_2}(1/4), b \in \mathcal{N}_{p_3,r_3}(1/4)} \|\mathcal{M}_1(Z)V_2 \otimes V_3\|,$$

or simply

$$\psi \leq 2 \sup_{V_2 \in \mathcal{N}_{p_2,r_2}(1/4), b \in \mathcal{N}_{p_3,r_3}(1/4)} \|\mathcal{M}_1(Z)V_2 \otimes V_3\|.$$

**Step 3.** By Proposition 8 in Pajor (1998), the cardinality of $\mathcal{N}_{p_2,r_2}(\epsilon)$ is bounded $(\frac{C}{\epsilon})^{p_2 r_2}$, and $\mathcal{N}_{p_3,r_3}(\epsilon)$ is bounded $(\frac{C}{\epsilon})^{p_3 r_3}$. Therefore

$$\mathbb{P}(\psi \geq 2t) \leq \mathbb{P}\left( \sup_{V_2 \in \mathcal{N}_{p_2,r_2}(1/4), b \in \mathcal{N}_{p_3,r_3}(1/4)} \|\mathcal{M}_1(Z)V_2 \otimes V_3\| \geq t \right)$$

$$\leq C_2^{p_2 r_2} C_3^{p_3 r_3} \sup_{V_2 \in \mathcal{N}_{p_2,r_2}(1/4), b \in \mathcal{N}_{p_3,r_3}(1/4)} \mathbb{P}\left( \|\mathcal{M}_1(Z)V_2 \otimes V_3\| \geq t \right)$$

$$\leq 2 \exp\left( C(p_1 + r_2 r_3 + p_2 r_2 + p_3 r_3) - ct^2 \kappa^{-2} \right).$$

Here $C$ and $C_3$ are positive constants. The desired result follows by noting

$$\psi = \sup_{\substack{V_2 \in \mathbb{R}^{p_2 \times r_2}, \|V_2\| \leq 1 \\ V_3 \in \mathbb{R}^{p_3 \times r_3}, \|V_3\| \leq 1}} \left\| \mathcal{M}_1(Z)(V_2 \otimes V_3) \right\|.$$

$\square$

**Lemma 13.** *Suppose $Z \in \mathbb{R}^{n \times m}$, with $Z_{ij} \overset{i.i.d}{\sim} \text{subGaussian}(0, \kappa^2)$. Let $A \in \mathbb{R}^{p \times n}$ and $B \in \mathbb{R}^{m \times q}$ be non-random matrices. Then for any $t > 0$*

$$\mathbb{P}\left( \|AZB\| > t \right) \leq C_1 \exp\left( C(p+q) - \frac{ct^2}{\kappa^2 \|A\|^2 \|B\|^2} \right). \tag{30}$$

*Proof.*
**Step 1.** Let $u \in \mathbb{R}^n$ and $v \in \mathbb{R}^m$ be non-random. Then that $u^\top Z v = \sum_{i=1}^n \sum_{j=1}^n u_i Z_{ij} v_j$. Since $Z_{ij}$ are i.i.d. sub-Gaussian with parameter $\kappa^2$, it follows that $u^\top Z v$ is sub-Gaussian with parameter $\kappa^2 \|u\|_2^2 \|v\|_2^2$. Consequently by Hoeffding's inequality,

$$\mathbb{P}\left( |u^\top Z v| > t \right) \leq 2 \exp\left( -\frac{ct^2}{\kappa^2 \|u\|_2^2 \|v\|_2^2} \right).$$

**Step 2.** Let $a \in \mathbb{R}^p$ and $b \in \mathbb{R}^q$ be non-random vectors such that $\|a\|_2, \|b\|_2 \leq 1$. By **Step 1**, it follows that

$$\mathbb{P}\left( |a^\top AZBb| > t \right) \leq 2 \exp\left( -\frac{ct^2}{\kappa^2 \|Aa\|_2^2 \|Bb\|_2^2} \right) \leq 2 \exp\left( -\frac{ct^2}{\kappa^2 \|A\|^2 \|B\|^2} \right).$$

**Step 3.** Let $\mathcal{N}_p(\epsilon)$ be an $\epsilon$-net of the unit ball in $\mathbb{R}^p$. It follows that for any $a \in \mathbb{R}^p$ with $\|a\|_2 = 1$, there exists $\widetilde{a} \in \mathcal{N}_p(\epsilon)$ such that

$$\|a - \widetilde{a}\|_2 \leq \epsilon.$$

Similarly let $\mathcal{N}_q(\epsilon)$ be an $\epsilon$-net of the unit ball in $\mathbb{R}^q$.

Denote the random quantity

$$\psi = \|AZB\| = \sup_{a \in \mathbb{R}^p, b \in \mathbb{R}^q \|a\|_2 = \|b\|_2 = 1} |a^\top AZBb|.$$

For any given $a \in \mathbb{R}^p$ and $b \in \mathbb{R}^q$, let $\widetilde{a} \in \mathcal{N}_p(1/4)$ and $\widetilde{b} \in \mathcal{N}_q(1/4)$ be such that

$$\|a - \widetilde{a}\|_2 \leq 1/4 \quad \text{and} \quad \|b - \widetilde{b}\|_2 \leq 1/4.$$

Then

$$|a^\top AZBb| \leq |(a - \widetilde{a})^\top AZBb| + |\widetilde{a}^\top AZB(\widetilde{b} - b)| + |\widetilde{a}^\top AZB\widetilde{b}|.$$

Note that $\|a - \widetilde{a}\| \leq \frac{1}{4}$. So

$$|(a - \widetilde{a})^\top AZBb| = \frac{1}{4}|\{4(a - \widetilde{a})^\top\}AZBb| \leq \frac{\psi}{4}.$$

Similarly

$$|\widetilde{a}^\top AZB(\widetilde{b} - b)| \leq \frac{\psi}{4}.$$

In addition,

$$|\widetilde{a}^\top AZB\widetilde{b}| \leq \sup_{a \in \mathcal{N}_p(1/4), b \in \mathcal{N}_q(1/4)} |a^\top AZBb|.$$

So for any $a$ and $b$,

$$|a^\top AZBb| \leq \frac{1}{2}\psi + \sup_{a \in N_p(1/4), b \in \mathcal{N}_q(1/4)} |a^\top AZBb|.$$

Taking sup over all unit vectors $a \in \mathbb{R}^p$ and $b \in \mathbb{R}^q$, it follows that

$$\psi \leq \frac{1}{2}\psi + \sup_{a \in \mathcal{N}_p(1/4), b \in \mathcal{N}_q(1/4)} |a^\top AZBb|,$$

or simply

$$\psi \leq 2 \sup_{a \in \mathcal{N}_p(1/4), b \in \mathcal{N}_q(1/4)} |a^\top AZBb|.$$

**Step 4.** By Vershynin (2018), the cardinality of $\mathcal{N}_p(\epsilon)$ is bounded by $(\frac{C}{\epsilon})^p$, and the cardinality of $\mathcal{N}_q(\epsilon)$ is bounded by $(\frac{C}{\epsilon})^q$. Therefore

$$\mathbb{P}(\psi \geq 2t) = \mathbb{P}\left( \sup_{a \in \mathcal{N}_p(1/4), b \in \mathcal{N}_q(1/4)} |a^\top AZBb| \geq t \right) \leq C_2{}^p C_2{}^q \sup_{a \in \mathcal{N}_p(1/4), b \in \mathcal{N}_q(1/4)} \mathbb{P}(|a^\top AZBb| \geq t)$$

$$\leq 2C_2{}^p C_2{}^q \exp\left( -\frac{ct^2}{\kappa^2 \|A\|^2 \|B\|^2} \right),$$

where $c, C$ are positive constants. The desired result follows by noting $\psi = \|AZB\|$. $\qquad \square$

**Lemma 14.** *Suppose $Z \in \mathbb{R}^{n \times m}$, with $Z_{ij} \overset{i.i.d}{\sim} \text{subGaussian}(0, \kappa^2)$. Let $A \in \mathbb{R}^{m \times p}$ and $B \in \mathbb{R}^{m \times q}$ be non-random matrices. Then for any $t > 0$*

$$\mathbb{P}\left( \|A^\top Z^\top ZB - n\kappa^2 A^\top B\| > t \right) \leq C_1 \exp\left( C(p + q) - \min\left( \frac{t^2}{n\,\kappa^4 \|B\|^2 \|A\|^2}, \frac{t}{\kappa^2 \|B\| \|A\|} \right) \right), \tag{31}$$

*where $C$ and $C_1$ are positive constants.*

*Proof.* For any non-random $u, v \in \mathbb{R}^m$, it follows that

$$u^\top Z^\top Z v - n\kappa^2 u^\top v = \sum_{j=1}^n (u^\top Z_j)(v^\top Z_j) - \mathbb{E}\{(u^\top Z_j)(v^\top Z_j)\},$$

where $Z_j$ is the $j$-th row of $Z$. Note that $(u^\top Z_j)$ is sub-Gaussian with parameter $\kappa^2 \|u\|_2^2$, and $(v^\top Z_j)$ is sub-Gaussian with parameter $\kappa^2 \|v\|_2^2$. Since $Z$ have i.i.d. entries, it follows that $\{(u^\top Z_j)(v^\top Z_j)\}_{j=1}^n$ are i.i.d. sub-exponential with parameter $\kappa^4 \|u\|_2^2 \|v\|_2^2$. So

$$\mathbb{P}\bigg(|u^\top Z^\top Z v - n\kappa^2 u^\top v| \geq t\bigg) \leq 2\exp\bigg(-c\min\bigg\{\frac{t^2}{n\kappa^4\|u\|_2^2\|v\|_2^2}, \frac{t}{\kappa^2\|u\|_2\|v\|_2}\bigg\}\bigg).$$

**Step 1.** Let $\mathcal{N}_p(\epsilon)$ be the $\epsilon$-net of the unit ball in $\mathbb{R}^p$. It follows that for any $a \in \mathbb{R}^p$ with $\|a\|_2 = 1$, there exists $\widetilde{a} \in \mathcal{N}_p(\epsilon)$ such that
$$\|a - \widetilde{a}\|_2 \leq \epsilon.$$
Similarly let $\mathcal{N}_q(\epsilon)$ be the $\epsilon$-net of the unit ball in $\mathbb{R}^q$.

Denote the random quantity

$$\psi = \|A^\top Z^\top Z B - n\kappa^2 A^\top B\| = \sup_{a \in \mathbb{R}^p, b \in \mathbb{R}^q \|a\|_2 = \|b\|_2 = 1} |a^\top (A^\top Z^\top Z B - n\kappa^2 A^\top B)b|.$$

For any given $a \in \mathbb{R}^p$ and $b \in \mathbb{R}^q$, let $\widetilde{a} \in \mathcal{N}_p(1/4)$ and $\widetilde{b} \in \mathcal{N}_q(1/4)$ be such that

$$\|a - \widetilde{a}\|_2 \leq 1/4 \quad \text{and} \quad \|b - \widetilde{b}\|_2 \leq 1/4.$$

Then

$$|a^\top (A^\top Z^\top Z B - n\kappa^2 A^\top B)b| \leq |(a - \widetilde{a})^\top (A^\top Z^\top Z B - n\kappa^2 A^\top B)b|$$
$$+ |\widetilde{a}^\top (A^\top Z^\top Z B - n\kappa^2 A^\top B)(\widetilde{b} - b)| + |\widetilde{a}^\top (A^\top Z^\top Z B - n\kappa^2 A^\top B)\widetilde{b}|.$$

Note that $\|a - \widetilde{a}\|_2 \leq \frac{1}{4}$. So

$$|(a - \widetilde{a})^\top (A^\top Z^\top Z B - n\kappa^2 A^\top B)b| = \frac{1}{4}|\{4(a - \widetilde{a})^\top\}(A^\top Z^\top Z B - n\kappa^2 A^\top B)b| \leq \frac{\psi}{4}.$$

Similarly

$$|\widetilde{a}^\top (A^\top Z^\top Z B - n\kappa^2 A^\top B)(\widetilde{b} - b)| \leq \frac{\psi}{4}.$$

In addition,

$$|\widetilde{a}^\top (A^\top Z^\top Z B - n\kappa^2 A^\top B)\widetilde{b}| \leq \sup_{a \in \mathcal{N}_p(1/4), b \in \mathcal{N}_q(1/4)} |a^\top (A^\top Z^\top Z B - n\kappa^2 A^\top B)b|.$$

So for any $a$ and $b$,

$$|a^\top (A^\top Z^\top Z B - n\kappa^2 A^\top B)b| \leq \frac{1}{2}\psi + \sup_{a \in N_p(1/4), b \in \mathcal{N}_q(1/4)} |a^\top (A^\top Z^\top Z B - n\kappa^2 A^\top B)b|.$$

Taking sup over all unit vectors $a \in \mathbb{R}^p$ and $b \in \mathbb{R}^q$, it follows that

$$\psi \leq \frac{1}{2}\psi + \sup_{a \in \mathcal{N}_p(1/4), b \in \mathcal{N}_q(1/4)} |a^\top (A^\top Z^\top Z B - n\kappa^2 A^\top B)b|,$$

or simply

$$\psi \leq 2 \sup_{a \in \mathcal{N}_p(1/4), b \in \mathcal{N}_q(1/4)} |a^\top (A^\top Z^\top Z B - n\kappa^2 A^\top B)b|.$$

**Step 2.** By Vershynin (2018), the cardinality of $\mathcal{N}_p(\epsilon)$ is bounded by $(\frac{C}{\epsilon})^p$, and the cardinality of $\mathcal{N}_q(\epsilon)$ is bounded by $(\frac{C}{\epsilon})^q$, for a positive constant $C$. Therefore

$$\mathbb{P}(\psi \geq 2t) = \mathbb{P}\bigg(\sup_{a \in \mathcal{N}_p(1/4), b \in \mathcal{N}_q(1/4)} |a^\top (A^\top Z^\top Z B - n\kappa^2 A^\top B)b| \geq t\bigg)$$

$$\leq C_2{}^p C_2{}^q \sup_{a \in \mathcal{N}_p(1/4), b \in \mathcal{N}_q(1/4)} \mathbb{P}(|a^\top (A^\top Z^\top ZB - n\kappa^2 A^\top B)b| \geq t)$$

$$= C_2{}^p C_2{}^q \sup_{a \in \mathcal{N}_p(1/4), b \in \mathcal{N}_q(1/4)} \mathbb{P}(|(Aa)^\top Z^\top Z(Bb) - n\kappa^2 (Aa)^\top (Bb)| \geq t)$$

$$\leq 2 C_2{}^p C_2{}^q \exp\left( -c \min\left\{ \frac{t^2}{n\kappa^4 \|Aa\|_2^2 \|Bb\|_2^2}, \frac{t}{\kappa^2 \|Aa\|_2 \|Bb\|_2} \right\} \right),$$

where $C_2$ is a positive constant. The desired result follows from the observation that $\|Aa\|_2 \leq \|A\|\|a\|_2 \leq \|A\|$, and $\|Bb\|_2 \leq \|B\|\|b\|_2 \leq \|B\|$. $\qquad \square$

**Lemma 15.** *Suppose $Z \in \mathbb{R}^{m \times n}$ is a sub-Gaussian random matrix in the sense that for any $u \in \mathbb{R}^m, v \in \mathbb{R}^n$, it holds that*
$$\|u^\top Zv\|_{\psi_2} \leq \kappa \|u\|_2 \|v\|_2.$$
*Then with probability at least $1 - \exp(-c(m+n))$, it holds that*

$$\mathbb{P}\left(\|Z\| > t\right) \leq C_1 \exp\left( C(m+n) - \frac{c\, t^2}{\kappa^2} \right),$$

*where $c, C$ and $C_1$ are positive constants.*

*Proof.* By assumption,

$$\mathbb{P}\left(\left|u^\top Z\, v\right| > t\right) \leq 2 \exp\left( -\frac{c\, t^2}{\kappa^2 \, \|u\|_2^2 \, \|v\|_2^2} \right).$$

The rest of the proof is similar and simpler than Lemma 13 and is omitted.

$\qquad \square$

### E.1 SVD FOR UNBALANCED MATRICES

**Lemma 16.** *Suppose*
$$Y = X + Z \in \mathbb{R}^{n \times m},$$
*where $X$ is a non-random matrix of arbitrary rank, and $Z$ is a random matrix whose entries are i.i.d. sub-Gaussian random variables with mean zero and the sub-Gaussian norm $\|Z_{ij}\|_{\psi_2} = \kappa < \infty$. For any $r \leq \min\{n, m\}$, write the full SVD of $X$ as*

$$X = U\Sigma V^\top = [U_r \; U_\perp] \cdot \begin{bmatrix} \Sigma_r & \\ & \Sigma_\perp \end{bmatrix} \cdot \begin{bmatrix} V_r^\top \\ V_\perp^\top \end{bmatrix} = X_r + X_\perp.$$

*Here $U_r \in \mathbb{O}_{m,r}$, $V_r \in \mathbb{O}_{m,r}$ correspond to the leading $r$ left and right singular vectors of $X$. Suppose that*
$$\{\sigma_r(X) - \sigma_{r+1}(X)\}^2 \geq C_{\mathrm{gap}} \kappa^2 \{\sqrt{mn} + m\}$$
*where $C_{\mathrm{gap}} > 0$ is a sufficient large constant. Then with probability at least $1 - C_1 \exp(-C_2 n)$, it holds that*

$$\left\| \sin\Theta\left( \widehat{V}_r, V_r \right) \right\|^2 \leq C_3 \left\{ \frac{m\kappa^2}{(\sigma_r(X) - \sigma_{r+1}(X))^2} + \frac{\kappa^4 nm}{(\sigma_r(X) - \sigma_{r+1}(X))^4} \right\},$$

*where $C_1, C_2, C_3 > 0$ are absolute constants only depending on $C_{\mathrm{gap}}$.*

*Proof.* Note that by assumption, we have

$$\mathbb{E}[Z^\top Z] = n\kappa^2 I_m, \quad \mathbb{E}[Y^\top Y] = V_r \Sigma_r^2 V_r^\top + V_\perp \Sigma_\perp^2 V_\perp^\top + n\kappa^2 I_m, \quad \mathbb{E}[V_r^\top Y^\top Y V_r] = \Sigma_r^2 + n\kappa^2 I_r.$$

Define the diagonal weighting matrix

$$M = \mathrm{diag}\left( (\sigma_1^2 + n\kappa^2)^{-1/2}, \ldots, (\sigma_r^2 + n\kappa^2)^{-1/2} \right) \in \mathbb{R}^{r \times r}.$$

Then it holds that

$$YV_r M = (X_r + X_\perp + Z) V_r M = (X_r + Z) V_r M,$$

$$M^\top \mathbb{E}\left[V_r^\top Y^\top Y V_r\right] M = M^\top \mathbb{E}\left[V_r^\top (X_r + Z)^\top (X_r + Z) V_r\right] M = I_r.$$

**Step 1**. For $\sigma_r(Y V_r)$, observe that

$$
\begin{aligned}
\sigma_r^2(Y V_r) &= \sigma_r^2\left(\{X_r + Z\} V_r\right) = \sigma_r^2\left(\{X_r + Z\} V_r M M^{-1}\right) \geq \sigma_r^2(\{X_r + Z\} V_r M) \sigma_{\min}^2(M^{-1}) \\
&= \sigma_r^2\left(\{X_r + Z\} V_r M\right)\{\sigma_r^2(X) + n\kappa^2\} \\
&= \sigma_r\left(M^\top V_r^\top \{X_r + Z\}^\top \{X_r + Z\} V_r M\right)\{\sigma_r^2(X) + n\kappa^2\}
\end{aligned}
\tag{32}
$$

where the inequality follows from Lemma 20.

Consider the term $\sigma_r\left(M^\top V_r^\top \{X_r + Z\}^\top \{X_r + Z\} V_r M\right)$. Note that

$$
\begin{aligned}
&M^\top V_r^\top (X_r + Z)^\top (X_r + Z) V_r M - I_r \\
&= M^\top V_r^\top (X_r + Z)^\top (X_r + Z) V_r M - \mathbb{E}\left[M^\top V_r^\top (X_r + Z)^\top (X_r + Z) V_r M\right] \\
&= \underbrace{M^\top V_r^\top X_r^\top X_r V_r M - \mathbb{E}\left[M^\top V_r^\top X_r^\top X_r V_r M\right]}_{=0} + M^\top V_r^\top X_r^\top Z V_r M \\
&\quad + M^\top V_r^\top Z^\top X_r V_r M + M^\top V_r^\top Z^\top Z V_r M - \mathbb{E}\left[M^\top V_r^\top Z^\top Z V_r M\right] \\
&\quad + \underbrace{\mathbb{E}\left[M^\top V_r^\top X_r^\top Z V_r M\right]}_{=0} + \underbrace{\mathbb{E}\left[M^\top V_r^\top Z^\top X_r V_r M\right]}_{=0} \\
&= M^\top V_r^\top X_r^\top Z V_r M + M^\top V_r^\top Z^\top X_r V_r M + M^\top V_r^\top \left(Z^\top Z - n\kappa^2 I_m\right) V_r M.
\end{aligned}
\tag{33}
$$

Since,

$$\|X_r V_r M\|^2 = \max_{k=1,\ldots,r} \frac{\sigma_k^2(X)}{\sigma_k^2(X) + n\kappa^2} \leq 1, \qquad \text{and} \qquad \|V_r M\|^2 = \|M\|^2 = \frac{1}{\sigma_r^2(X) + n\kappa^2},$$

it follows by Lemma 13 that

$$\mathbb{P}\left(\left\|M^\top V_r^\top X_r^\top Z V_r M\right\| \geq x\right) \leq 2 \exp\left(Cr - cx^2 \frac{\sigma_r^2(X) + n\kappa^2}{\kappa^2}\right).
\tag{34}$$

Similarly, Lemma 14 implies that

$$
\begin{aligned}
&\mathbb{P}\left(\left\|M^\top V_r^\top \left(Z^\top Z - n\kappa^2 I_m\right) V_r M\right\| \geq x\right) \\
&\leq 2 \exp\left(Cr - c\min\left\{x^2 \frac{\left(\sigma_r^2(X) + n\kappa^2\right)^2}{n\kappa^4}, \ x \frac{\sigma_r^2(X) + n\kappa^2}{\kappa^2}\right\}\right), \\
&\leq 2 \exp\left(Cr - c \frac{\sigma_r^2(X) + n\kappa^2}{\kappa^2} \min\left\{x^2, x\right\}\right),
\end{aligned}
\tag{35}
$$

where the last inequality follows from the fact that $\frac{\{\sigma_r^2(X) + n\kappa^2\}^2}{\kappa^4 n} \geq \frac{\sigma_r^2(X) + n\kappa^2}{\kappa^2}$. Thus, combining (33), (34), and (35), we have

$$\mathbb{P}\left(\left\|M^\top V_r^\top (X_r + Z)^\top (X_r + Z) V_r M - I_r\right\| \geq x\right) \leq 6 \exp\left(Cr - c\frac{\sigma_r^2(X) + n\kappa^2}{\kappa^2} \min\{x^2, x\}\right).$$

$$\tag{36}$$

Here we apply a union bound to the three terms in (33) with thresholds $x/3$ each. The same tail bound as in (34) holds for $M^\top V_r^\top Z^\top X_r V_r M$ by symmetry, and (35) controls the centered quadratic term $M^\top V_r^\top (Z^\top Z - n\kappa^2 I_m) V_r M$. All absolute constants are absorbed into $C, c$. This implies that

$$
\begin{aligned}
&\mathbb{P}\left(\sigma_r\left(M^\top V_r^\top \{X_r + Z\}^\top \{X_r + Z\} V_r M\right) \geq 1 - x\right) \\
&\geq 1 - 6 \exp\left(Cr - c\frac{\sigma_r^2(X) + n\kappa^2}{\kappa^2} \min\left\{x^2, x\right\}\right),
\end{aligned}
\tag{37}
$$

(37) and (32) together imply that

$$\mathbb{P}\Big(\sigma_r^2(YV_r) \geq \{\sigma_r^2(X) + n\kappa^2\}(1-x)\Big) \geq 1 - 6\exp\left(Cr - c\,\frac{\sigma_r^2(X) + n\kappa^2}{\kappa^2}\min\{x^2,\,x\}\right).$$

Setting $x = \frac{1}{6}\frac{\sigma_r^2(X) - \sigma_{r+1}^2(X)}{\sigma_r^2(X) + n\kappa^2}$, we have

$$\mathbb{P}\left(\sigma_r^2(YV_r) \geq \sigma_r^2(X) + n\kappa^2 - \frac{\sigma_r^2(X) - \sigma_{r+1}^2(X)}{6}\right)$$

$$\geq 1 - 6\exp\left(Cr - c\min\left\{\frac{1}{36\kappa^2}\frac{\left(\sigma_r^2(X) - \sigma_{r+1}^2(X)\right)^2}{\sigma_r^2(X) + n\kappa^2},\ \frac{\sigma_r^2(X) - \sigma_{r+1}^2(X)}{6\kappa^2}\right\}\right). \tag{38}$$

**Step 2**. We upper bound the term $\sigma_{r+1}^2(Y)$. Note that

$$\sigma_{r+1}(Y) = \min_{\mathrm{rank}(B)\leq r}\|Y - B\| \leq \|Y - Y\cdot V_r V_r^\top\| = \sigma_{\max}(YV_\perp).$$

Moreover,

$$\begin{aligned}
\sigma_{\max}^2(YV_\perp) &= \left\|V_\perp^\top Y^\top Y V_\perp\right\| = \left\|V_\perp^\top(X_r + X_\perp + Z)^\top(X_r + X_\perp + Z)V_\perp\right\| \\
&= \left\|V_\perp^\top(X_\perp + Z)^\top(X_\perp + Z)V_\perp\right\| \\
&\leq \left\|V_\perp^\top Z^\top Z V_\perp\right\| + \left\|V_\perp^\top Z^\top X_\perp V_\perp\right\| + \left\|V_\perp^\top X_\perp^\top Z V_\perp\right\| + \left\|V_\perp^\top X_\perp^\top X_\perp V_\perp\right\| \\
&= \underbrace{\left\|V_\perp^\top Z^\top Z V_\perp\right\|}_{\mathrm{I}_1} + 2\underbrace{\left\|V_\perp^\top Z^\top X_\perp V_\perp\right\|}_{\mathrm{I}_2} + \underbrace{\left\|V_\perp^\top X_\perp^\top X_\perp V_\perp\right\|}_{\mathrm{I}_3}.
\end{aligned} \tag{39}$$

For the term $\mathrm{I}_3$, we have

$$\mathrm{I}_3 = \sigma_1^2(X_\perp V_\perp) = \sigma_1^2(U_\perp \Sigma_\perp V_\perp^\top V_\perp) = \sigma_1^2(\Sigma_\perp) = \sigma_{r+1}^2(X). \tag{40}$$

For $\mathrm{I}_2$, note that

$$\|X_\perp V_\perp\|^2 = \|U_\perp \Sigma_\perp\|^2 = \sigma_{r+1}^2(X), \qquad \text{and} \qquad \|V_\perp\|^2 = 1.$$

it follows from Lemma 13 that

$$\mathbb{P}\Big(\mathrm{I}_2 \geq x\Big) \leq 2\exp\left(C_1 m - \frac{c_1 x^2}{\kappa^2\,\sigma_{r+1}^2(X)}\right). \tag{41}$$

For $\mathrm{I}_1$, note that by Lemma 14 and the fac that $V_\perp^\top V_\perp = I_{m-r}$, we have

$$\mathbb{P}\Big(\left\|V_\perp^\top Z^\top Z V_\perp - n\kappa^2 I_{m-r}\right\| \geq t\Big) \leq 2\exp\left(C_2 m - c_2\min\left\{\frac{t^2}{n\,\kappa^4},\ \frac{t}{\kappa^2}\right\}\right),$$

$$\implies \mathbb{P}\Big(\mathrm{I}_1 \geq n\kappa^2(1+t)\Big) \leq 2\exp\left(C_2 m - c_2 n\min\{t^2,\,t\}\right). \tag{42}$$

Combining the calculations in this step, with $x = \kappa\sigma_{r+1}(X)\sqrt{\frac{2C_1}{c_1}m}$ in (41), and with $t = \frac{\sigma_r^2(X) - \sigma_{r+1}^2(X)}{6n\kappa^2}$ in (42), it follows that

$$\mathbb{P}\left(\sigma_{r+1}^2(Y) \geq n\kappa^2 + \frac{\sigma_r^2(X) - \sigma_{r+1}^2(X)}{6} + \kappa\sigma_{r+1}(X)\sqrt{\frac{2C_1}{c_1}m} + \sigma_{r+1}^2(X)\right)$$

$$\leq 2\exp\left(C_2 m - c_2\min\left\{\frac{\left(\sigma_r^2(X) - \sigma_{r+1}^2(X)\right)^2}{36\,\kappa^4\,n},\ \frac{\sigma_r^2(X) - \sigma_{r+1}^2(X)}{6\,\kappa^2}\right\}\right) + 2\exp(-C_3\,m). \tag{43}$$

**Step 3**. Recall we define

$$M = \mathrm{diag}\left((\sigma_1^2 + n\kappa^2)^{-1/2},\ldots,(\sigma_r^2 + n\kappa^2)^{-1/2}\right) \in \mathbb{R}^{r\times r}.$$

We have

$$\|\mathcal{P}_{YV_r}YV_\perp\| = \|\mathcal{P}_{YV_rM}YV_\perp\|$$

$$= \left\|(YV_rM)\left((YV_rM)^\top(YV_rM)\right)^{-1}(YV_rM)^\top YV_\perp\right\|$$

$$\leq \left\|(YV_rM)\left((YV_rM)^\top(YV_rM)\right)^{-1}\right\|\left\|M^\top V_r^\top Y^\top YV_\perp\right\|$$

$$\leq \sigma_{\min}^{-1}(YV_rM)\left\|M^\top V_r^\top Y^\top YV_\perp\right\| = \sigma_r^{-1}(YV_rM)\left\|M^\top V_r^\top Y^\top YV_\perp\right\|, \quad (44)$$

where the first equality follows from the fact that $YV_r$ and $YV_rM$ have the same column spaces (since $M$ is invertible), the last inequality follows from Lemma 18, and for the last equality we use that the singular values of $YV_rM$ are in nonincreasing order so that its smallest singular value equals $\sigma_r(YV_rM)$.

By (36), we have for every $x > 0$

$$\mathbb{P}\left(\left\|M^\top V_r^\top(X_r+Z)^\top(X_r+Z)V_rM - I_r\right\| \geq x\right) \leq 6\exp\left(Cr - c\,\frac{\sigma_r^2(X) + n\kappa^2}{\kappa^2}\min\{x^2, x\}\right).$$

Taking $x = \frac{1}{2}$ gives

$$\mathbb{P}\left(\left\|M^\top V_r^\top(X_r+Z)^\top(X_r+Z)V_rM - I_r\right\| < \tfrac{1}{2}\right) \geq 1 - 6\exp\left(Cr - c\,\frac{\sigma_r^2(X) + n\kappa^2}{4\kappa^2}\right). \quad (45)$$

In particular, on this event all eigenvalues of $M^\top V_r^\top(X_r+Z)^\top(X_r+Z)V_rM$ are at least $1/2$, so $\sigma_r^2(YV_rM) \geq 1/2$ with the same probability bound. Consider $\left\|M^\top V_r^\top Y^\top YV_\perp\right\|$. Since $V_r^\top X_\perp^\top = 0$, $X_rV_\perp = 0$ and $X_\perp^\top X_r = 0$, it follows that

$$M^\top V_r^\top Y^\top YV_\perp = M^\top V_r^\top(X_r + X_\perp + Z)^\top(X_r + X_\perp + Z)V_\perp$$

$$= M^\top V_r^\top X_r^\top ZV_\perp + M^\top V_r^\top Z^\top X_\perp V_\perp + M^\top V_r^\top Z^\top ZV_\perp$$

$$= M^\top V_r^\top X_r^\top ZV_\perp + M^\top V_r^\top Z^\top X_\perp V_\perp + M^\top V_r^\top Z^\top ZV_\perp - \underbrace{M^\top V_r^\top\left(n\kappa^2 I_m\right)V_\perp}_{=0},$$

Since,

$$\|X_rV_rM\|^2 \leq 1, \ \|V_rM\|^2 = \frac{1}{\sigma_r^2(X) + n\kappa^2} \text{ and } \|X_\perp V_\perp\|^2 = \|X_\perp\|^2 = \sigma_{r+1}^2(X),$$

it follows from Lemma 13 that

$$\mathbb{P}\left(\left\|M^\top V_r^\top X_r^\top ZV_\perp\right\| \geq x\right) \leq 2\exp\left(Cm - \frac{cx^2}{\kappa^2}\right),$$

$$\mathbb{P}\left(\left\|M^\top V_r^\top Z^\top X_\perp V_\perp\right\| \geq x\right) \leq 2\exp\left(Cm - \frac{cx^2}{\kappa^2}\,\frac{\sigma_r^2(X) + n\kappa^2}{\sigma_{r+1}^2(X)}\right) \leq 2\exp\left(Cm - \frac{cx^2}{\kappa^2}\right). \quad (46)$$

Similarly, Lemma 14 implies that

$$\mathbb{P}\left(\left\|M^\top V_r^\top\left(Z^\top Z - n\kappa^2 I_m\right)V_\perp\right\| \geq x\right)$$

$$\leq 2\exp\left(Cm - c\min\left\{x^2\,\frac{\sigma_r^2(X) + n\kappa^2}{n\kappa^4}, x\,\frac{\sqrt{\sigma_r^2(X) + n\kappa^2}}{\kappa^2}\right\}\right), \quad (47)$$

$$\leq 2\exp\left(Cm - c\min\left\{\frac{x^2}{\kappa^2}, x\,\frac{\sqrt{\sigma_r^2(X) + n\kappa^2}}{\kappa^2}\right\}\right),$$

where the last inequality uses $\frac{\sigma_R^2(X) + n\kappa^2}{n\,\kappa^2} \geq 1$. Thus, combining (44), (45), (46) and (47) with $x = \kappa\sqrt{\frac{2C}{c}m}$, we have

$$\mathbb{P}\left(\left\|\mathcal{P}_{YV_r}YV_\perp\right\|^2 \geq \frac{36C}{c}\,m\kappa^2\right) \leq 6\exp\left(Cr - c\frac{\sigma_r^2(X) + n\kappa^2}{4\kappa^2}\right) + 4\exp(-Cm) \quad (48)$$

$$+ 2\exp\left(Cm - c\min\left\{\frac{2C}{c}\,m,\ \sqrt{\frac{2C}{c}\,m\,\frac{\sigma_r^2(X)+n\kappa^2}{\kappa^2}}\right\}\right).$$

**Step 4**. Define the event

$$\mathcal{E} = \left\{\sigma_r^2(YV_r) \geq \sigma_r^2(X) + n\kappa^2 - \frac{\sigma_r^2(X) - \sigma_{r+1}^2(X)}{6};\right.$$

$$\sigma_{r+1}^2(Y) \leq n\kappa^2 + \frac{\sigma_r^2(X) - \sigma_{r+1}^2(X)}{6} + \kappa\sigma_{r+1}(X)\sqrt{\frac{2C}{c}m} + \sigma_{r+1}^2(X);$$

$$\left.\|\mathcal{P}_{YV_r}YV_\perp\|^2 \leq \frac{8C}{c}\,m\kappa^2\right\}.$$

It follows from (38), (43) and (48) that by the union bound,

$$\mathbb{P}(\mathcal{E}^c) \leq 6\exp\left(Cr - c\min\left\{\frac{1}{36\kappa^2}\frac{\left(\sigma_r^2(X)-\sigma_{r+1}^2(X)\right)^2}{\sigma_r^2(X)+n\kappa^2},\ \frac{\sigma_r^2(X)-\sigma_{r+1}^2(X)}{6\kappa^2}\right\}\right) \tag{49}$$

$$+ 2\exp\left(Cm - c\min\left\{\frac{\left(\sigma_r^2(X)-\sigma_{r+1}^2(X)\right)^2}{36\,\kappa^4\,n},\ \frac{\sigma_r^2(X)-\sigma_{r+1}^2(X)}{6\,\kappa^2}\right\}\right) + 2\exp\left(-Cm\right) \tag{50}$$

$$+ 6\exp\left(Cr - c\frac{\sigma_r^2(X)+n\kappa^2}{4\kappa^2}\right) + 4\exp(-Cm) \tag{51}$$

$$+ 2\exp\left(Cm - c\min\left\{\frac{2C}{c}\,m,\ \sqrt{\frac{2C}{c}\,m\frac{\sigma_r^2(X)+n\kappa^2}{\kappa^2}}\right\}\right). \tag{52}$$

In what follows, we show that under the SNR assumption

$$\left(\sigma_r(X) - \sigma_{r+1}(X)\right)^2 \geq C_{\text{gap}}\kappa^2\left(\sqrt{nm}+m\right)$$

with sufficient large absolute constant $C_{\text{gap}} > 0$, we have

$$\mathbb{P}(\mathcal{E}^c) \leq C\exp\left(-Cm\right),$$

where $C > 0$ is an absolute constant, appropriately scaled to absorb the other constants.

We illustrate how to bound (49), as the rest of the terms can be bounded in a similar and simpler way. Note that

$$\frac{1}{36\kappa^2}\frac{\left(\sigma_r^2(X)-\sigma_{r+1}^2(X)\right)^2}{\sigma_r^2(X)+n\kappa^2} \geq \frac{\left(\sigma_r^2(X)-\sigma_{r+1}^2(X)\right)^2}{72\kappa^2}\min\left\{\frac{1}{\sigma_r^2(X)},\ \frac{1}{n\kappa^2}\right\}$$

$$\geq \min\left\{\frac{\left(\sigma_r^2(X)-\sigma_{r+1}^2(X)\right)^2}{72\,\kappa^2\sigma_r^2(X)},\ \frac{\left(\sigma_r^2(X)-\sigma_{r+1}^2(X)\right)^2}{72n\kappa^4}\right\}.$$

We have

$$\frac{\left(\sigma_r^2(X)-\sigma_{r+1}^2(X)\right)^2}{72\,\kappa^2\sigma_r^2(X)} = \frac{\left(\sigma_r(X)+\sigma_{r+1}(X)\right)^2\left(\sigma_r(X)-\sigma_{r+1}(X)\right)^2}{72\,\kappa^2\sigma_r^2(X)} \geq \frac{\left(\sigma_r(X)-\sigma_{r+1}(X)\right)^2}{72\,\kappa^2} \geq \frac{2C}{c}m,$$

$$\frac{\left(\sigma_r^2(X)-\sigma_{r+1}^2(X)\right)^2}{72\,\kappa^4\,n} \geq \frac{\left(\sigma_r(X)-\sigma_{r+1}(X)\right)^4}{72\,\kappa^4\,n} \geq \frac{2C}{c}m.$$

So

$$\frac{1}{36\kappa^2}\frac{\left(\sigma_r^2(X)-\sigma_{r+1}^2(X)\right)^2}{\sigma_r^2(X)+n\kappa^2} \geq \frac{2C}{c}m.$$

In addition,

$$\frac{\sigma_R^2(X)-\sigma_{R+1}^2(X)}{6\kappa^2} \geq \frac{\left(\sigma_R^2(X)-\sigma_{R+1}^2(X)\right)^2}{72\,\kappa^2\sigma_R^2(X)} \geq \frac{2C}{c}m.$$

So (49) $\leq C \exp\left(-Cm\right)$, for some absolute constant $C > 0$.

**Step 5.** Under the event $\mathcal{E}$, by Lemma 24, we have that

$$\left\|\sin\Theta\left(\widehat{V}_r, V_r\right)\right\|^2 \leq \frac{\sigma_r^2(YV_r)\left\|\mathcal{P}_{YV_r}YV_\perp\right\|^2}{\left(\sigma_r^2(YV_r) - \sigma_{r+1}^2(Y)\right)^2} \leq C_7 \frac{\sigma_r^2(YV_r)m\kappa^2}{\left(\sigma_r^2(YV_r) - \sigma_{r+1}^2(Y)\right)^2}$$

$$\leq C_8 \frac{\left(\sigma_r^2(X) + n\kappa^2 - \frac{\sigma_r^2(X) - \sigma_{r+1}^2(X)}{6}\right)m\kappa^2}{\left(\sigma_r^2(X) + n\kappa^2 - \frac{\sigma_r^2(X) - \sigma_{r+1}^2(X)}{6} - \sigma_{r+1}^2(Y)\right)^2}$$

$$\leq C_8 \frac{\left(\sigma_r^2(X) + n\kappa^2 - \frac{\sigma_r^2(X) - \sigma_{r+1}^2(X)}{6}\right)m\kappa^2}{\left(\left(1 - \frac{1}{3}\right)\left(\sigma_r^2(X) - \sigma_{r+1}^2(X)\right) - \kappa\sigma_{r+1}(X)\sqrt{\frac{2C}{c}m}\right)^2}$$

$$\leq C_9 \frac{\left(\sigma_r^2(X) + n\kappa^2 - \frac{\sigma_r^2(X) - \sigma_{r+1}^2(X)}{6}\right)m\kappa^2}{(1 - \frac{1}{2})^2\left(\sigma_r^2(X) - \sigma_{r+1}^2(X)\right)^2}$$

$$\leq C_{10} \frac{\left(\sigma_r^2(X) + n\kappa^2 - \frac{\sigma_r^2(X) - \sigma_{r+1}^2(X)}{6}\right)m\kappa^2}{\left(\sigma_r^2(X) - \sigma_{r+1}^2(X)\right)^2},$$

Here, the third inequality follows from the fact that $x^2/(x^2 - y^2)^2$ is a decreasing function of $x$ and an increasing function of $y$ when $x > y \geq 0$, together with the fact that the event $\mathcal{E}$ holds. The fifth inequality follows from the fact that, under the assumption $\left(\sigma_r(X) - \sigma_{r+1}(X)\right)^2 \geq C_{\text{gap}}\kappa^2(\sqrt{nm} + m)$ with $C_{\text{gap}} > 0$ being large enough,

$$\frac{\left(\sigma_r^2(X) - \sigma_{r+1}^2(X)\right)^2}{36} = \left(\sigma_r(X) + \sigma_{r+1}(X)\right)^2 \frac{\left(\sigma_r(X) - \sigma_{r+1}(X)\right)^2}{36}$$

$$\geq \sigma_r^2(X)\frac{C_{\text{gap}}m\kappa^2}{36} \geq Cm\kappa^2\sigma_r^2(X) \geq Cm\kappa^2\sigma_{r+1}^2(X).$$

Therefore, with probability at least $1 - C\exp(-Cm)$,

$$\left\|\sin\Theta\left(\widehat{V}_r, V_r\right)\right\|^2 \leq C_3 \frac{\left(\sigma_r^2(X) + n\kappa^2 - \frac{\sigma_r^2(X) - \sigma_{r+1}^2(X)}{6}\right)m\kappa^2}{(\sigma_r^2(X) - \sigma_{r+1}^2(X))^2}$$

$$\leq C_3 \frac{\left(\sigma_r^2(X) + n\kappa^2\right)m\kappa^2}{(\sigma_r^2(X) - \sigma_{r+1}^2(X))^2} = C_3 \frac{\sigma_r^2(X)m\kappa^2}{(\sigma_r^2(X) - \sigma_{r+1}^2(X))^2} + C_3 \frac{nm\kappa^4}{(\sigma_r^2(X) - \sigma_{r+1}^2(X))^2}$$

$$\leq C_4 \left\{\frac{m\kappa^2}{(\sigma_r(X) - \sigma_{r+1}(X))^2} + \frac{\kappa^4 nm}{(\sigma_r(X) - \sigma_{r+1}(X))^4}\right\},$$

where the third inequality follows from the observation that

$$\frac{\sigma_r^2(X)}{(\sigma_r^2(X) - \sigma_{r+1}^2(X))^2} = \frac{\sigma_r^2(X)}{(\sigma_r(X) + \sigma_{r+1}(X))^2(\sigma_r(X) - \sigma_{r+1}(X))^2} \leq \frac{1}{(\sigma_r(X) - \sigma_{r+1}(X))^2},$$

and last display follows from

$$\sigma_r^2(X) - \sigma_{r+1}^2(X) = (\sigma_r(X) + \sigma_{r+1}(X))(\sigma_r(X) - \sigma_{r+1}(X)) \geq \left(\sigma_r(X) - \sigma_{r+1}(X)\right)^2.$$

$\square$

**Corollary 17.** *Suppose the conditions of Theorem 1 hold, in particular the condition in Equation (3) for each mode-$k$ unfolding $\mathcal{M}_k(X^*)$ with target rank $r_k$. Let $U_k^{(0)}$ be the matrix of top $r_k$ left singular vectors of the mode-$k$ unfolding $Y^{(k)} := \mathcal{M}_k(Y)$. Then for each $k \in \{1, 2, 3\}$, with probability at least $1 - C_1 \exp(-C_2 p_k)$,*

$$\left\|\sin\Theta\left(U_k^{(0)}, U_k^*\right)\right\| \leq \frac{1}{2\sqrt{r_{\max}}}.$$

*Proof.* Apply Lemma 16 to $Y^{(k)} = \mathcal{M}_k(Y) \in \mathbb{R}^{p_k \times p_{\neg k}}$ with signal $X^{(k)} = \mathcal{M}_k(X^*)$ and noise $Z^{(k)} = \mathcal{M}_k(Z)$, where $p_{\neg k} := \prod_{j \neq k} p_j$. Use rank $r = r_k$ and identify $n := p_k$ (rows) and $m := p_{\neg k}$ (columns). The lemma (applied to left singular vectors, or equivalently to the transpose) gives

$$\left\| \sin \Theta(U_k^{(0)}, U_k^*) \right\|^2 \leq C \left\{ \frac{p_k \, \kappa^2}{\Delta_k^2} + \frac{\kappa^4 \, p_k \, p_{\neg k}}{\Delta_k^4} \right\},$$

where $\Delta_k := \sigma_{r_k}\big(\mathcal{M}_k(X^*)\big) - \sigma_{r_k+1}\big(\mathcal{M}_k(X^*)\big)$ is the mode-$k$ spectral gap.

By the assumption (3) in Theorem 1 (applied to mode $k$),

$$\Delta_k^2 \geq C_{\text{gap}} \kappa^2 \left( \sqrt{p_k \, p_{\neg k} \, r_{\max}} + r_{\max} \sum_{j=1}^{3} p_j \right).$$

Choosing $C_{\text{gap}}$ sufficiently large makes the right–hand side above at most $1/(4r_{\max})$, hence

$$\left\| \sin \Theta(U_k^{(0)}, U_k^*) \right\| \leq \frac{1}{2\sqrt{r_{\max}}}.$$

The probability bound $1 - C_1 \exp(-C_2 p_k)$ matches the row dimension in the matrix lemma. $\square$

## F    MATRIX PERTURBATION BOUNDS

**Lemma 18.** *Suppose that $A \in \mathbb{R}^{n \times r}$. Then*

$$\|A(A^\top A)^{-1}\| \leq \sigma_r^{-1}(A).$$

*Proof.* If $\sigma_r(A) = 0$, then the desired result trivially follows. So suppose $\text{rank}(A) = r$. Therefore $A^\top A$ is invertible. Let the SVD of $A$ satisfies $A = U_A \Sigma_A V_A^\top$, then

$$\left\| A(A^\top A)^{-1} \right\| = \left\| U_A \Sigma_A V_A^\top (V_A \Sigma_A^2 V_A^\top)^{-1} \right\| = \left\| U_A \Sigma_A^{-1} V_A^\top \right\| = \sigma_{\min}^{-1}(A) = \sigma_r^{-1}(A).$$

$\square$

**Lemma 19.** *Suppose $A \in \mathbb{R}^{m \times n}$ and $B \in \mathbb{R}^{n \times k}$ are any two matrices. Then*

$$\sigma_j(AB) \leq \sigma_j(A)\sigma_{\max}(B).$$

*Proof.* Let $\lambda_j(M)$ denote the $j - th$ eigenvalues of $M$ in the absolute value order. Then

$$\sigma_j(AB) = \sqrt{\lambda_j(ABB^\top A^\top)} \quad \text{and} \quad \sigma_j(A) = \sqrt{\lambda_j(AA^\top)}. \tag{53}$$

Since $BB^\top \preceq \sigma_{\max}^2(B)I_n$, it follows that

$$ABB^\top A^\top \preceq A(\sigma_{\max}^2(B)I_n)A^\top = \sigma_{\max}^2(B)AA^\top.$$

By the monotonicity of eigenvalues under the positive definite matrices, it follows that

$$\lambda_j(ABB^\top A^\top) \leq \sigma_{\max}^2(B)\lambda_j(AA^\top). \tag{54}$$

The desired result follows from (53). $\square$

**Lemma 20.** *Suppose $A \in \mathbb{R}^{m \times n}$ and $B \in \mathbb{R}^{n \times n}$ are any two matrices. Then*

$$\sigma_j(AB) \geq \sigma_j(A)\sigma_{\min}(B).$$

*Proof.* Suppose $\sigma_{\min}(B) = 0$. Then the desired result immediately follows. Therefore it suffices to assume $\sigma_{\min}(B) > 0$ and $B$ is invertible. It suffices to observe that

$$\sigma_j(A) = \sigma_j(ABB^{-1}) \leq \sigma_j(AB)\sigma_{\max}(B^{-1}) = \sigma_j(AB)\sigma_{\min}^{-1}(B),$$

where the inequality follows from Lemma 19.

$\square$

**Lemma 21.** *For any real matrices $A \in \mathbb{R}^{n \times m}$ and $B \in \mathbb{R}^{m \times m}$, it holds that*

$$\|AB\|_{\mathrm{F}} \geq \sigma_{\min}(B) \|A\|_{\mathrm{F}}.$$

*Proof of Lemma 21.* Since $B$ is a square matrix, it follows that

$$\lambda_{\min}(BB^{\top}) = \sigma_{\min}^2(B),$$

where $\lambda_{\min}(\cdot)$ denotes the minimum eigenvalue. Note that

$$BB^{\top} \succeq \lambda_{\min}(BB^{\top}) I_m.$$

Therefore

$$ABB^{\top}A^{\top} \succeq A\{\lambda_{\min}(BB^{\top})I_m\}A^{\top},$$

and so

$$\mathrm{tr}(ABB^{\top}A^{\top}) \geq \mathrm{tr}(A\{\lambda_{\min}(BB^{\top})I_m\}A^{\top}).$$

Then

$$\|AB\|_{\mathrm{F}}^2 = \mathrm{tr}(ABB^{\top}A^{\top}) \geq \mathrm{tr}(A\{\lambda_{\min}(BB^{\top})I_m\}A^{\top}) = \lambda_{\min}(BB^{\top})\,\mathrm{tr}(AA^{\top}) = \sigma_{\min}^2(B)\|A\|_{\mathrm{F}}^2.$$

$\square$

**Lemma 22.** *Let $A \in \mathbb{R}^{p \times q}$ and $U \in \mathbb{O}^{q \times r}$. Then*

$$\|AUU^{\top}\|_{\mathrm{F}} = \|AU\|_{\mathrm{F}}.$$

*Proof.* Observe that

$$\|AUU^{\top}\|_{\mathrm{F}}^2 = \mathrm{tr}(AUU^{\top}UU^{\top}A^{\top}) = \mathrm{tr}(AUU^{\top}A^{\top}) = \|AU\|_{\mathrm{F}}^2.$$

$\square$

**Lemma 23.** *Suppose $B, Z \in \mathbb{R}^{n \times m}$. For all $1 \leq R \leq \min\{n, m\}$, write the full SVD of $A$ as*

$$A = B + Z = \widehat{U}\widehat{\Sigma}\widehat{V}^{\top} = \begin{bmatrix} \widehat{U}_{(R)} & \widehat{U}_{\perp} \end{bmatrix} \cdot \begin{bmatrix} \widehat{\Sigma}_{(R)} & \\ & \widehat{\Sigma}_{\perp} \end{bmatrix} \cdot \begin{bmatrix} \widehat{V}_{(R)}^{\top} \\ \widehat{V}_{\perp}^{\top} \end{bmatrix},$$

*where $\widehat{U}_{(R)} \in \mathbb{O}_{n,R}$, $\widehat{V}_{(R)} \in \mathbb{O}_{m,R}$ correspond to the leading $R$ left and right singular vectors; and $\widehat{U}_{\perp} \in \mathbb{O}_{n,n-R}$, $\widehat{V}_{\perp} \in \mathbb{O}_{m,m-R}$ correspond to their orthonormal complement. We have*

$$\left\|\mathcal{P}_{\widehat{U}_{\perp}} B\right\|_{\mathrm{F}} \leq 3\sqrt{\sum_{j=R+1}^{\min\{n,m\}} \sigma_j^2(B)} + 2\min\left\{\sqrt{R}\|Z\|, \|Z\|_{\mathrm{F}}\right\}$$

$$= 3\left\|B_{(R)} - B\right\|_{\mathrm{F}} + 2\min\left\{\sqrt{R}\|Z\|, \|Z\|_{\mathrm{F}}\right\},$$

*where $B_{(R)}$ denote the rank-$R$ truncated SVD of $B$, this is, if $B = U\Sigma V^{\top}$ then $B_{(R)} := U_{(R)}\Sigma_{(R)}V_{(R)}^{\top}$.*

*Proof.* Without loss of generality, assume $n \leq m$. For $A \in \mathbb{R}^{n \times m}$, let $\Sigma(A) \in \mathbb{R}^{n \times m}$ denote the non-negative diagonal matrices whose diagonal entries are the non-increasingly ordered singular values of $A$. For all $1 \leq R \leq n$, let $B_{(R)}$ denote the truncated SVD of $B$ with rank $R$, and we have by the Eckart–Young–Mirsky theorem

$$\left\|B_{(R)} - B\right\|_{\mathrm{F}} = \sqrt{\sum_{j=R+1}^{n} \sigma_j^2(B)}.$$

For a matrix $A \in \mathbb{R}^{m \times n}$, let $\Sigma(A) \in \mathbb{R}^{m \times n}$ be a non-negative (rectangular) diagonal matrix whose diagonal entries are the non-increasingly ordered singular values of $A$.

We have that

$$
\left\|\mathcal{P}_{\widehat{U}_\perp} B\right\|_F \le \left\|\mathcal{P}_{\widehat{U}_\perp} B_{(R)}\right\|_F + \left\|\mathcal{P}_{\widehat{U}_\perp}(B - B_{(R)})\right\|_F = \sqrt{\sum_{j=1}^R \sigma_j^2(\mathcal{P}_{\widehat{U}_\perp} B_{(R)})} + \left\|\mathcal{P}_{\widehat{U}_\perp}(B - B_{(R)})\right\|_F
$$

$$
\le \sqrt{\sum_{j=1}^R \sigma_j^2(\mathcal{P}_{\widehat{U}_\perp} B_{(R)})} + \left\|B - B_{(R)}\right\|_F = \sqrt{\sum_{j=1}^R \sigma_j^2(\mathcal{P}_{\widehat{U}_\perp} B_{(R)})} + \sqrt{\sum_{j=R+1}^n \sigma_j^2(B)}
$$

$$
\le \left\|(\sigma_1(\mathcal{P}_{\widehat{U}_\perp} B_{(R)}) - \sigma_1(\mathcal{P}_{\widehat{U}_\perp} B), \ldots, \sigma_R(\mathcal{P}_{\widehat{U}_\perp} B_{(R)}) - \sigma_R(\mathcal{P}_{\widehat{U}_\perp} B))^\top\right\|_2 + \left\|(\sigma_1(\mathcal{P}_{\widehat{U}_\perp} B), \ldots, \sigma_R(\mathcal{P}_{\widehat{U}_\perp} B))^\top\right\|_2
$$

$$
+ \sqrt{\sum_{j=R+1}^n \sigma_j^2(B)}
$$

$$
\le \left\|\Sigma(\mathcal{P}_{\widehat{U}_\perp} B_{(R)}) - \Sigma(\mathcal{P}_{\widehat{U}_\perp} B)\right\|_F + \left\|(\sigma_1(\mathcal{P}_{\widehat{U}_\perp} B), \ldots, \sigma_R(\mathcal{P}_{\widehat{U}_\perp} B))^\top\right\|_2 + \sqrt{\sum_{j=R+1}^n \sigma_j^2(B)}
$$

$$
\le \left\|\mathcal{P}_{\widehat{U}_\perp}(B_{(R)} - B)\right\|_F + \sqrt{\sum_{j=1}^R \sigma_j^2(\mathcal{P}_{\widehat{U}_\perp} B)} + \sqrt{\sum_{j=R+1}^n \sigma_j^2(B)}
$$

$$
\le \sqrt{\sum_{j=1}^R \sigma_j^2(\mathcal{P}_{\widehat{U}_\perp} B)} + 2\sqrt{\sum_{j=R+1}^n \sigma_j^2(B)},
$$

where the first equality follows from $\mathrm{rank}(B_{(R)}) = R$, and the fifth inequality follows from Theorem 26. To upper bound $\sqrt{\sum_{j=1}^R \sigma_j^2(\mathcal{P}_{\widehat{U}_\perp} B)}$, we first consider $\sqrt{\sum_{j=1}^R \sigma_j^2(\mathcal{P}_{\widehat{U}_\perp} A)}$. Note that

$$
\mathcal{P}_{\widehat{U}_\perp} A = \sum_{j=R+1}^n \sigma_j(A)\widehat{u}_j \widehat{v}_j^\top,
$$

where $\widehat{u}_j$ and $\widehat{v}_j$ are the left and right singular vectors associated with the $j$th largest singular value $\sigma_j(A)$. Note that $\sigma_j(A) = \sigma_j(B) = 0$ for $j > n$. It follows that

$$
\sqrt{\sum_{j=1}^R \sigma_j^2(\mathcal{P}_{\widehat{U}_\perp} A)} = \sqrt{\sum_{j=R+1}^{2R} \sigma_j^2(A)} = \left\|(\sigma_{R+1}(A), \ldots, \sigma_{2R}(A))^\top\right\|
$$

$$
\le \left\|(\sigma_{R+1}(A) - \sigma_{R+1}(B), \ldots, \sigma_{2R}(A) - \sigma_{2R}(B))^\top\right\| + \left\|(\sigma_{R+1}(B), \ldots, \sigma_{2R}(B))^\top\right\|
$$

$$
\le \min\left\{\sqrt{R}\|Z\|, \|Z\|_F\right\} + \sqrt{\sum_{j=R+1}^n \sigma_j^2(B)}, \tag{55}
$$

where the first inequality follows from the triangle inequality, and second inequality follows from Weyl's inequality (Weyl, 1912), i.e. $|\sigma_j(A) - \sigma_j(B)| \le \|A - B\|$ for all $1 \le j \le n$, as well as the fact that

$$
\left\|(\sigma_{R+1}(A) - \sigma_{R+1}(B), \ldots, \sigma_{2R}(A) - \sigma_{2R}(B))^\top\right\| \le \|\Sigma(A) - \Sigma(B)\|_F \le \|Z\|_F,
$$

where the last inequality follows from Theorem 26. It then follows from (55),

$$
\sqrt{\sum_{j=1}^R \sigma_j^2(\mathcal{P}_{\widehat{U}_\perp} B)} = \left\|(\sigma_1(\mathcal{P}_{\widehat{U}_\perp}(A - Z)), \ldots, \sigma_R(\mathcal{P}_{\widehat{U}_\perp}(A - Z))^\top\right\|
$$

$$
\le \left\|(\sigma_1(\mathcal{P}_{\widehat{U}_\perp}(A - Z)) - \sigma_1(\mathcal{P}_{\widehat{U}_\perp} A), \ldots, \sigma_R(\mathcal{P}_{\widehat{U}_\perp}(A - Z)) - \sigma_R(\mathcal{P}_{\widehat{U}_\perp} A))^\top\right\|
$$

$$
+ \left\|(\sigma_1(\mathcal{P}_{\widehat{U}_\perp} A), \ldots, \sigma_R(\mathcal{P}_{\widehat{U}_\perp} A))^\top\right\|
$$

$$\leq \min\left\{\sqrt{R}\|\mathcal{P}_{\widehat{U}_\perp}Z\|, \|\mathcal{P}_{\widehat{U}_\perp}Z\|_{\mathrm{F}}\right\} + \sqrt{\sum_{j=1}^{R}\sigma_j^2(\mathcal{P}_{\widehat{U}_\perp}A)}$$

$$\leq \min\left\{\sqrt{R}\|Z\|, \|Z\|_{\mathrm{F}}\right\} + \sqrt{\sum_{j=1}^{R}\sigma_j^2(P_{\widehat{U}_\perp}A)}$$

$$\leq 2\min\left\{\sqrt{R}\|Z\|, \|Z\|_{\mathrm{F}}\right\} + \sqrt{\sum_{j=R+1}^{n}\sigma_j^2(B)},$$

where the first two inequalities follow from the same arguments as in (55). Consequently,

$$\left\|\mathcal{P}_{\widehat{U}_\perp}B\right\|_{\mathrm{F}} \leq 3\sqrt{\sum_{j=R+1}^{n}\sigma_j^2(B)} + 2\min\left\{\sqrt{R}\|Z\|, \|Z\|_{\mathrm{F}}\right\}.$$

$\square$

**Lemma 24** (Proposition 1 of Cai & Zhang (2018))**.** *Suppose* $Y \in \mathbb{R}^{m \times n}$, $\widehat{V} = [\widehat{V}_r \ \widehat{V}_\perp] \in \mathbb{O}_n$ *where* $\widehat{V}_r \in \mathbb{O}_{n,r}$, $\widehat{V}_\perp \in \mathbb{O}_{n,n-r}$ *correspond to the first $r$ and last $(n-r)$ right singular vectors of $Y$ respectively. Let* $V = [V_r \ V_\perp] \in \mathbb{O}_{n,n}$ *be any orthogonal matrix with* $V_r \in \mathbb{O}_{n,r}$, $V_\perp \in \mathbb{O}_{n,n-r}$. *Given that* $\sigma_R(YV_r) > \sigma_{r+1}(Y)$, *we have*

$$\|\sin\Theta(V_r, \widehat{V}_r)\| \leq \frac{\sigma_r(YV_r)\|\mathcal{P}_{YV_r}YV_\perp\|}{\sigma_r^2(YV_r) - \sigma_{r+1}^2(Y)} \wedge 1, \tag{56}$$

*where* $\mathcal{P}_A$ *is the projection operator onto the column space of $A$.*

**Lemma 25** (Properties of the $\sin\Theta$ distances )**.**
*The following properties hold for the $\sin\Theta$ distances.*

1. *(Equivalent Expressions) Suppose* $V, \widehat{V} \in \mathbb{O}_{p,R}$. *If $V_\perp$ is an orthogonal extension of $V$, namely* $[V \quad V_\perp] \in \mathbb{O}_p$, *we have the following equivalent forms for* $\|\sin\Theta(\widehat{V}, V)\|$ *and* $\|\sin\Theta(\widehat{V}, V)\|_{\mathrm{F}}$,

$$\|\sin\Theta(\widehat{V}, V)\| = \sqrt{1 - \sigma_{\min}^2(\widehat{V}^T V)} = \|\widehat{V}^T V_\perp\|,$$

$$\|\sin\Theta(\widehat{V}, V)\|_{\mathrm{F}} = \sqrt{r - \|V^T\widehat{V}\|_{\mathrm{F}}^2} = \|\widehat{V}^T V_\perp\|_{\mathrm{F}}.$$

2. *(Triangle Inequality) For all* $V_1, V_2, V_3 \in \mathbb{O}_{p,R}$,

$$\|\sin\Theta(V_2, V_3)\| \leq \|\sin\Theta(V_1, V_2)\| + \|\sin\Theta(V_1, V_3)\|,$$

$$\|\sin\Theta(V_2, V_3)\|_{\mathrm{F}} \leq \|\sin\Theta(V_1, V_2)\|_{\mathrm{F}} + \|\sin\Theta(V_1, V_3)\|_{\mathrm{F}}.$$

3. *(Equivalence with Other Metrics)*

$$\|\sin\Theta(\widehat{V}, V)\| \leq \sqrt{2}\|\sin\Theta(\widehat{V}, V)\|,$$

$$\|\sin\Theta(\widehat{V}, V)\|_{\mathrm{F}} \leq \sqrt{2}\|\sin\Theta(\widehat{V}, V)\|_{\mathrm{F}},$$

$$\|\sin\Theta(\widehat{V}, V)\| \leq \|\widehat{V}\widehat{V}^\top - VV^\top\| \leq 2\|\sin\Theta(\widehat{V}, V)\|,$$

$$\|\widehat{V}\widehat{V}^\top - VV^\top\|_{\mathrm{F}} = \sqrt{2}\|\sin\Theta(\widehat{V}, V)\|_{\mathrm{F}}.$$

**Theorem 26** (Mirsky's singular value inequality in Mirsky (1960))**.** *For any matrices* $A, B \in \mathbb{R}^{m \times n}$, *let* $A = V_1\Sigma(A)W_1^\top$ *and* $B = V_2\Sigma(B)W_2^\top$ *be the full SVDs of $A$ and $B$, respectively. Note that* $\Sigma(A), \Sigma(B) \in \mathbb{R}^{m \times n}$ *are non-negative (rectangular) diagonal matrices whose diagonal entries are the non-increasingly ordered singular values of $A$ and $B$, respectively. Then*

$$\|\Sigma(A) - \Sigma(B)\| \leq \|A - B\| \tag{57}$$

*for any unitarily invariant norm* $\|\cdot\|$ *on* $\mathbb{R}^{m \times n}$.

**Theorem 27** (Weyl's Inequality for Singular Values ). *Let $A, B \in \mathbb{R}^{m \times n}$ and denote their singular values (in nonincreasing order) by $\{\sigma_i(A)\}$ and $\{\sigma_i(B)\}$ respectively. In addition denote the singular values of $A+B$ as $\{\sigma_i(A+B)\}$. Then for all indices $i, j$ satisfying $i+j-1 \leq \min\{m, n\}$,*

$$\sigma_{i+j-1}(A + B) \leq \sigma_i(A) + \sigma_j(B).$$

**Lemma 28** (Ky Fan-type Inequality for Sums of Matrices). *Let $A, B \in \mathbb{R}^{m \times n}$ and denote their singular values (in nonincreasing order) by $\{\sigma_i(A)\}$ and $\{\sigma_i(B)\}$ respectively. In addition denote the singular values of $A + B$ as $\{\sigma_i(A + B)\}$. Then for any $1 \leq k \leq \min\{m, n\}$, it holds that*

$$\sqrt{\sum_{i=1}^{k} \sigma_i^2(A + B)} \leq \sqrt{\sum_{i=1}^{k} \sigma_i^2(A)} + \sqrt{\sum_{i=1}^{k} \sigma_i^2(B)}.$$

*Proof.* For a symmetric matrix $M \in \mathbb{R}^{n \times n}$, by Ky Fan's maximum principle (see e.g. II.1.13 in Bhatia (2013)), for any $1 \leq k \leq n$,

$$\sum_{i=1}^{k} \lambda_i(M) = \sup_{P \in \mathbb{O}^{n \times k}} \operatorname{tr}(P^\top M P).$$

Therefore

$$\sum_{i=1}^{k} \sigma_i^2(A) = \sum_{i=1}^{k} \lambda_i(A^\top A) = \sup_{P \in \mathbb{O}^{n \times k}} \operatorname{tr}(P^\top A^\top A P) = \sup_{P \in \mathbb{O}^{n \times k}} \|AP\|_{\mathrm{F}}^2,$$

and so

$$\sqrt{\sum_{i=1}^{k} \sigma_i^2(A)} = \sup_{P \in \mathbb{O}^{n \times k}} \|AP\|_{\mathrm{F}}.$$

Then

$$\sqrt{\sum_{i=1}^{k} \sigma_i^2(A + B)} = \max_{U \in \mathbb{O}_{n,k}} \|(A + B)U\|_{\mathrm{F}} \leq \max_{U \in \mathbb{O}_{n,k}} \|AU\|_{\mathrm{F}} + \max_{U \in \mathbb{O}_{n,k}} \|BU\|_{\mathrm{F}}$$

$$= \sqrt{\sum_{i=1}^{k} \sigma_i^2(A)} + \sqrt{\sum_{i=1}^{k} \sigma_i^2(B)}.$$

$\square$

Let $\mathcal{T}_{(r_1, r_2, r_3)}$ denote the class of tensor in $\mathbb{R}^{p_1 \times p_2 \times p_3}$ with tucker ranks at most $(r_1, r_2, r_3)$. More precisely

$$\mathcal{T}_{(r_1, r_2, r_3)} = \{A \in \mathbb{R}^{p_1 \times p_2 \times p_3} : \operatorname{rank}(\mathcal{M}_k(A)) \leq r_k, k = 1, 2, 3\}.$$

**Lemma 29.** *Let $X^* \in \mathbb{R}^{p_1 \times p_2 \times p_3}$. For $k \in \{1, 2, 3\}$, suppose the $k$-th matricization of $X^*$ satisfies*

$$\mathcal{M}_k(X^*) = [U_k^* \quad U_{k\perp}^*] \begin{bmatrix} \Sigma_k^* & 0 \\ 0 & \Sigma_{k\perp}^* \end{bmatrix} [V_k^* \quad V_{k\perp}^*]^\top$$

*where $U_k^* \in \mathbb{O}_{p_k, r_k}$ corresponds to the the top $r_k$ singular vectors of $\mathcal{M}_k(X^*)$. Then for $k \in \{1, 2, 3\}$, it holds that*

$$\|X^* \times_k U_{k\perp}^*\|_{\mathrm{F}} = \|X^* \times_k (I_{p_k} - \mathcal{P}_{U_k^*})\|_{\mathrm{F}} = \sqrt{\sum_{j=r_k+1}^{\operatorname{rank}(\mathcal{M}_k(X^*))} \sigma_j^2(\mathcal{M}_k(X^*))} \leq \xi_{(r_1, r_2, r_3)},$$

*where*

$$\xi_{(r_1, r_2, r_3)} = \inf_{A \in \mathcal{T}_{(r_1, r_2, r_3)}} \|A - X^*\|_{\mathrm{F}}.$$

*Proof.* By symmetry, it suffices to consider $k = 1$. Note that

$$\left\|X^* \times_k (I_{p_k} - \mathcal{P}_{U_k^*})\right\|_{\mathrm{F}} = \left\|X^* \times_k \mathcal{P}_{U_{k\perp}^*}\right\|_{\mathrm{F}} = \left\|X^* \times_k U_{k\perp}^*\right\|_{\mathrm{F}}.$$

In addition

$$\left\|X^* \times_1 (I_{p_1} - \mathcal{P}_{U_1^*})\right\|_{\mathrm{F}} = \left\|(I_{p_1} - \mathcal{P}_{U_1^*}) \cdot \mathcal{M}_1(X^*)\right\|_{\mathrm{F}} = \left\|U_{1\perp}^* U_{1\perp}^{*\top} \cdot (U_1^* \Sigma_1^* V_1^{*\top} + U_{1\perp}^* \Sigma_{1\perp}^* V_{1\perp}^{*\top})\right\|_{\mathrm{F}}$$

$$= \left\|U_{1\perp}^* U_{1\perp}^{*\top} \cdot (U_{1\perp}^* \Sigma_{1\perp}^* V_{1\perp}^{*\top})\right\|_{\mathrm{F}} = \sqrt{\sum_{j=r_1+1}^{\mathrm{rank}(\mathcal{M}_1(X^*))} \sigma_j^2(\mathcal{M}_1(X^*))}.$$

Note that by the properties of SVD, for any $W \in \mathbb{R}^{p_1 \times p_2 p_3}$ such that $\mathrm{rank}(W) \le r_1$, it holds that

$$\sqrt{\sum_{j=r_1+1}^{\mathrm{rank}(\mathcal{M}_1(X^*))} \sigma_j^2(\mathcal{M}_1(X^*))} \le \|\mathcal{M}_1(X^*) - W\|_{\mathrm{F}}.$$

For any $A \in \mathcal{T}_{(r_1, r_2, r_3)}$, it holds that $\mathrm{rank}(\mathcal{M}_1(A)) \le r_1$. Therefore for any $A \in \mathcal{T}_{(r_1, r_2, r_3)}$,

$$\sqrt{\sum_{j=r_1+1}^{\mathrm{rank}(\mathcal{M}_1(X^*))} \sigma_j^2(\mathcal{M}_1(X^*))} \le \|\mathcal{M}_1(X^*) - A\|_{\mathrm{F}}.$$

Taking the inf over all $A \in \mathcal{T}_{(r_1, r_2, r_3)}$, it follows that

$$\sqrt{\sum_{j=r_1+1}^{\mathrm{rank}(\mathcal{M}_1(X^*))} \sigma_j^2(\mathcal{M}_1(X^*))} \le \xi_{(r_1, r_2, r_3)}.$$

$\square$

