# OpenReview forum: "Bias-variance Tradeoff in Tensor Estimation"
_ICLR.cc/2026/Conference — Submitted to ICLR 2026_

### Official Review · Reviewer_28be · 2025-10-28

**Soundness:** 2
**Presentation:** 2
**Contribution:** 3
**Rating:** 4
**Confidence:** 4

**Summary:**

This paper is focused on denoising a third-order tensor signal corrupted by additive noise, by means of a low-multilinear-rank approximation.

Its main message is that, by employing a simple spectral algorithm that is essentially a one-step HOOI initialized by truncated HOSVD (THOSVD), the resulting approximation error (with respect to the sought signal) can be decomposed into the sum of a bias term which decays in (each component of) the multilinear rank of the model, plus a variance term which grows in (the components of) the rank.

**Strengths:**

1) This bias-variance analysis is certainly novel and of interest to the community.

2) The raised connections with existing results, in particular those of Zhang and Xia (2018), are also of interest.

**Weaknesses:**

The paper has several presentation issues, thus needing substantial polishing. My main concerns in this regard are:

1) The designed algorithm per se is not novel, simply being a variant of a one-step HOOI initialized by truncated HOSVD (THOSVD).

2) The paper fails to cite and acknowledge important results and previous work on this topic, notably the fact that the THOSVD and similar algorithms such as the sequentially truncated HOSVD are quasi-optimal, as shown in the relatively celebrated paper:
[Vannieuwenhoven, N., Vandebril, R., & Meerbergen, K. (2012). A new truncation strategy for the higher-order singular value decomposition. SIAM Journal on Scientific Computing, 34(2), A1027-A1052.]
In particular, some of the proof techniques used in this paper, such as the telescoping sums introduced to control the approximation error by multilinear projection (see Appendix A), seem to be inspired by the classical proof of this quasi-optimality result, as given in the above paper. These connections should be pointed out.

3) The numerical experiments are in my view ill-designed, and the corresponding section is not sufficiently clear:
- In Section 4.1, it is quite difficult to see differences among the given images. The paper should include zoom boxes or arrows for highlighting specific details which support the conclusions, or even change the parameters in order to illustrate its findings in a more clear way. Also, why does the discussion here stops short of considering high values or rank $r$ for seeing its effect on the variance?
- While the paper focuses on the Tucker model, the synthetic tensors described in 4.2 follow instead an orthogonal canonical polyadic decomposition (due to the diagonal core). Why that choice? Moreover, this model is exactly low-rank (as seen in Table 1, which by the way is redundant since this information is already shown in Table 2), contrarily to the whole point of the Introduction of handling the case of approximately low-rank signals.
- The following conclusion given in Section 4.2 is quite vague, and makes us think that the goal here is to evaluate the given algorithm instead of illustrating the bias-variance tradeoff: "We observe that the error consistently decreases as the SNR parameter $\lambda$ increases. Overall, one-step HOSVD is robust across the tested sizes and ranks, yielding accurate estimates on the synthetic tensors." Yet, if that is the case, then a comparison with other similar algorithms (at least THOSVD as a baseline) is required. But in my view, given that the algorithm is not novel, the paper should rather focus on showing how the utility of the derived result on the bias-variance tradeoff, for instance by plotting separate curves for the bias and the variance terms as a function of the ranks and showing that they follow the predicted trends (bias decay and variance growth) with the correct scaling.
- By the same token, if an exactly low-rank model is to be used, then experiments such as those reported on Table 2 should not stop at a model rank $r$ below the true rank $s$: this would allow seeing the effect of a rank overestimation on the variance.

4) The paper states that the results given for the matrix are a "byproduct" of the tensor analysis, but this claim seems misleading to me. The statement of Theorem 2 is not a particular case of Theorem 1; these results are in fact of a quite different nature: Theorem 1 has a probabilistic flavor and requires a spectral gap condition in order to hold, while Theorem 2 is completely deterministic and valid for any signal. Even the proof arguments are quite dissimilar.

5) Last but not least, a careful revision is in order. In particular:
- Several results referred to as Theorems (for instance, in lines 280 and 281) are actually other types of results (Lemma, Corollary).
- It seems that the constant in (6) should be denoted by C, according to the discussion in lines 242-243.

**Questions:**

1) Would it be possible have a deterministic result similar to that of Theorem 2 for tensors, with a bound depending on the model degrees of freedom and, say, some norm (spectral?) of the noise tensor?

2) The paper argues that its results extend in a way those of Zhang & Xia, and in particular match those previous results in the case of an exactly low-rank model. Have the authors also checked that their bounds match the known results for rank-one tensor PCA? Namely, for $p_1=p_2=p_3=p$, an appropriately normalized noise (say, $\kappa = 1/\sqrt{p}$) and $r_1=r_2=r_3=1$, the condition on the signal magnitude for having an $O(1)$ error should match the known conjectured computational threshold $p^{1/4}$. It seems to me that this is the case, which would be interesting to point out.

---

> ### Author Response · Authors · 2025-11-20
> **Response to the Reviewer 28be**
>
> [1/2]
>
> We sincerely thank the reviewer for their careful and detailed assessment, and we greatly appreciate their recognition that our bias–variance tradeoff is of interest to the community. In the following, we address their comments point-by-point.
>
> > The designed algorithm per se is not novel, simply ... (THOSVD).
>
> We acknowledge that the estimator in Algorithm 1 is a simple variant of HOSVD/one-step HOOI. Our aim in this work is not to propose a new algorithmic framework, but to obtain a clean, rigorous bias–variance tradeoff for this estimator in the realistic setting where the signal is not exactly low rank. Our analysis shows that the one-step HOSVD algorithm can handle ground-truth tensors of arbitrary ranks, rather than being restricted to exactly low-rank models as in prior works. In particular, we develop a new technique to characterize the optimal bias term in tensor estimation, and our analysis achieves a clean, rank-adaptive bias–variance tradeoff for any user-specified ranks.\
> In addition, in the revision we further complement these upper bounds with matching minimax lower bounds for both the matrix and tensor models, which, to our knowledge, have not previously appeared in the literature.
>
> > The paper fails to cite ... should be pointed out.
>
> We thank the reviewer for pointing out this related work and apologize for the omission. In the revised manuscript, we now cite Vannieuwenhoven et al. (2012) and briefly acknowledge the connection at beginning of the proof of Theorem 1.
>
> > In Section 4.1, it is quite difficult to see differences ... variance?
>
> We thank the reviewer for this helpful suggestion. Following their feedback, we have refined and detailed our real-data experiments on the MRI volumes. In the revised manuscript, we now analyze five volumes and place particular emphasis on intermediate and higher ranks under large noise levels (large $\\lambda$). In addition, we provide a quantitative summary of these results in Table 1.
>
> > While the paper focuses on the ... low-rank signals.
>
> We thank the reviewer for pointing this out. In the revised manuscript, we have updated the synthetic tensor experiment in Section 4.2 in two ways: (i) we now consider a full-rank tensor model rather than an exactly low-rank one, and (ii) we use a non-diagonal core so that the setting corresponds to a genuine Tucker model rather than merely its special case, the orthogonal CP decomposition (i.e. Tucker with a diagonal core). These changes better align the simulations with our framework.
>
> > The following conclusion given ... scaling.
>
> We agree the need to illustrate the bias–variance tradeoff. To this end, in addition to the error table reported in Section 4.2, we now consider a representative synthetic tensor example and explicitly compute the bias and variance contributions separately. The resulting curves, shown in Figure 3 of Appendix A for both $\lambda \in \\{10,50\\}$, exhibit the predicted behavior of decreasing bias and increasing variance as the target ranks grow, in line with our theoretical results.
>
> > By the same token ... on the variance
>
> We thank the reviewer for this suggestion. As discussed above, we have updated the synthetic tensor experiment to focus on a full-rank setting in the revised manuscript.\
> In addition, to directly examine the effect of rank overestimation, we ran an auxiliary experiment where the target rank $r$ is chosen above the true rank $s$ where the core $\mathcal{G} \in \mathbb{R}^{s\times s \times s}$. The resulting relative errors (mean and standard deviation over 50 trials) for $\lambda\in\\{10,50\\}$ are reported in the table below.
>
> *Table: Relative error under exact and over-estimated ranks.*
> |$\lambda$|$p$|$s$|$r$|Mean (SE)|
> |-|-|-|-|-|
> |10|20|12|12|0.05047 (0.00081)|
> |10|20|12|18|0.08869 (0.00072)|
> |10|50|25|25|0.04099 (0.00025)|
> |10|50|25|40|0.07607 (0.00018)|
> |10|75|20|25|0.02308 (0.00013)|
> |10|75|20|50|0.05936 (0.00012)|
> |10|100|60|60|0.05113 (0.00007)|
> |10|100|60|80|0.07512 (0.00007)|
> |50|20|12|12|0.01004 (0.00016)|
> |50|20|12|18|0.01774 (0.00015)|
> |50|50|25|25|0.00758 (0.00004)|
> |50|50|25|40|0.01505 (0.00004)|
> |50|75|20|25|0.00457 (0.00003)|
> |50|75|20|50|0.01184 (0.00002)|
> |50|100|60|60|0.01015 (0.00002)|
> |50|100|60|80|0.01499 (0.00001)|
>
> The overestimation of the rank increases the error, reflecting the growth of the variance term.
>
> > The paper states that the results ... dissimilar.
>
> Our intent is to emphasize that the matrix and tensor analyses share a similar underlying bias–variance decomposition, rather than to claim that Theorem 2 is a corollary of Theorem 1. In the final version, we will instead write: "As a related result, we also obtain a convenient bias–variance decomposition for the vanilla low-rank SVD matrix estimators."
>
> > Last but not least ... in lines 242-243
>
> We thank the reviewer for pointing this out. In the revised manuscript, we have carefully proofread the text and corrected these typos, inconsistent labels, and related minor mistakes.

---

> > ### Comment · Reviewer_28be · 2025-11-24
> >
> > I appreciate the authors' efforts in addressing the concerns raised by all Reviewers.
> >
> > As far as my remarks are concerned, the current version has been improved with respect to the original submission.
> >
> > However, I still think that Section 4 (Numerical experiments) is not satisfying in its current form. By inspecting that section, one gets the impression that the goal here is that of assessing the studied algorithm (one-step THOSVD), which is not one of the paper's contributions. Indeed, in the discussion of the results given in the main document, there is no mention of the bias-tradeoff, which should be the central focus in my view.
> >
> > Anyway, because the theoretical results are interesting and useful, I am willing to slightly raise my rating.

---

> ### Author Response · Authors · 2025-11-20
>
> [2/2]
>
> > Would it be possible have a deterministic result similar to that of Theorem 2 for tensors, with a bound depending on the model degrees of freedom and, say, some norm (spectral?) of the noise tensor?
>
> We agree this is an interesting direction. In our current tensor results, the bias component is deterministic, whereas the variance term is controlled via concentration inequalities. We note that it is in principle possible to develop a fully deterministic result for tensor analysis and express the variance term directly in terms of the noise tensor $Z$. However, we believe that such a result would be difficult to summarize in an intuitive, reader-friendly way. Therefore, we choose to present the variance term using parameters of the tensors instead, in line with the existing literature. Developing results with explicit dependence on the model degrees of freedom and a suitable norm of the noise tensor is an interesting direction for future work.
>
> > The paper argues that its results extend in a way ... would be interesting to point out
>
> Thank you for this helpful suggestion. Consider the rank-one spiked model $X^\ast = \lambda\, u \otimes v \otimes w$ with $p_1 = p_2 = p_3 = p, r_1 = r_2 = r_3 = 1$, and noise entries $Z_{i,j,k} \overset{\text{i.i.d.}}{\sim} \mathrm{subGaussian}(0,\kappa^2)$. For each mode-$k$ unfolding we have $\sigma_1(\mathcal{M}_k(X^\ast))=\tau$ and $\sigma_2(\mathcal{M}_k(X^\ast)) = 0$.  Our singular–gap  condition therefore reduces to
>
> $$
> \tau^2 \ge C_{\\mathrm{gap}} \kappa^2\big(p^{3/2} + 3p\big),
> $$
> that is, $\tau \gtrsim \kappa p^{3/4}$ for large $p$.  This matches, up to constants, the currently known computational threshold for order-3 tensor PCA in the spiked model  $T = \tau v^{\otimes 3} + A$ with $A_{i,j,k} \overset{\text{i.i.d.}}{\sim} \mathcal{N}(0,1)$ , where polynomial-time algorithms are known to succeed once $\tau \gtrsim p^{3/4}$ (Anandkumar et al., 2017). We will add a  remark after Theorem 1 pointing out this specialization in the final version of the paper, space permitting.
>
>
>
>
> __Anandkumar, Anima, et al.__ "_Homotopy analysis for tensor PCA._" Conference on Learning Theory. PMLR, 2017.

---

> ### Author Response · Authors · 2025-11-28
>
> Thank you for your positive note on the improvements in the revised manuscript and for raising your rating. Your constructive feedback has been very helpful in strengthening the work.
>
> As the table below illustrates, when the chosen rank is too low (e.g., $r=20$) the estimator is overly biased and the reconstruction error is high, whereas when the rank is too large (e.g., $r=95$) the variance dominates and the error increases again. For intermediate ranks the errors are smallest. This empirical behavior across all $\lambda$ and all three variants (original, db6, sym8) is consistent with the predicted bias–variance tradeoff.
>
> | λ   | r  | original        | db6             | sym8            |
> | --- | -- | --------------- | --------------- | --------------- |
> | 0.4 | 20 | 0.2798 (0.0226) | 0.2792 (0.0386) | 0.2719 (0.0310) |
> | 0.4 | 35 | 0.2129 (0.0332) | 0.2099 (0.0541) | 0.2032 (0.0464) |
> | 0.4 | 50 | 0.1753 (0.0330) | 0.1747 (0.0537) | 0.1681 (0.0466) |
> | 0.4 | 65 | 0.1553 (0.0268) | 0.1561 (0.0435) | 0.1501 (0.0373) |
> | 0.4 | 80 | 0.1490 (0.0183) | 0.1499 (0.0291) | 0.1451 (0.0245) |
> | 0.4 | 95 | 0.1550 (0.0102) | 0.1558 (0.0159) | 0.1525 (0.0128) |
> | 0.5 | 20 | 0.2802 (0.0226) | 0.2796 (0.0386) | 0.2722 (0.0310) |
> | 0.5 | 35 | 0.2144 (0.0330) | 0.2114 (0.0537) | 0.2048 (0.0461) |
> | 0.5 | 50 | 0.1797 (0.0322) | 0.1793 (0.0521) | 0.1728 (0.0451) |
> | 0.5 | 65 | 0.1655 (0.0248) | 0.1668 (0.0400) | 0.1610 (0.0341) |
> | 0.5 | 80 | 0.1677 (0.0154) | 0.1691 (0.0244) | 0.1647 (0.0201) |
> | 0.5 | 95 | 0.1840 (0.0073) | 0.1847 (0.0116) | 0.1819 (0.0089) |
> | 0.6 | 20 | 0.2806 (0.0225) | 0.2801 (0.0385) | 0.2727 (0.0309) |
> | 0.6 | 35 | 0.2161 (0.0327) | 0.2133 (0.0532) | 0.2067 (0.0455) |
> | 0.6 | 50 | 0.1851 (0.0311) | 0.1850 (0.0503) | 0.1787 (0.0434) |
> | 0.6 | 65 | 0.1776 (0.0224) | 0.1793 (0.0362) | 0.1738 (0.0307) |
> | 0.6 | 80 | 0.1890 (0.0124) | 0.1904 (0.0200) | 0.1866 (0.0162) |
> | 0.6 | 95 | 0.2150 (0.0049) | 0.2154 (0.0085) | 0.2132 (0.0062) |
>
> **Table:** Mean relative reconstruction error (Mean (SE) across MRI volumes).
>
> Please let us know if you have any additional questions or concerns. We would be happy to address them during the discussion period.

---

### Official Review · Reviewer_eK8H · 2025-11-01

**Soundness:** 3
**Presentation:** 2
**Contribution:** 2
**Rating:** 4
**Confidence:** 4

**Summary:**

This work studies the problem of noisy tensor estimation when the the ground-truth tensor is not necessarily Tucker low-rank. By utilizing some  classical linear algebra results such as Mirsky’s and Ky Fan’s theorems, the authors derive a clean rank-adaptive bias–variance tradeoff. A series of simulations is carried out to confirm the existence of tradeoffs across different regimes of spectral decay and noise.

**Strengths:**

1. A new tensor estimation setting has been not considered in the literature.
2. A rigorous theoretical analysis is carried out to show clean the rank-adaptive bias–variance tradeoff in such a tensor estimation problem.
3. As a byproduct, a convenient bias-variance tradeoff in matrix estimation has been obtained.

**Weaknesses:**

1. The minx lower bounds for both tensor and matrix cases are not provided.
2. The main theoretical derivations seem to be direct extensions of the work Zhang & Xia (2018).
3. How to tune the target Tucker rank for the proposed Algorithm 1 has not been provided.

**Questions:**

1. Why the authors consider the simple one-step HOSVD algorithm rather than the popular HOOI algorithm?
2. Can the proposed proof strategy be readily extended to accommodate general $d$th-order tensors, and if so, what specific steps or modifications would be required to achieve this?

---

> ### Author Response · Authors · 2025-11-20
> **Response to the Reviewer eK8H**
>
> We sincerely thank the reviewer for their constructive feedback, and greatly appreciate their recognition of our tensor setting, the rigor of our theoretical analysis, and the usefulness of the resulting bias–variance tradeoff for both tensors and matrices. In the following, we address their comments point-by-point.
>
> > The minx lower bounds for both tensor and matrix cases are not provided.
>
> We thank the reviewer for pointing this out. In the revised manuscript, we now provide minimax lower bound results for both the tensor and matrix settings as Theorem 2 and Theorem 7 respectively. Their proofs are added in the Section B of the revised manuscript.
>
> > The main theoretical derivations seem to be direct extensions of the work Zhang & Xia (2018).
>
> We agree that existing variance bounds provide an natural starting point, but our contributions go beyond a direct extension. Our analysis shows that a one-step HOSVD algorithm can handle ground-truth tensors of arbitrary ranks, rather than being restricted to exactly low-rank models as in prior works. In particular, our analysis achieves a clean, rank-adaptive bias–variance tradeoff for any user-specified ranks.\
> Regarding proof techniques, we leverage classical linear algebra tools, such as Mirsky’s and Ky Fan’s theorems, to develop a pipeline for revealing  the optimal approximation error in tensor estimation. To the best of our knowledge, this has not been explored in the existing literature.\
> Finally, in the revision we further complement these upper bounds with matching minimax lower bounds for both the matrix and tensor models, which, to our knowledge, have not previously appeared in the literature.
>
> > How to tune the target Tucker rank ... been provided.
>
> There are several well-known ways to select the target Tucker ranks. We have added the following remark on how to select ranks in a data-driven way in Section A of the appendix of the revised manuscript:\
> "To select the rank parameters in a data-driven way, we follow the discussion in (Ozdemir et al., 2017) and adopt an adaptive thresholding scheme. More precisely, we choose $ r_{k} $ so that $ \\sigma^{2}_{r_k}  ( \\mathcal{M}_k (Y) ) \\ge \\tau $, where $ \\tau $ is a threshold.
>
> A simple choice is $\\tau = \\alpha \\| Y \\|^2_{\\mathrm{F}}$ with $\\alpha = 0.01$; we then retain all singular values of $\\mathcal{M}_{k}(Y)$ up to the largest index $j$ satisfying
>
> $\\sigma_{j}^{2}(\\mathcal{M}_k(Y)) \\ge \\tau$ and discard the rest."
>
>
>
> > Why the authors consider the simple one-step HOSVD algorithm rather than the popular HOOI algorithm?
>
> We focus on the simple one-step HOSVD because,   our analysis shows that a single step already achieves the minimax rate. Therefore, additional iterations may not be necessary from a statistical perspective. By contrast, the vanilla HOOI typically requires additional iterations.\
> In practice, it could be hard to come up with a principled stopping rule for the iterations, so further iterations can be computationally costly. Extra HOOI iterations may be beneficial in certain regimes, but understanding their precise statistical gains requires a separate, more delicate analysis, which we view as an interesting direction for future work.
>
> > Can the proposed proof strategy be readily extended ... if so, what specific steps or modifications would be required to achieve this?
>
> We thank the reviewer for this question. Yes, the proof strategy extends naturally to general $d$-th order tensors.\
> Concretely, consider an order-$d$ tensor with dimensions $p_1,\dots,p_d$ and target ranks $r_1,\dots,r_d$. Under an SNR (singular-gap) condition for each $k = 1,\dots,d,$ similar to as in Theorem 1, we can extend our analysis to show that
> $$
> \\| \\widetilde{X} - X^\ast \\|^2_{\\mathrm{F}} = O \\Big( \\kappa^2 \\Big\\{ \\prod_{k =1}^d r_k + \\sum_{k=1}^d p_{k} r_{k} \\Big\\} \\; + \\; \\xi_{(r_{1},\\ldots,r_{d})}^2 \\Big) \\quad \\text{  with high probability.}
> $$
> At the level of the proof, each of the steps used in Theorem 1 has a $d$-mode analogue:
> - The high-probability events in (16)–(18) are replaced by their $d$-mode versions, obtained via extensions of Lemma 10, Lemma 11 and Corollary 17 to $k=1,\dots,d$. This gives the required deviation bounds for projected sub-Gaussian noise in every mode.
> - The bias residual representation (Step 2) is written once in a mode-symmetric way and then analyzed for a generic mode $k$, and the same argument is reused across all $k$.
> - The remaining parts of the argument (control of the variance term and combination with the bias) follow the same structure as in the third-order case, with more cumbersome but conceptually identical bookkeeping over $d$ modes.
>
> We focus on the third-order case $d=3$ in this work to avoid overburdening the presentation  while   illustrating the bias–variance trade-off. In the final version of the paper, we will include a remark in the Appendix that expands the discussion of our results to cover higher-order tensor settings.

---

> > ### Comment · Reviewer_eK8H · 2025-11-25
> >
> > I appreciate the authors' detailed response to my comments. I would like to raise my score to 6

---

> > > ### Author Response · Authors · 2025-11-28
> > >
> > > Thank you for your thoughtful feedback and for acknowledging the detailed response. We appreciate your updated rating and are grateful that your comments helped improve the manuscript.
> > >
> > > If you have any additional questions or concerns, we would be happy to address them during the discussion period.

---

### Official Review · Reviewer_GWqa · 2025-11-02

**Soundness:** 4
**Presentation:** 4
**Contribution:** 3
**Rating:** 8
**Confidence:** 5

**Summary:**

The paper proves a new bias-variance tradeoff for Tucker decompositions of general third-order tensors.  No assumption is made on the input tensor being close to low-rank, which makes the results highly applicable to real data.  The authors provide demonstrations with real datasets to reinforce this.  Overall, this is a very nice submission.

**Strengths:**

- No assumption is made on the input tensor being exactly low-rank or close to low-rank.  This makes the result applicable to any input tensor, and thus very useful.

- As the authors explain, the proven bound is optimal up to constants.

- The authors show that the HOSVD algorithm achieves the bound.

- There are nice real data illustrations.

- The paper is very clearly written.

**Weaknesses:**

- It would have been nice if the authors extended their analysis to tensors beyond third order tensors.

- A few useful references are missing.

**Questions:**

(1) Line 35: missing both older and more recent papers on latent variable learning.  Obviously the authors should cite
	Animashree Anandkumar, Rong Ge, Daniel J Hsu, Sham M Kakade, Matus Telgarsky, et al. "Tensor decompositions for learning latent variable models." J. Mach. Learn. Res., 15(1):2773–2832, 2014.
here instead of the community detection part.
I also suggest the authors cite
	Yifan Zhang, Joe Kileel, "Moment estimation for nonparametric mixture models through implicit tensor decomposition", SIAM Math. Data Sci. 5.4 (2023), 1130-1159.
as well as
	Samantha Sherman, Tamara G. Kolda. "Estimating higher-order moments using symmetric tensor decomposition." SIAM Matrix Anal. Appl. 41.3 (2020), 1369-1387.



(2) Lines 43-50: regarding the low-rank Tucker tensor decomposition, the authors should cite
	Ruhui Jin, Joe Kileel, Tamara G. Kolda, Rachel Ward. "Scalable symmetric Tucker tensor decomposition." SIAM Matrix Anal. Appl. 45.4 (2024), 1746-1781



(3) Can the authors add some remarks on how the analysis should go for higher than third order tensors?

---

> ### Author Response · Authors · 2025-11-20
> **Response to the Reviewer GWqa**
>
> We sincerely thank the reviewer for their thoughtful feedback and greatly appreciate their recognition of the generality and practical usefulness of our results. In the following, we address their comments point-by-point.
>
> > A few useful references are missing.
>
> > Line 35: missing both older and more recent papers on latent variable learning. Obviously the authors should cite Animashree Anandkumar, ... 2014. here instead of the community detection part. I also suggest the authors cite Yifan Zhang ... SIAM Math. Data Sci. 5.4 (2023), 1130-1159. as well as Samantha Sherman ... SIAM Matrix Anal. Appl. 41.3 (2020), 1369-1387.
>
> > Lines 43-50: regarding the low-rank Tucker tensor decomposition, the authors should cite Ruhui Jin, Joe Kileel, Tamara G. Kolda, Rachel Ward. "Scalable symmetric Tucker tensor decomposition." SIAM Matrix Anal. Appl. 45.4 (2024), 1746-1781
>
> We thank the reviewer for pointing out these important related works and apologize for the oversight. In the revised manuscript, we have added the suggested citations.
>
> > Can the authors add some remarks on how the analysis should go for higher than third order tensors?
>
> Thank you for this insightful question. The analysis extends naturally to general $d$-th order tensors. Concretely, consider an order-$d$ tensor with dimensions $p_1,\dots,p_d$ and target ranks $r_1,\dots,r_d$, and let \(r_{\max} = \max_k r_k\). Under an SNR (singular-gap) condition for each $k = 1,\dots,d$, similar to as in Theorem 1, we can extend our analysis to show that
> $$
> \\| \\widetilde{X} - X^\ast \\|^2_{\\mathrm{F}} = O \\Big( \\kappa^2 \\Big\\{ \\prod_{k =1}^d r_k + \\sum_{k=1}^d p_{k} r_{k} \\Big\\} \\; + \\; \\xi_{(r_{1},\\ldots,r_{d})}^2 \\Big) \\quad \\text{  with high probability.}
> $$All four steps in the proof of Theorem 1 extend to this setting with more involved but routine technical modifications. We focus on the third-order case $d=3$ in this work to avoid overburdening the presentation  while   illustrating the bias–variance trade-off. In the final version of the paper, we will include a remark in the Appendix that expands the discussion of our results to cover higher-order tensor settings

---

### Official Review · Reviewer_6hRm · 2025-11-02

**Soundness:** 4
**Presentation:** 3
**Contribution:** 3
**Rating:** 6
**Confidence:** 4

**Summary:**

This paper investigates the problem of denoising a third-order tensor $Y = X^* + Z$, where $Y$ is the observed noisy tensor, $ X^* $ is the ground-truth signal, and $ Z $ is a noise tensor. Existing work mainly assumes that $ X^* $ is of low Tucker-rank as compared to the dimension. In contrast, this paper does not impose this assumption, and considers a simple variant of the HOSVD algorithm. They show that the recovery error of this algorithm can be upper bounded by the sum of two terms: a bias term that depends on the best achievable estimation error of $X^*$ over the space of tensors of low Tucker-rank $(r_1, r_2, r_3)$, a variance term that characterizes the estimation error if the true signal itself is indeed of Tucker-rank $(r_1, r_2, r_3)$.

As a byproduct, the paper provides a similar (and even simpler) non-asymptotic bias-variance decomposition for the standard truncated SVD estimator for matrices.

**Strengths:**

This paper's main result (Theorem 1, the bias-variance tradeoff error bound) is novel and interesting, as it extends the exact low-rank setting, which is common in existing literature, to the approximate low-rank setting. This is a natural generalization and one would expect both the variance and bias terms in the error bound to be optimal up to a constant. Further, this error bound holds uniformly for all low Tucker-rank tensors, making it adaptive to all target Tucker ranks. The paper is also clearly written.

**Weaknesses:**

(i) Given the upper bound on the variance term in existing literature, it seems to me that the main contributions of this paper is a bit incremental, especially that the proof techniques are also similar.
(ii) Theorem 1 has an assumption on the singular value gap of the true signal $X^*$. Is this generally satisfied in practice? It would be better to give some examples for which this assumption holds.

**Questions:**

Is it possible to compute the bias term $\xi (r_1, r_2, r_3)$ for some examples? Again, this would be helpful in understanding how the approximate low-rank case compares to the exact low-rank case.

---

> ### Author Response · Authors · 2025-11-20
> **Response to the Reviewer 6hRm**
>
> We thank the reviewer for their valuable feedback and for highlighting the adaptivity and tightness of our results. In the following, we address their comments point-by-point.
>
> > Given the upper bound on the variance term in existing literature, it seems to me that the main contributions of this paper is a bit incremental, especially that the proof techniques are also similar.
>
> We agree that existing variance bounds provide an natural starting point, but our contributions go beyond a direct extension. Our analysis shows that a one-step HOSVD algorithm can handle ground-truth tensors of arbitrary ranks, rather than being restricted to exactly low-rank models as in prior works. In particular, our analysis achieves a clean, rank-adaptive bias–variance tradeoff for any user-specified ranks.\
> Regarding proof techniques, we leverage classical linear algebra tools, such as Mirsky’s and Ky Fan’s theorems, to develop a pipeline for revealing  the optimal approximation error in tensor estimation. To the best of our knowledge, this has not been explored in the existing literature.\
> Finally, in the revision we further complement these upper bounds with matching minimax lower bounds for both the matrix and tensor models, which, to our knowledge, have not previously appeared in the literature.
>
> > Theorem 1 has an assumption on the singular value gap of the true signal $X^\ast$. Is this generally satisfied in practice? It would be better to give some examples for which this assumption holds.
>
> Thanks for this insightful question. Indeed, the singular value gap assumption can often be satisfied in applications. We computed the singular value gap spectrum over 50 randomly drawn T1-weighted MRI volumes from the IXI dataset. The medians and interquartile ranges, reported in Section A of the appendix of the revised manuscript, show that the empirical medians are on the order of at least 100 over a wide range of ranks for all modes. This provides concrete evidence for the practicality of this assumption. Moreover,if our singular gap condition does not hold, then Zhang \& Xia (2018, Sec.~4.2) show that no polynomial-time algorithm can provide a consistent estimate of the ground-truth tensor. This indicates that our singular gap condition is computationally optimal.
>
> > Is it possible to compute the bias term $\xi(r_{1},r_{2},r_{3})$ for some examples? Again, this would be helpful in understanding how the approximate low-rank case compares to the exact low-rank case.
>
> Thank you for this helpful suggestion. To empirically quantify the two contributions, we explicitly computed the bias and variance terms in our synthetic tensor experiment; the results are reported in Figure 3 of Section A of the revised appendix. We observe that, as the target rank increases, the bias term decays rapidly.

---

### Author Response · Authors · 2025-11-20
**Main revision update**

We sincerely thank all four reviewers for their time and thoughtful feedback. We also appreciate their recognition of the strength and generality of our theoretical results, as well as their comments on the accessibility and applicability of our analysis.

We would like to point out that our manuscript provides a framework to handle ground-truth tensors of arbitrary ranks, and our analysis achieves a clean, rank-adaptive bias–variance tradeoff. To our knowledge, these results have not been well-explored in the existing literature.
In addition, we have made the following main revisions to the manuscript:
- Following Reviewer eK8H's suggestion, we added matching lower bound results as Theorem 2 and Theorem 7 with their proofs in Section B of the appendix for both the tensor and matrix settings, which establish that our results are minimax optimal.
- We added the additional references suggested by the Reviewers GWqa and 28be.
- We have revised the tensor simulation setting to employ a full-rank model and have replaced the diagonal core tensor with a full core tensor, as suggested by Reviewer 28be.
- Following the reviewers’ feedback, we have refined our real-data experiments on MRI volumes. In the revised manuscript, we now analyze five volumes and place particular emphasis on intermediate and higher ranks under large noise levels (large $\lambda$). In addition, we provide a quantitative summary of these results in Table 1.
- To address Reviewer 6hRm's concern, we validated the singular-gap assumption on our real-data examples; this is now documented in Section A of the appendix. This provides concrete evidence that singular-gap assumption is not merely an idealized technical assumption of the analysis, but is indeed observed in practice.
- For the synthetic tensor setup, we selected a representative example to empirically quantify the individual contributions of bias and variance. The new Figure 3 in Section A of the appendix illustrates this behavior.
- We corrected several typographical and presentation errors pointed out by the reviewers.

All changes in the revised manuscript are highlighted in red.

---

### Meta-Review · Area_Chair_5yN1 · 2026-01-04

**Summary:**

Several concerns regarding both the theoretical and empirical aspects were raised by the reviewers.
On the theoretical side, reviewers pointed out the incremental nature of the main theorem. In their response, the authors argue that the novelty lies in showing that a one-step HOSVD algorithm can handle ground-truth tensors of arbitrary rank, rather than being restricted to exactly low-rank models as in prior work, thereby yielding a clean, rank-adaptive bias–variance tradeoff. This partially addresses the concern, and I acknowledge this point.
On the empirical side, reviewers mentioned issues with the design of the experimental evaluation. While the empirical results should directly evaluate the paper’s main claim of the bias-variance tradeoff, the primary experiments instead focus on overall reconstruction error.
I have carefully read the paper and agree with these concerns. In addition, the quality of the presentation remains suboptimal. Although it has been partially improved in the revised version, the manuscript remains redundant in several places. Moreover, results for higher-order tensors are important and should be incorporated earlier in the paper.

In summary, while the theoretical findings of this paper are potentially interesting, the work requires at least one additional round of major revision. I therefore recommend rejection.

**Reviewer Concerns:**

- Concerns regarding the novelty of the theoretical results and the assumptions made in the theoretical analysis have been partially addressed in the rebuttal.
- Concerns regarding the empirical evaluation have not been properly addressed. Although the authors added results on bias and variance using synthetic data, these additions are insufficient, and the issues remain outstanding.
- Other issues related to the discussion of prior work and the quality of presentation have been addressed in the revision, while I believe they also remain outstanding.

**Reviewer Scores:**

**Reviewer 6hRm** raised concerns about the novelty of the theoretical results and the validity of the assumptions used in the analysis. The authors attempted to address these issues, so there may be a possibility that the reviewer will change the score.

**Reviewer GWqa** raised concerns regarding higher-order tensors and missing references. These issues have been addressed by the authors, and therefore this reviewer is likely to maintain the favorable score.

**Reviewer eK8H** raised several concerns about the novelty, motivation, and significance of the findings. These issues have been partially addressed in the revision, and as a result, the reviewer has increased the score.

The main concern of **Reviewer 28be** relates to the experimental design, which has not been sufficiently addressed in the rebuttal. However, the reviewer appears willing to increase the score, as other concerns have been addressed.

---

### Decision · Program_Chairs · 2026-01-26

Reject